# The Potential of Open-Access Data for Flood Estimations: Uncovering Inundation Hotspots in Ho Chi Minh City, Vietnam, through a Normalized Flood Severity Index

Leon Scheiber[1,*], Mazen Hoballah Jalloul[1], Christian Jordan[1], Jan Visscher[1], Hong Quan Nguyen[2,3] and Torsten Schlurmann[1]

[1]Ludwig-Franzius-Institute of Hydraulic, Estuarine and Coastal Engineering, Leibniz University of Hannover, D-30167 Hannover, Germany
[2]Institute for Circular Economy Development, Vietnam National University – Ho Chi Minh City, 700000 Ho Chi Minh City, Vietnam
[3]Institute for Environment and Resources, Vietnam National University – Ho Chi Minh City, 700000 Ho Chi Minh City, Vietnam

*Correspondence to*: Leon Scheiber (scheiber@lufi.uni-hannover.de)

**Abstract.** Hydro-numerical models are increasingly important to determine the adequacy and evaluate the effectiveness of potential flood protection measures. However, a significant obstacle in setting up hydro-numerical and associated flood damage models is the tedious and oftentimes prohibitively costly process of acquiring reliable input data, which particularly applies to coastal megacities in developing countries and emerging economies. To help alleviate this problem, this paper explores the usability and reliability of flood models built on open-access data in regions where highly-resolved (geo)data are either unavailable or difficult to access, yet where knowledge about elements at risk is crucial for mitigation planning. The example of Ho Chi Minh City, Vietnam, is taken to describe a comprehensive, but generic methodology for obtaining, processing and applying the required open-access data. The overarching goal of this study is to produce preliminary flood risk maps that provide first insights into potential flooding hotspots demanding closer attention in subsequent, more detailed risk analyses. As a key novelty, a normalized flood severity index ($I_{NFS}$), which combines flood depth and duration, is proposed to deliver key information in a preliminary flood hazard assessment. This index serves as an indicator that further narrows down the focus to areas where flood hazard is significant. Our approach is validated by a comparison with more than 300 flood samples locally observed during three heavy rain events in 2010 and 2012, which correspond to $I_{NFS}$-based inundation hotspots in over 73% of all cases. These findings corroborate the high potential of open-access data in hydro-numerical modeling and the robustness of the newly introduced flood severity index, which may significantly enhance the interpretation and trustworthiness of risk assessments in the future. The proposed approach and developed indicators are generic and may be replicated and adopted in other coastal megacities around the globe.

*Keywords:* urban flooding, disaster risk, open-access data, numerical modelling, surface runoff model, Ho Chi Minh City, Southeast Asia

## 1 Introduction

With more than half a million deaths between 1980 and 2009 and nearly three billion people affected, flood events are
doubtlessly the most common and impactful natural disasters worldwide (Hong et al., 2018; Hallegatte et al., 2013; Doocy et al., 2013). Climate change is expected to significantly amplify the probability of extreme flood events over the next decades, especially in Southeast Asia where the number of coastal cities is disproportionately high (Hanson et al., 2011). This trend is especially worrisome since half of the people living in cities with at least 100,000 inhabitants are not farther than 100 km from the coast (Barragán and Andrés, 2015). Some of these cities are also accompanied by uncontrolled urban sprawl (Phung, 2016; Kontgis et al., 2014; Huong and Pathirana, 2013; Storch, 2011), which exacerbates the risk of disaster-induced damages and losses due to the combination of increased exposure and vulnerability (IPCC, 2022). To respond to this problem, local decision-makers require a sound understanding of the complex inter-play of underlying natural processes and oftentimes hidden socio-economic drivers that dictate the feasibility and effectiveness of possible adaptation strategies (Beven, 2011; Thorne et al., 2015). This knowledge can be advanced through the application of hydro-numerical models, which are increasingly becoming the preferred option for inundation mapping (Dasallas et al., 2022). These, in turn, rely on information about prevailing environmental constraints, such as the topography and hydro-meteorological conditions (Quan et al., 2020; Nkwunonwo et al., 2020; Kim et al., 2019; Ozdemir et al., 2013).

With respect to Southeast Asia, many national institutions still refrain from making this crucial input data available for various (technical or political) reasons (Kim et al., 2018; Hamel and Tan, 2021; Liu et al., 2020), which complicates numerical studies, especially for independent parties. Not only is the acquisition of these data sets prohibitively costly, but they also often lack the required spatial and temporal coverage needed for proper derivation of boundary conditions and model set-up. Furthermore, it is often the case that such data are badly described and lack the necessary meta-data. However, relevant information is increasingly published, either in connection with scientific articles or at freely accessible repositories (Di Baldassarre and Uhlenbrook, 2012; René et al., 2014). An increasing number of online media articles, open climate models and code repositories further add to this trend. Accordingly, several studies have recently discussed the possibility and implications of deriving modeling inputs from open-access data sources. This includes local hydro- and meteorological boundary conditions, such as rainfall intensities (Zhao et al., 2021) and sea level rise scenarios (Brown et al., 2016), as well as topographic elevation models (Schellekens et al., 2014; Sanders, 2007). In addition, the expansion of social media applications continuously improves the potential to validate the results of urban flood models (Wang et al. 2018; Feng et al., 2020). Increasing efforts are being made to build models based in part on open-access data in regions where data is scarce (Mehta et al., 2022; Trinh and Molkenthin, 2021; Pandya et al., 2021; Ekeu-wei and Blackburn, 2020), including models capable of mapping urban inundation during or shortly after an extreme event by leveraging data generated from social media (Guan et al., 2023). However, all aforementioned attempts still relied partially on locally sourced, non-open-access data. In fact, to this date, no study is known to utilize a urban surface runoff model, which is exclusively built on freely available data, although this would be a worthwhile target to illustrate the necessity as well as the benefits of comprehensive data accessibility. Even though such

open-access data cannot always be the basis for flood maps that can be considered as truth (especially when validation data is lacking), their potential usefulness should not be overlooked. Especially, when the overarching goal is to improve system understanding (i.e. knowledge about the causalities between drivers and resulting impacts), generating flood estimation maps can open up opportunities to gain insights for subsequent decision-making processes regarding more detailed modelling for

critical areas. Furthermore, no efforts are known to develop a simple flood severity index that combines flood depth and duration, both of which have a significant impact (Rättich et al., 2020). Such an index could deliver a more complete picture of the potential damage of flooding, even in the absence of extensive data necessary for a sophisticated damage model.

Studying the metropolitan area of Ho Chi Minh City (HCMC), Vietnam, a city that epitomizes the complex interplay of disaster risk components in an environment where accessibility to official data or capacities are limited (Kreibich et al., 2022), this

paper explores if and by what means an urban flood model can be developed without acquiring any exclusive (geo)spatial or hydro-meteorological data. With the overarching goal of providing a methodology for researchers to build low-cost, low-effort and fully transparent hydro-numerical models for any part of the globe where either data is scarce or capacities and competence are limited, this manuscript investigates the usability and reliability of hydro-numerical models that are built exclusively on open-access data. The paper focuses on the methodological steps required to derive boundary conditions from cross-

referencing of several freely accessible and reliable sources. These include open-access satellite imagery, governmental and scientific databases as well as data and information from open-access journal articles. Such low-cost, low-effort models are ideal for preliminary food hazard assessment in any flood risk analysis, especially in rapidly developing urban agglomerations where data are scarce and modeling expertise is often limited. Secondly, the paper introduces a new perspective on flood intensity by proposing a normalized index, which integrates simulated flood depth and duration to paint a more complete

picture of flood hazard, while facilitating an estimation of damage potential, especially for cities located in low-elevation coastal zones (LECZ) where flow velocity due to pluvial flooding plays a secondary role. Both approaches are finally validated by contrasting the individual model components and resulting inundation hotspots with conventionally acquired information and data from local partners. It, therefore, justifies the developed concept, accounts for the feasibility of the primary objective and legitimates the call for open-access data and open science (Miedema, 2022) in the field of urban flood modeling on a

worldwide scale. The presented methodology can be seen as an orientation for city planners and authorities from data-scarce regions, helping them to readily estimate where inundation hotspots with particularly high damage potential are located in a first flood hazard assessment. It allows them to focus, subsequently, on building more detailed damage models for the most heavily exposed city districts. Such detailed damage models usually require more extensive and expensive data collection (e.g. detailed topography, detailed time series for certain flood events, drainage networks, flood protection systems, land use, socio-

economic vulnerability, etc.) and are indispensable for quantifying risk as a function of hazard, exposure and vulnerability. The methodology proposed in the following is especially beneficial in those situations where such highly resolved data are (still) missing, inaccessible or require significant resources.

## 2 Materials and Methods

There are generally two essential inputs that a hydro-numerical model needs to produce reliable results. These are elevation data including the hydraulic roughness as well as the model domain based on topographic boundaries (Figure 1 (a)) and, secondly, hydro-meteorological data, such as tidal water levels, river discharge and precipitation data depending on the investigated environment (Figure 1 (b)). The ensuing simulation results can be interpreted using model outputs like flood depth and duration, which can be combined into flood severity (Figure 1 (c)) as will be explained within this work. The acquisition, processing and implementation of the input as well as the processing of the output data require further methodological steps, which will be discussed in the following subsections. Regarding data acquisition, special attention needs to be given to the source, since it dictates the reliability and completeness of the data. Generally, the search priority of terrain data, as well as hydro-meteorological data follows the same path, with official sources at the top, followed by global repositories, peer-reviewed literature, *grey* literature (i.e. publicly available reports and assessments) and finally regional and global models. This workflow will be demonstrated in the following sections using the example of a HEC-RAS 2D model – a capable and freely available program by the U.S. Army Corps of Engineers (USACE) based on the 2D shallow water equations – built for the metropolitan region of HCMC.

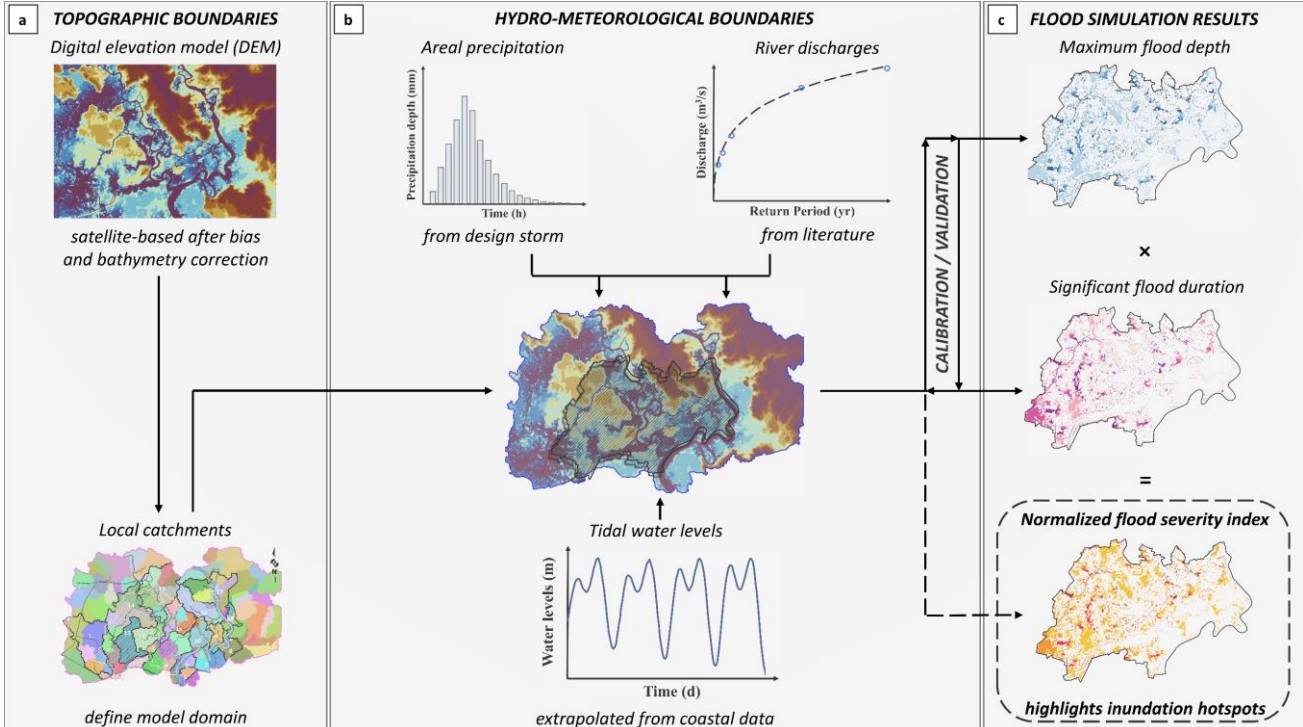

**Figure 1: WORK FLOW: (a) The first panel shows the topographic data from which the local catchments can be determined that define the final model domain; (b) in the second step, hydro-meteorological time-series are defined, which serve as boundary conditions for the numerical model; (c) thirdly, simulation results are presented for the HCMC urban districts (hatched area in b) in terms of maximum flood depth, significant flood duration as well as in the integrated form of a normalized flood severity index ($I_{NFS}$) that is to be defined within this work. Topographic data visualized using scientific color maps created by Crameri (2021). All other maps use colors for illustration purposes only.**

**2.1 Surface Elevation Data**

**2.1.1 Topographic Data**

For most parts of the world, accurate and reliable data on local topography is hard to acquire without significant financial efforts. Data from high-resolution Light Detection and Ranging (LiDAR) is freely available only for the coastal USA, coastal Australia and parts of Europe, but not for the majority of developing countries or emerging economies like Vietnam (Meesuk et al., 2015). This is particularly problematic when setting up urban surface runoff models, which heavily depend on terrain

elevation. For the rest of the world, the only alternative to own (self-conducted) measurements or unvalidated commercial digital elevations models (DEMs) (Planet Observer, 2017; Takaku and Tadono, 2017; Intermap, 2018), both of which are prohibitively costly (Hawker et al., 2018), are open-access satellite-based DEMs. An example of such open-access DEMs is the highly popular Shuttle Radar Topography Mission (SRTM) (Hu et al., 2017; Sampson et al., 2016; Jarihani et al., 2015; Rexer and Hirt, 2014), which was acquired in 2000 and covers around 99.7% of the global populated areas (Bright et al., 2011).

However, these models have substantial vertical errors and relatively coarse resolutions. Accordingly, they cannot reflect micro-topographic features or infrastructure developments in relatively flat terrain (Gallien et al., 2011; Chu and Lindenschmidt, 2017). This is particularly evident for urban settings with a significant positive bias created by the backscatter of buildings and vegetation (Becek, 2014; Shortridge and Messina, 2011; Tighe, M. & Chamberlain, D., 2009; LaLonde et al., 2010), making them unsuitable to resolve terrain features that actually control flood extents and dynamics (Schumann et al.,

2014). In fact, the mean error of SRTM can reach up to 3.7 m when compared to LiDAR (Kulp and Strauss, 2019), significantly distorting simulated flood extents for coastal areas under considerable tidal influences. Furthermore, considerable problems may arise due to differences in geodetic referencing for various digital elevation models, which can lead to false absolute surface elevations (Minderhoud et al., 2019). An attempt to rectify these errors was undertaken by Kulp and Strauss (2018), who developed a novel CoastalDEM by using a neural network to perform a nonlinear, non-parametric regression analysis of

SRTM errors, suggesting better performance and adequacy in urban environments. Another attempt at correcting a satellite-based DEM was done by Hawker et al. (2022), who created FABDEM by removing buildings and forests from COPERNICUS DEM through the use of machine learning. Although CoastalDEM and FABDEM have the ability to provide better elevation accuracy in urban settings, its plausibility still needs to be checked for each individual study area. This can be done through the inspection of terrain elevation at key locations, which can either be structures (canal banks, dikes, flood protection

structures) or locations where flooding is frequently reported (hotspots), all the while taking the elevation data of other satellite-DEMs like ALOS, ASTER, SRTM and COPERNICUS into account. Another issue with the freely available version of CoastalDEM is its resolution of 3 arc seconds, whereas other open access satellite-based DEMs are available in a 1 arc second resolution. A list of available DEM data sets, their resolution and providing agencies is given in Table 1.

**Table 1. A list of freely available DEMs along with their different versions, their issuers and their date of issuance as well as their resolution and vertical accuracy.**

| Freely Available Satellite DEMs | | | | | |
|---|---|---|---|---|---|
| Name | Version | Issuer with Link (Reference) | Publication Date | Horizontal Resolution | Vertical Accuracy |
| SRTM | 1 | NASA (EROS, 2018) | 2004 | 3-arcsecond | 16 m absolute error (Globe) (Farr et al., 2007) |
|  | 2.1 |  | 2005 | 3-arcsecond |  |
|  | 3 |  | 2013 | 1-arcsecond |  |
| ALOS | 1 | JAXA (OpenTopography, 2016) | 2015 | 1-arcsecond | 4.10 m RMSE (Globe) (Tadono et al., 2015) |
|  | 2 |  | 2017 | 1-arcsecond |  |
|  | 3 |  | 2020 | 1-arcsecond |  |
| ASTER | 1 | NASA/METI (ASTER) | 2009 | 1-arcsecond | 9.34 m RMSE (US) (Gesch et al., 2012) |
|  | 2 |  | 2011 | 1-arcsecond | 8.68 m RMSE (US) (Gesch et al., 2012) |
|  | 3 |  | 2016 | 1-arcsecond | 8.52 m RMSE (US) (Gesch et al., 2016) |
| COPERNICUS | 1 | ESA (Copernicus DEM, 2019) | 2019 | 1-arcsecond | <4 m absolute error (Copernicus DEM, 2019) |
| CoastalDEM | 1.1 | Climate Central (Kulp and Strauss, 2018) | 2018 | 3-arcsecond | 4.02 m RMSE (Globe <5 m) (Kulp and Strauss, 2021) |
|  | 2.1 |  | 2022 | 3-arcsecond | 2.63 m RMSE (Globe <5 m) (Kulp and Strauss, 2021) |
| FABDEM | 1.0 | Fathom Global (Hawker et al., 2022) | 2022 | 1-arcsecond | <2.88 m absolute error (Hawker et al., 2022) |
|  | 1.2 |  | 2023 | 1-arcsecond |  |

To utilize an open-access satellite-based DEM in reliable flood simulations, several processing steps are necessary, which, for the case of HCMC, are summarized in Figure 2 below. One solution to circumvent the limitation of vertical errors can be a height correction of SRTM (Figure 2(b)) based on CoastalDEM (Figure 2(a)). To that end, an offset map representing the

difference between SRTM and CoastalDEM is created (Figure 2 (c)) and downscaled using a surface spline interpolation. This offset map is then added to the SRTM, which results in a height-corrected, higher-resolution elevation model (Figure 2 (d)). Depending on the use case, the resulting elevation model can be further processed through the use of a 2D median filter (Figure 2 (e)) to smooth out the surface and reduce noise (Ansari and Buddhiraju, 2018). Furthermore, filling algorithms can be used to counteract artifactual sinks and holes with no physical meaning that typically arise in remote sensing. These sinks and holes

can be closed by a variety of methods. A comprehensive list of filling algorithms can be found in the works of Lindsay (2016).

It is recommended to only use these after incorporating bathymetric data (Section 2.1.2) into the DEM (Figure 2 (f)) to guarantee proper water routing (i.e. from higher-lying to lower-lying cells).

In the case of HCMC, the adequacy of these five elevation models was assessed by considering their terrain elevation at the inner-city canal banks that are well-known inundation hotspots. Only CoastalDEM delivered a plausible average terrain
elevation of 0 m above mean sea level (MSL) at this location, while all others returned average terrain elevations of +6 m and higher. As this level is far above storm surge peak water heights (FIM, 2013), the comparison suggests best accuracy for CoastalDEM. An adequate representation of the canal bank elevations is especially important for flood modeling, since riparian areas are highly exposed to flooding through storm surges and because such events cause significant backwater effects that have a crucial impact on water drainage.

To evaluate the accuracy of the end result, a statistical comparison using the mean absolute error (MAE), the mean error (ME), the root mean square error (RMSE) and the standard deviation (STD) was made between SRTM, CoastalDEMv1 and the generated DEM, on the one hand side, and LiDAR data from 2020 at three locations across HCMC on the other (Table 2). These locations, their extents and corresponding LiDAR characteristics can be found in the Supplementary Material of this article. The generated DEM shows a reduced error when compared to SRTM and CoastalDEMv1 versus the LiDAR data set
across all three areas. Specifically, the positive bias of SRTM is eliminated, all the while halving the negative bias of the CoastalDEMv1 across all presented metrics. Although the ME of the generated DEM was calculated to be -0.45 m, it still offers a substantial improvement not only over SRTM (mean error of 1.22 m) and CoastalDEMv1 (mean error of -0.91 m), but also over all other DEMs presented in Table 1. The same applies for the absolute mean error, the RMSE and the STD of the error. A detailed comparison for all DEMs of Table 1 is provided in the Supplementary Material.

**Table 2. A statistical comparison of SRTM, CoastalDEMv1 and the generated DEM with LiDAR data across three areas in HCMC**

| Statistical Comparison Relative to LiDAR Data | | | | | | | | | | | | | |
|---|---|---|---|---|---|---|---|---|---|---|---|---|---|
| | | MAE (m) | | | ME (m) | | | RMSE (m) | | | STD (m) | | |
| Area (Km²) | | SRTM | Coastal DEMv1 | Generated DEM | SRTM | Coastal DEMv1 | Generated DEM | SRTM | Coastal DEMv1 | Generated DEM | SRTM | Coastal DEMv1 | Generated DEM |
| 1 | 96 | 2.47 | 1.34 | 0.81 | 1.28 | -1.0 | -0.51 | 3.32 | 1.81 | 0.96 | 3.07 | 1.52 | 0.81 |
| 2 | 48 | 2.51 | 1.22 | 0.80 | 1.20 | -0.73 | -0.38 | 3.21 | 1.62 | 0.95 | 3.03 | 1.41 | 0.86 |
| 3 | 21 | 8.44 | 4.58 | 0.62 | 0.98 | -1.1 | -0.39 | 3.56 | 1.74 | 0.75 | 3.33 | 1.43 | 0.64 |
| Total | 165 | 2.5 | 1.3 | 0.77 | 1.22 | -0.91 | -0.45 | 3.33 | 1.71 | 0.93 | 3.12 | 1.45 | 0.81 |

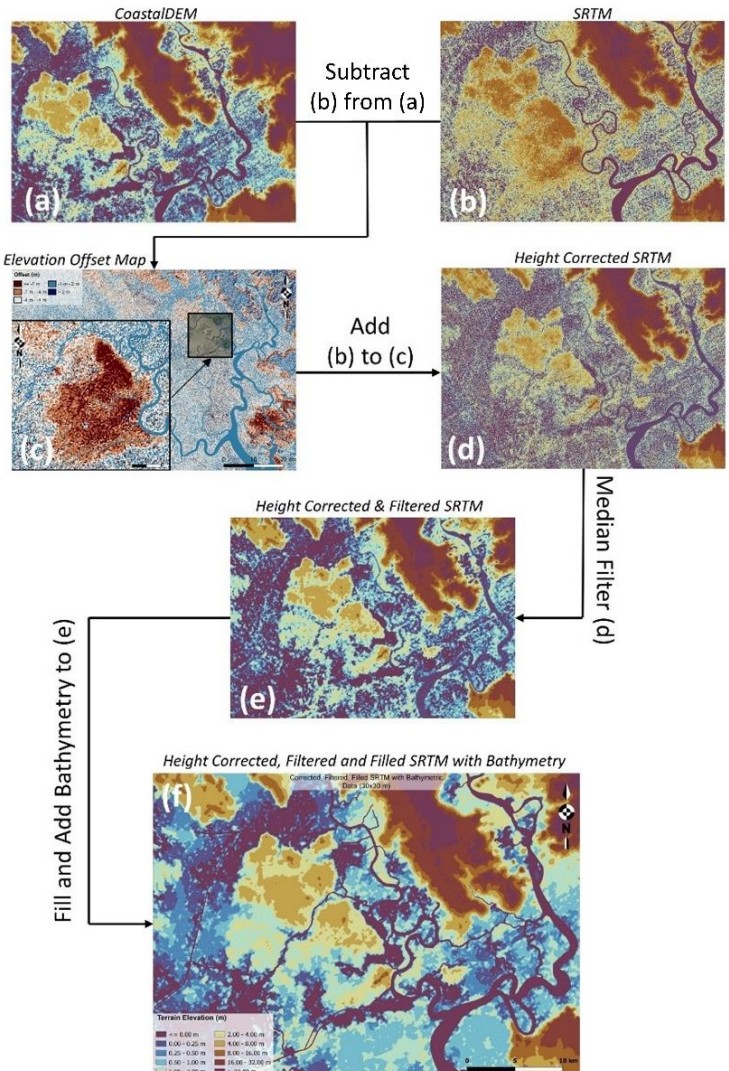

**Figure 2. TERRAIN DATA: (a) represents the CoastalDEM, which is subtracted from the SRTM shown in (b) to produce the elevation offset map presented in (c). (d) is the result of adding (b) to (c). (e) depicts the result of the 3×3 2D median filter, which was then filled and enriched with bathymetric data, leading to the final elevation model shown in (f). Topographic data visualized using scientific color maps created by Crameri (2021).**


### 2.1.2 Bathymetric Data

An intrinsic drawback of satellite-based DEMs is the inability of the synthetic aperture sensors (SAR) to determine the geometry of river beds (Farr et al., 2007). Additionally, the generated pixels include surrounding regions, resulting in greatly overestimated channel depths (Yan et al., 2015b). Therefore, bathymetric data from other sources has to be incorporated into

any satellite-based DEM. The availability of reliable open-access bathymetric data, with a resolution sufficient for use in flood modeling, greatly differs between countries and is generally more difficult to acquire. In fact, the availability of such data is restricted even in many developed countries (Moramarco et al., 2019), oftentimes requiring expensive surveys that are limited

to local scale (Guan et al., 2023). To circumvent this problem, more extensive research for bathymetric data into peer-reviewed articles as well as engineering reports (grey literature) is recommended. Where such literature does not exist, river width and depth can either be approximated (Patro et al., 2009; Neal et al., 2012; Yan et al., 2015a), obtained from calculated global river width and depth databases (Yamazaki et al., 2014; Andreadis et al., 2013), or surveyed in waterways with unknown navigational depths.

In the example of HCMC, the hydrological situation (Figure 3 (a)) is defined by two major streams, namely the Dong Nai River, which passes the urban districts at the eastern city boundary, and the Sai Gon River, which enters the urban area at the central north and flows into the larger Dong Nai at the central south. These waterbodies are fed by a complex network of artificial canals that drain the inner city. Both the natural and man-made waterways have to be incorporated into the DEM. To that end, the bathymetry of the Dong Nai River can be approximated from a research article by Gugliotta et al. (2020), who digitized bathymetric maps originally prepared by the US Army Corps of Engineers (USACE) in 1965 (Figure 3 (c)). No open-access data exists for the Sai Gon River, thus requiring an assumption based on official navigation depths at different shipping terminals along the river. The Sai Gon bed elevation was approximated through interpolation between locations with known navigation depths (10.5 m below MSL at Ben Nghe Port, 8.5 m below MSL at Tan Thuan Port, 6.5 m below MSL at Truong Tho Port) (Ben Nhge Port Company Ltd., 2014; Trameco S.A., 2014; Saigon Port Joint Stock Company, 2019) and extrapolation beyond the most upstream value with a slope of 0.1%. This slope represents the average of the Sai Gon at its midsection (IGES, 2007) and was extended until the northern boundary of the model (Figure 3 (b)).

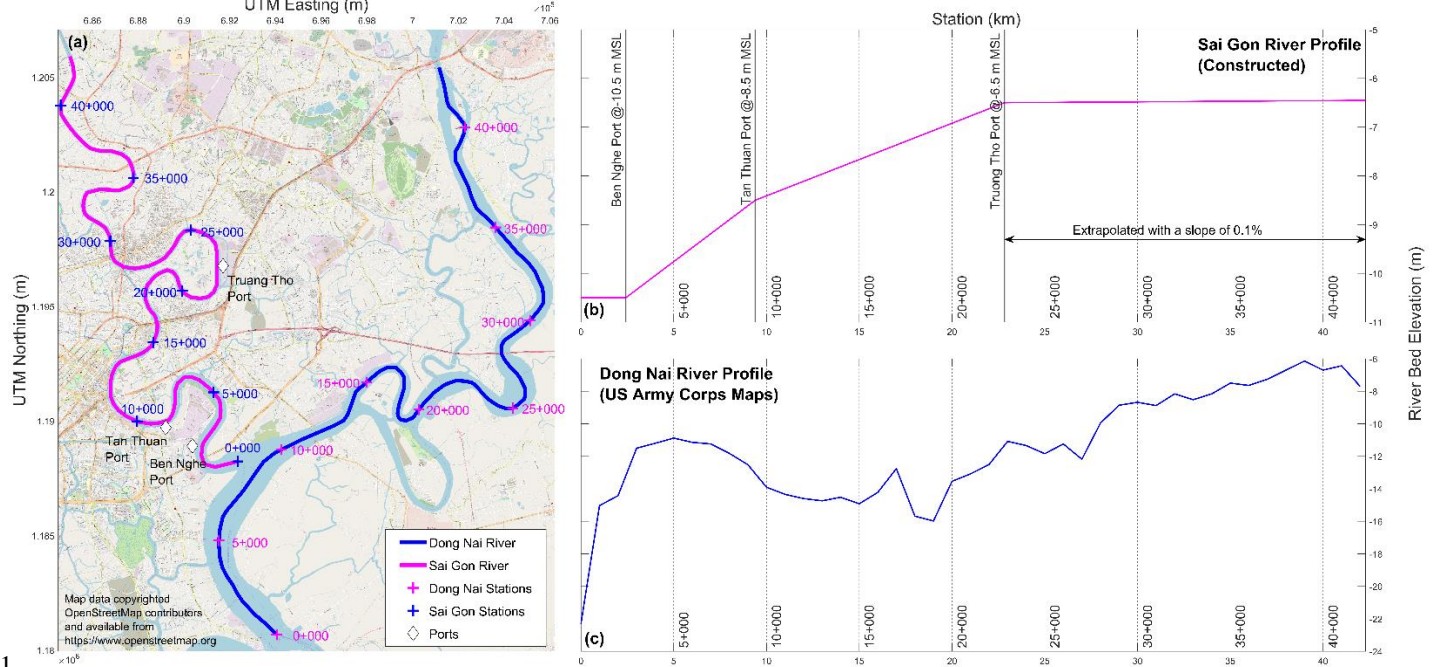

**Figure 3. RIVER BATHYMETRY: (a)** shows the location of the ports used to determine the depth of the Sai Gon River as well as the stationing for the river bed elevation. **(b)** and **(c)** show the constructed Sai Gon River and digitized Dong Nai River longitudinal profiles, respectively.

The results of a sensitivity analysis to quantify the impact of this assumption on the simulation results is presented in Section
3.2. For the inner-city canals, a survey conducted by the Japan International Cooperation Agency (JICA, (2001) determined
the average depth of these canals to range between 1.82 m and 3.82 m below MSL. Given that neither detailed cross-sections
nor profiles were available, all identified canals and channels were set to a depth of 3 m below MSL. For the specific case of
HCMC, the aforementioned processing steps lead to the final elevation model: a height corrected, 2D median filtered and filled
SRTM topography with a 1 arc second resolution that incorporates bathymetric data for all relevant waterbodies (Figure 2 (f)).
Based on this model, various local flow catchments can be defined of which, however, not all contribute to pluvial flooding in
the metropolitan area. Therefore, the perimeter of the flood model is set to include the central 18 key urban catchments which
contribute to flooding inside HCMC (Figure 4). This allows to limit simulations to the area of interest and hence to decrease
computation times without affecting simulated flood depths. Although based on several case-specific simplifications, this
methodology illustrates how free satellite-derived DEMs can readily be combined with public information on river
bathymetries and finally produce a terrain model that can be used for hydro-numerical simulations.

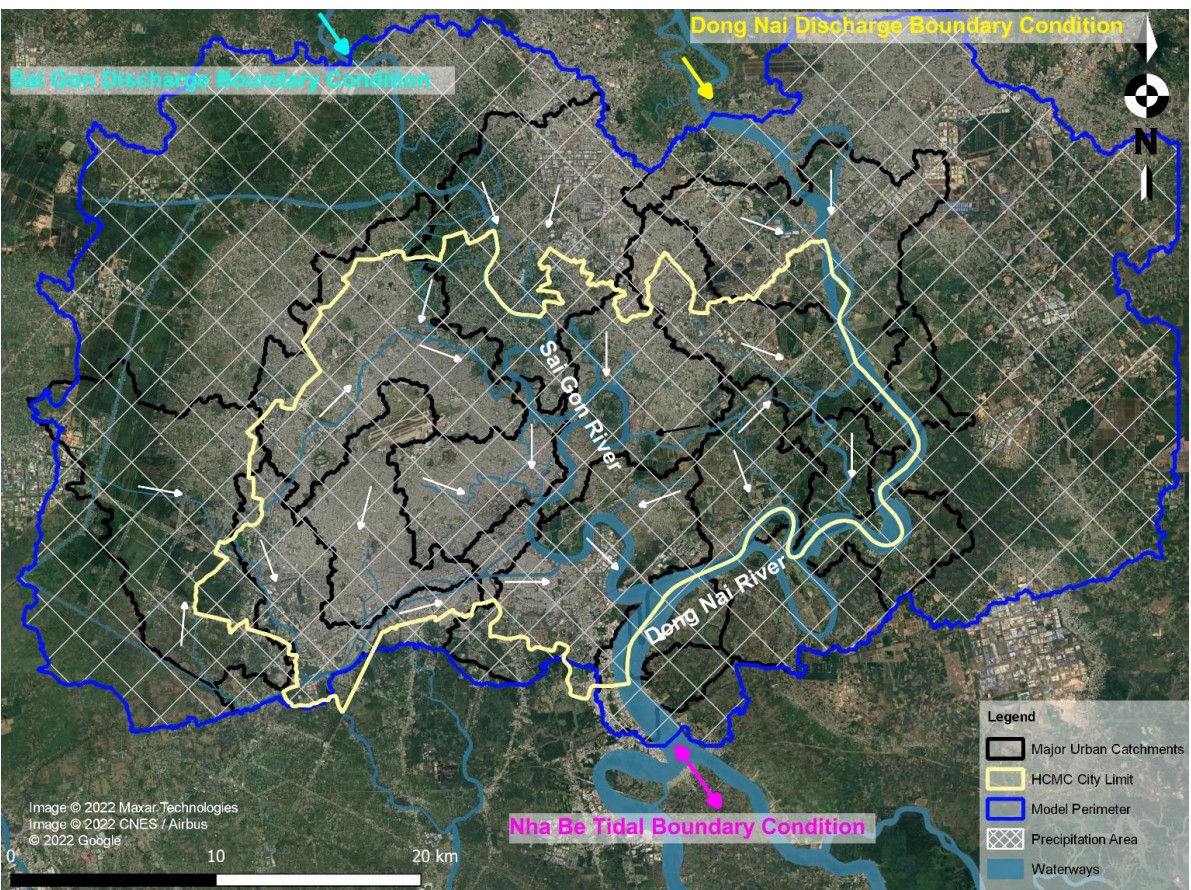

**Figure 4. URBAN CATCHMENTS: The hydrological make-up of HCMC where all of the local catchments that could be determined through the processed DEM are presented. On this basis, 18 key urban catchments were defined, which contribute the greatest part to pluvial flooding within the city. The boundary of the hydro-numerical model equals the perimeter of these catchments in order**
**to decrease computation times without affecting simulated flood depths.**

### 2.1.3 Hydraulic Roughness Coefficient and Model Calibration

Due to the 1 arc second resolution, buildings and extensive vegetation that significantly reduce the available cross-section for water routing are not represented as no-flow areas in the final DEM. Instead, an equivalent Manning friction coefficient was considered in the simulated hydraulic roughness, representing an additional macro-roughness effect that would be neglected
if set to the value of, for example, concrete (Chen et al., 2012; Taubenböck et al., 2009; Vojinovic and Tutulic, 2009). HCMC, for instance, is a densely built urban city, whose surface is mostly composed of asphalt or concrete with very low roughness. To allow for this effect, a roughness coefficient range of 0.05 to 0.105 $s/m^{1/3}$ for urban environments has been proposed (Hejl, 1977), whereby specific values depend on the ratio of built-up to non-built-up areas. In order to determine the optimal Manning friction coefficient for the presented model (uniformly applied across the whole modelling domain), a calibration was
undertaken using inundation depths and locations across HCMC provided by local partners for three severe rain events. The simulated flood depths for the respective boundary conditions (Precipitation depth (P) and high-water level (HWL)) for inundation events on the 01/07/2010, 09/07/2012 and 01/10/2012), respectively, are then compared at the observation points using the RMSE, the Nash-Sutcliffe Efficiency (NSE) and the percentage bias (PBIAS) to assess the quality of the results (Table 3).

**Table 3. Model calibration for different Manning friction coefficients focusing on reported inundations during three rain events (left column) and corresponding RMSE, NSE and PBIAS values.**

| Calibration Events | Manning Friction Coefficient | | | | | | | | |
| --- | --- | --- | --- | --- | --- | --- | --- | --- | --- |
| | n = 0.08 $s/m^{1/3}$ | | | n = 0.10 $s/m^{1/3}$ | | | n = 0.12 $s/m^{1/3}$ | | |
| | RMSE | NSE | PBIAS | RMSE | NSE | PBIAS | RMSE | NSE | PBIAS |
| Event 1 Date: 01/07/2010 P = 79 mm HWL = 1.10 m 23 Observations | 0.02 | -5.25 | 37.5 | 0.01 | 0.50 | 5 | 0.02 | -1.75 | -25.6 |
| Event 2 Date: 09/07/2012 P = 58 mm HWL = 1.12 m 19 Observations | 0.03 | 0.14 | 21.4 | 0.02 | 0.64 | 10.7 | 0.03 | 0.29 | -15.3 |
| Event 3 Date: 01/10/2012 P = 74 mm HWL = 1.15 m 18 Observations | 0.04 | -3.23 | 33.7 | 0.03 | 0.52 | 6.2 | 0.05 | -1.42 | -17.9 |

Following this approach, the best results for the RMSE, NSE and PBIAS are obtained for a Manning friction coefficient of 0.10 $s/m^{1/3}$, which corresponds to the higher bound of the proposed range for mimicking urban settings (Schlurmann et al.,

2010). The achieved NSE values of 0.50 to 0.64 are particularly encouraging when compared to the calibration of the flood model by Le Binh et al. (2019) that achieved values of 0.51 to 0.89 using 2 m resolution LiDAR data. The presented model was validated, subsequently, for a Manning friction coefficient of 0.10 s/m$^{1/3}$ using a fourth, independent rain event. Detailed results of this validation are presented in section 3.1.

## 2.2 Hydro-meteorological Boundary Conditions

As in the case of terrain and bathymetric data, the availability of data pertaining to hydro-meteorological boundary conditions varies widely depending on the region to be modeled. Nevertheless, a similar approach as proposed for the elevation data can be adopted, whereby information and data originating from official sources have the highest priority, followed by open-source repositories, peer-reviewed literature, grey literature and regional models in descending order of importance. Generally, raw time series allow for an independent determination of intensities and return periods of extreme events by fitting the data to a probability function, e.g. Gumbel, Fréchet, or Weibull distributions. A review of this methodological approach can be found in Hansen (2020). However, when there is consensus in the literature, such time series in sufficient temporal resolution, i.e. daily or even monthly cumulative data, are absent or an independent statistical analysis is not necessary, extreme values from the literature can be used. This process can be illustrated through the example of HCMC, ` riverine, tidal and precipitation boundary conditions are needed. Nonetheless, given that the greatest problem for the inhabitants and authorities of HCMC is frequent, economically disrupting flooding due to the combination of heavy rain and high tidal water levels, the focus of this manuscript was put on precipitation, which is why the exemplary probabilistic analysis will only be shown for precipitation data. The methodology, however, can be applied to all other hydro-meteorological boundary data as well.

### 2.2.1 River Discharge Data

Discharge data is typically readily available, especially in the presence of reservoirs along a river. For the Sai Gon and the Dong Nai Rivers, however, no open-access discharge data exists following the FAIR principles in data policy and stewardship (GO FAIR, 2016; Wilkinson et al., 2016; Mons et al., 2017), although both are regulated by upstream reservoirs. Nevertheless, singular extreme discharge rates and their respective return periods can be found in the additional material of a research article by Scussolini et al. (2017). Furthermore, long-term mean river discharges of 54 m$^3$/s for the Sai Gon and 890 m$^3$/s for the Dong Nai, respectively, were reported by Tran Ngoc et al. (2016), with the long-term mean river discharge of the Sai Gon River corresponding well to the net discharge of 30 and 65 m$^3$/s for 2017 and 2018 calculated by Camenen et al. (2021). Extreme values can be used to investigate fluvial flooding, while the average values are of use when investigating the influence of other flood drivers in isolation. Notwithstanding the indisputable temporal variability of river discharge in nature, stationary flow conditions can be assumed for the upstream boundaries of many flood models. Specifically, this holds for all settings, in which other flood drivers with significantly higher rates of change exist, such as in coastal storm surge or rainfall run-off models (Sandbach et al., 2018). For the case of HCMC, it is assumed that both the lowland location of the model domain and officially operated reservoirs upstream of the Sai Gon and Dong Nai Rivers justify this simplification.

### 2.2.2 Tidal Data

Although an official gauge station exists at Nha Be (cf. location in Figure 4), directly at the southern boundary of the HCMC model domain, the corresponding tidal time-series are not publicly available. Nevertheless, data from about 300 tide gauge stations are obtainable from the public repository of the University of Hawaii Sea Level Center including a station in Vung

Tau (Caldwell et al., 2015). This gauge is located around 70 km downstream of Nha Be at the South China Sea and documents the periods of 1986-2002 and 2007-2021 almost consistently. To extrapolate that time series to the southern boundary of the model, a linear increase in the water levels can be assumed: as Gugliotta et al. (2019) report, high and low water levels steadily increase with a scaling factor of 1.05 between Vung Tau and Nha Be. In order to validate this approach, official Nha Be tidal time series were compared to the publicly available Vung Tau tidal time series for the year 2016. In fact, after adjusting for a

temporal phase shift of 1.8 hours and adjusting the water levels by a factor of 1.05, a linear regression returns a coefficient of determination of $R^2 = 0.964$ and a RMSE of 0.157 m with a p-value of $p < 0.001$. Extrapolated and observed tidal time series of Nha Be are juxtaposed in Figure 5.

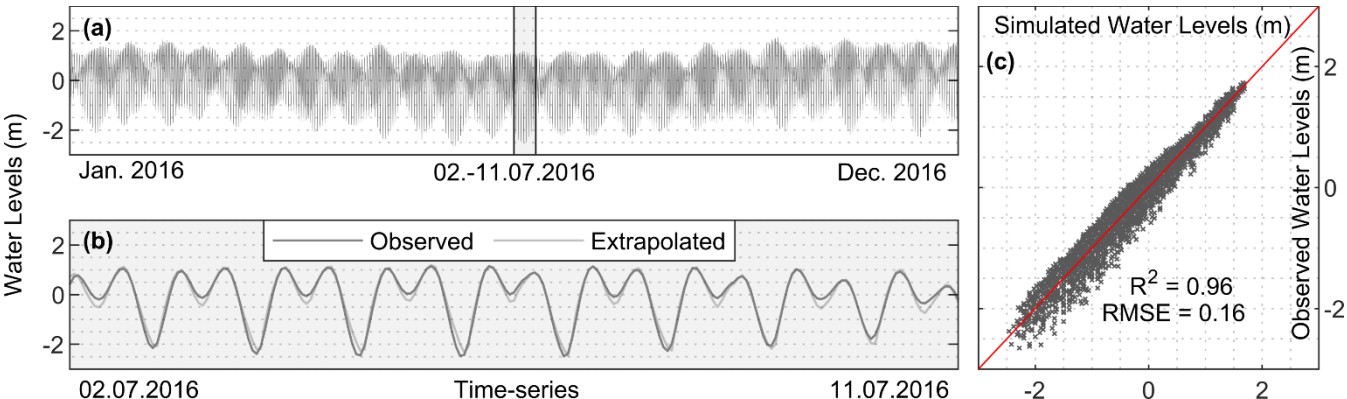

**Figure 5. WATER-LEVEL COMPARISON (a) Time series of the Nha Be tidal gauge for 2016 versus data from Vung Tau after**
**adjusting by a scaling factor of 1.05 and removing the temporal phase shift of 1.8 hrs. (b) Exemplary section of the same time series illustrating the fit of tidal high-water levels. (c) Linear regression for all hourly data points and corresponding quality estimates $R^2$ and RMSE.**

Especially the depicted quality estimates corroborate the findings of Gugliotta et al. (2019) in regards to the water level relation between Vung Tau and Nha Be all the while validating the proposed approach for water level extrapolation. A drawback of

this approach is the inability to calculate the temporal phase shift in water stages and discharges between Vung Tau and Nha Be. The reconstructed tidal data can be analyzed probabilistically for the determination of extreme tidal water levels if needed. In the present study case, an eight-day time series representing mean tidal conditions is used as the southern boundary of the hydro-numerical model. The eight-day timeframe was chosen following two purposes: first, to ensure a so-called spin-up time needed for the numerical stabilization of water levels, and second, to allow for physically realistic routing and concentration

of rainfall runoff within the model domain.

### 2.2.3 Precipitation Data

In the example of HCMC, precipitation depths with return periods of 5 years and less vary greatly in existing literature (Khiem et al., 2017; Quân et al., 2017; Loc et al., 2015; FIM, 2013; Viet, 2008; Nhat et al., 2006). In particular, the values for a storm of 3-hour duration and 2-year return period range from 28 mm/hour to 45 mm/hour, requiring an independent statistical analysis. Daily precipitation time series for the Tan Son Hoa weather station in central HCMC spanning from 1960 to 2012 can be obtained from the repository of the National Oceanic and Atmospheric Administration (NOAA), which publishes quality-checked precipitation data for several weather stations across the globe (NOAA, 2022). To determine the daily extreme precipitation depth for return periods of 2 years and greater, the data are fitted to a Gumbel distribution where the mean $\bar{y}_n$ and standard deviation $\sigma_n$ of the Gumbel variate are taken as a function of the record length, which is equal to the number of years ($n = 28$):

$$P_{T,24h} = \bar{P} + \left[ \frac{-\log\left(\log(T/(T-1))\right) - \bar{y}_n}{\sigma_n} \right] \sigma \qquad (1)$$

where $\bar{y}_n$ is 0.5343 and $\sigma_n$ is 1.1047 for $n = 28$ (Selaman et al., 2007). Using the Cramér-von Mises criterion, a $n\omega^2$ of 0.2831 is calculated, which satisfies testing for $\alpha = 0.1$ (Dyck, 1980). In contrast, the probability of occurrence for return periods of 2 years and less can be calculated by ranking the precipitation depth of the raw data using $(2i - 1)/2m$ where $i$ is the rank of the data point and $m$ is the total number of data points. Given the 24 hours temporal resolution of the raw data, a scaling function is applied to determine the intensities for lower durations (Menabde et al., 1999):

$$i_{T,d} = \frac{P_{T,D}}{D} \left( \frac{d}{D} \right)^{-\beta} \qquad (2)$$

where $i_{T,d}$ is the intensity for duration $d$ and return period $T$, $P_{T,D}$ is the precipitation depth to be scaled and $\beta$ is the scaling factor. Based on the literature average for HCMC, $\beta$ is assumed to equal 0.854 (Khiem et al., 2017; Nhat et al., 2006). The ensuing Intensity-Duration-Frequency (IDF) curves, which reflect the precipitation depth as a function of storm return period and duration, are presented in Figure 6.

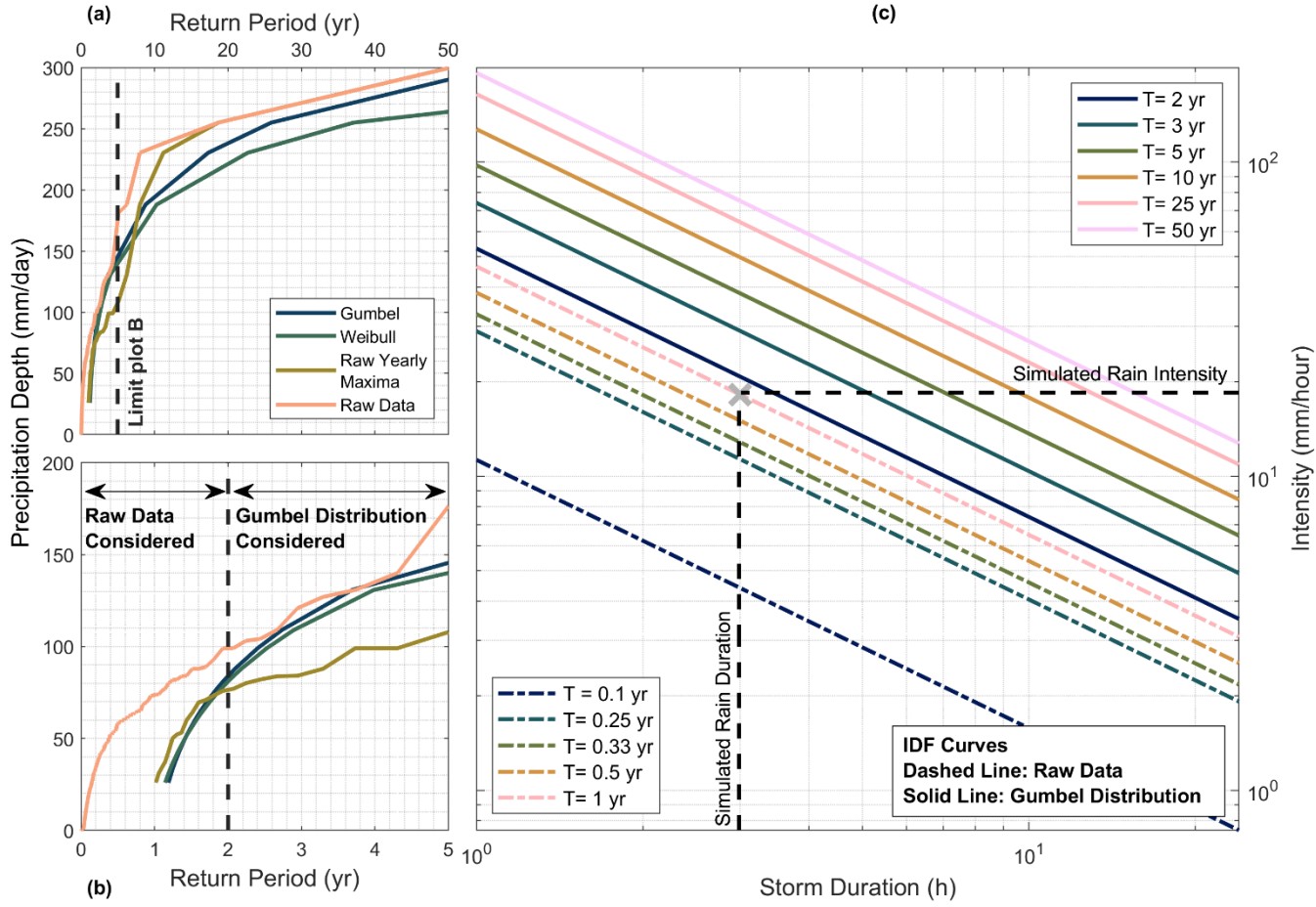

**Figure 6. INTENSITY-DURATION FREQUENCY: (a)** depicts the return period of heavy rain events plotted against the precipitation depths for the raw data, the raw yearly maxima, the Weibull distribution and the Gumbel distribution. **(b)** zooms in from **(A)** for the return period of 5 years and less, showing for which return periods the probability of occurrence and the Gumbel distribution are taken into consideration. **(c)** is the end result, showing the different IDF curves for return periods of 0.1 to 5 years. Data visualized using scientific color maps created by Crameri **(2021)** .

Using official hourly precipitation data for the Tan Son Hoa weather station over the same period, the performance of the NOAA time series as well as the adequacy of the temporal scaling factor $\beta$ was evaluated (Table 4). The mean value of the daily yearly maximum precipitation is 94.7 mm and 104.3 mm, while the standard deviation is 69.13 mm and 40.64 mm for the NOAA and the official hourly precipitation data, respectively. The similarities and differences between the statistical results of both time series will be further discussed in Section 4.

Table 3. A statistical comparison of the NOAA and official hourly precipitation data along with a measure of the goodness of fit using the average temporal scaling factor from literature as well as the temporal scaling factor fit to the official data.

| Return Period (Years) | Calculated Daily Cumulative Rain (mm) | | Best $\beta$ Value Fit to Official | Goodness of Fit using $\beta = 0.854$ (A) and Best $\beta$ Fit Value (B) | | | | | | | |
|---|---|---|---|---|---|---|---|---|---|---|---|
| | | | | SSE | | R² | | Adjusted R² | | RMSE | |
| | NOAA | Official | | A | B | A | B | A | B | A | B |
| 1 | 73.9 | 90.5 | 0.883 | 373 | 210 | 0.838 | 0.912 | 0.865 | 0.924 | 7.89 | 5.92 |
| 2 | 84.3 | 97.6 | 0.871 | 219 | 130 | 0.913 | 0.948 | 0.927 | 0.957 | 6.04 | 4.66 |
| 3 | 117.8 | 114.6 | 0.863 | 18 | 25 | 0.994 | 0.992 | 0.995 | 0.994 | 1.75 | 2.03 |
| 5 | 155.2 | 133.5 | 0.856 | 280 | 303 | 0.930 | 0.925 | 0.942 | 0.937 | 6.84 | 7.10 |
| 10 | 202.2 | 157.3 | 0.850 | 1324 | 1199 | 0.748 | 0.772 | 0.790 | 0.810 | 14.85 | 14.13 |
| 25 | 261.5 | 187.3 | 0.844 | 3779 | 3107 | 0.464 | 0.559 | 0.553 | 0.633 | 25.13 | 22.76 |
| 50 | 305.5 | 209.6 | 0.841 | 6421 | 5104 | 0.250 | 0.404 | 0.375 | 0.503 | 32.71 | 29.17 |

As for the creation of an adequate hyetograph, i.e. the development and representation of precipitation depth over time, numerous algorithms for the creation of a design storm are available (Balbastre-Soldevila et al., (2019). For rain events in HCMC, the linear/exponential synthetic storm of Watt et al. (1986) has been taken to create the hyetograph of a 3-hour duration, 1-year return period rain event, since it matches the hyetograph according to decision 752/QD TTg by the HCMC government. The simple example of deducing the river discharge, tidal water levels and precipitation hyetograph for HCMC
illustrates, how open data, even if not in the form of time series, can be utilized to define reasonable boundary conditions for an urban flood model.

## 2.3 Processing of Flood Simulation Results

### 2.3.1 Use of Difference Plots

Ultimately, the presented methodology allows for setting up a hydro-numerical flood model that simulates surface run-off in
a setting where urban features cannot be fully represented, e.g. exclusion of small-scale topographic elements like flood protection structures (artificial bank elevation, flood protection walls, etc.) or underground systems like technical details of a local stormwater drainage system. Given the regional scale of many models, however, it is assumed that the absence of the latter is compensated by the hydraulic efficiency of a smoothed and filled DEM, which guarantees that water always flows towards the lowest elevations driven by gravity, effectively mirroring the functions of a stormwater drainage system.
Furthermore, there is significant evidence for the ineffectiveness of the stormwater drainage system in the particular case of

HCMC (Le Dung et al., 2021; Nguyen, 2016). The local drainage system is not well maintained and has limited functionality (Nguyen et al., 2019). Drainage capacity is therefore strongly hampered in case of storm events, which justifies its exclusion from the model representing a conservative approach.

In contrast, the absence of flood protection structures in the model has a significant impact on the run-off dynamics, whereby
flooding can even occur in places where no inundation is plausible under normal conditions, i.e. no rain, mean tide and mean river flow. To counteract this effect, simulated water levels are corrected by taking the results of the regular conditions as a reference. This reference was defined based on flooding threshold values determined with local partners, information from grey literature like the JICA reports (JICA, 2001) as well as different media articles, whose URLs can be found in the Supplementary Material. Accordingly, only the additional flooding (above regular inundations) is considered as the actual
level of flooding when simulating events with more intense conditions. In order to isolate the impacts of additional flooding, the results of the simulation under normal conditions are then subtracted from the results of simulations under more intense conditions either occurring in combination or in isolation.

In the HCMC example, the 1-year return period, 3-hour duration (3h1y) rain event is taken for a detailed investigation. The reason for this choice is that these yearly recurring events are not usually put into focus when conducting flood simulations,
although they bring about major economic losses that are comparable to and sometimes even greater than those from extreme flood events (ADB, 2010). In turn, the results of the simulation under long-term average tidal and riverine conditions are subtracted from the results of the simulation for a 3h1y rain event with mean tide and mean river discharge. These difference plots finally reflect the extents and dynamics of typical inundations induced by the isolated 3h1y rain event. This methodological approach can be easily applied to a variety of scenarios and corresponding simulations.

**2.3.2 Flood Intensity Proxies**

In urban flood modelling, the intensity of flooding in a predefined area is typically expressed in terms of maximum simulated flood depths. Although this value is a good indicator for the exposure and scale of affected people during extreme events, it fails to provide an accurate estimate of projected damages or losses. This is especially important when taking into consideration that, particularly in coastal cities, certain flood depths can persist for a much longer time than others due to tidally induced
backwater effects (Andimuthu et al., 2019). This flood duration, on the other hand, is very important when events of marginal intensity, i.e. high probability of occurrence, are investigated, since it can be an indicator for the persistence of economic and social disruption (Debusscher et al., 2020; Ismail et al., 2020; Feng et al., 2017; Wagenaar et al., 2017; Shrestha et al., 2016; Wagenaar et al., 2016; Koks et al., 2015; Molinari et al., 2014; Thieken et al., 2005) in residential and industrial areas (Tang et al., 1992), as well as in an agricultural context (O'Hara et al., 2019). This effect can best be expressed through the creation
of a 'duration over threshold' map, which depicts how long a certain flood depth is exceeded. This threshold value can be adjusted according to the local constraints. In the case of HCMC, the threshold depth was set to 0.10 m, given that this value corresponds to the minimum reported flood depth provided by local partners.

In an attempt to combine the perspectives of flood intensity and duration, a simple 2-parametric but more integrative proxy, namely the 'Normalized Flood Severity Index ($I_{NFS}$)', is defined and tested in this study. This proxy helps to identify areas where the combination of both time-independent maximum flood depth and the duration over threshold is at its maximum and where the largest flood impacts and, accordingly, the most severe damage potential can be expected. This is particularly useful when considering the high economic damage caused by less severe but more frequent urban floods that HCMC regularly suffers from (ADB, 2010). In order to increase the robustness of the dimensionless $I_{NFS}$ against numerical divergence and artifacts, the normalization is based on the 95th (spatial) percentile of flood depth and duration. Depending on the specific case, however, this reference for normalization may be adjusted. The $I_{NFS}$ at each grid cell (x,y) can be expressed as follows:

$$I_{NFS}(x,y)(\%) = \frac{d_{max}(x,y) * T_{d>10cm}(x,y)}{d_{max,95\%}(x,y) * T_{d>10cm,95\%}(x,y)} * 100 \qquad (3)$$

where $d_{max}(x,y)$ refers to the maximum (temporal) simulated flood depth at the local cell with coordinates x and y and $T_{d>10cm}(x,y)$ refers to the scenario-based flood duration over the pre-defined threshold of 0.10 m.

Due to its normalization, the application of the $I_{NFS}$ is not restricted to singular analyses, but can also be considered as an indicator to express changes in flood severity due to changing boundary conditions. For example, when taking climate change scenarios into account, the $I_{NFS}$ can be computed for a particular case and then normalized according to the base case without climate change effects.

**3 Model Performance**

Even in cases where topographic and hydro-meteorological data is sparse or hard to obtain, it should always be possible to gather the most essential boundary conditions and compose a basic hydro-numerical model following the aforementioned methodology. To showcase the applicability and performance of this approach, the following section provides information regarding the validation results for the exemplary surface runoff model of HCMC as well as a sensitivity analysis that scrutinizes the validity of the described assumptions concerning the local bathymetry. Subsequently, the simulation results are analyzed using the indicators and parameters defined in Section 2.3 to determine local flooding hotspots. Data on inundation depths and locations provided by local partners are used in a subsequent step to cross-check the performance of the latter and newly proposed flood intensity proxy, the $I_{NFS}$.

**3.1 Model Validation**

Using a Manning friction coefficient of 0.10 s/m$^{1/3}$, the validation of the model was accomplished by simulating a torrential rain event that occurred during the monsoon season on 14/06/2010. During this event, a total of 73 mm of rain fell on HCMC, while tidal water levels reached maximum heights of 1.15 m. Scattered across the city, flooding was reported for 25 observation

points at street level. The maximum flood inundation depths were determined using the difference plot method described in section 2.3. The simulated and reported flood depths at these observation points are listed in Table 2 of the Supplemental Material. The performance of the validation run was quantified using the NSE, RMSE and PBIAS metrics, which were
calculated to be 0.7, 0.03 m and 4%, respectively. Additionally, Figure 7 shows that the simulated flood depths matched the observations at 62% of all points, while diverging by 5 cm and 10 cm at 33% and 5% of the observation points, respectively. The exact coordinates and locations of the observation points along with the accompanying street names are also included in section 3 of the Supplementary Material. The high resemblance of simulation results and observations underlines the validity of the employed methodology.

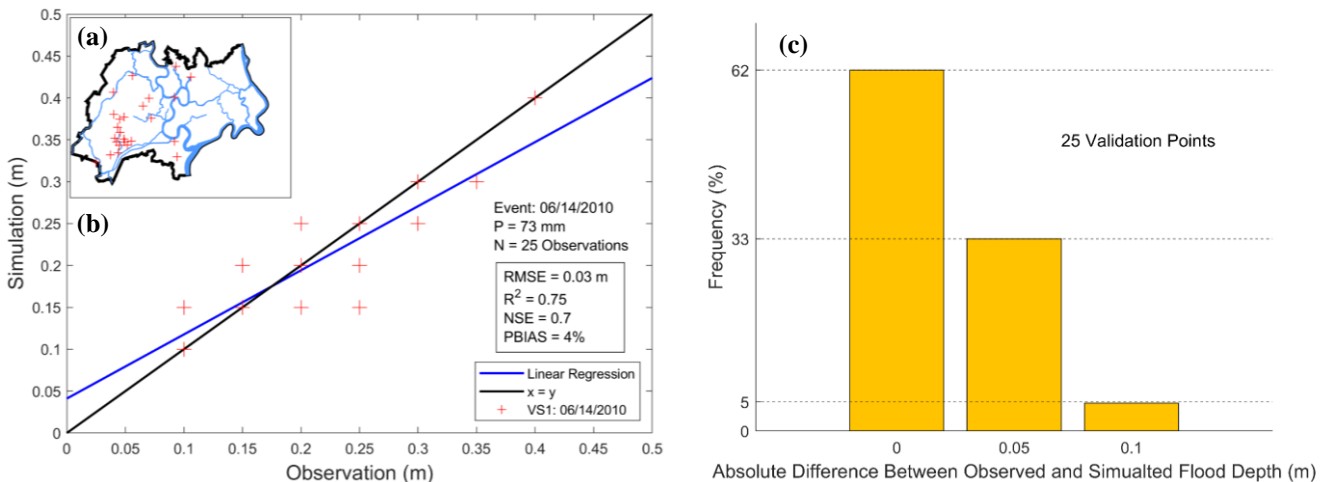


**Figure 7. MODEL VALIDATION: (a) Location of the 25 reported inundations (red crosses) that were used for validation, (b) simulated flood depths plotted against the reported flood depths along with the linear regression (in blue) and the calculated R2, NSE, RMSE and PBIAS (bottom right). (c) Frequency of absolute vertical differences between the observed and simulated flood depths at the 25 observation points across HCMC.**

**3.2 Sensitivity Analysis for the Assumed River Bed Elevation**

Given that the Sai Gon bathymetry is approximated by assumptions that are solely based on the officially maintained fairway depth, it seems mandatory to assess the sensitivity of simulation results to variations of water depth in the Sai Gon River. The river bed elevation is thus varied between the 1.0 and 1.8-fold of the navigation depth in increments of 0.2. The results of this simulation are shown by longitudinal sections in Figure 8. Specifically, the simulated water surface levels increase at points A
(inner-city low point that is a known flooding hotspot), B (canal intersection where frequent flooding occurs) and C (outlet of the Ben Nghe canal) with increasing river bed elevation. Nevertheless, the maximum nominal difference in the water surface levels is 7 cm at point A and 12 cm at both B and C. Comparing depths of 1.2 times and 1.8 times the fairway depth, this difference is 4 cm at point A, which can be considered negligible. Given the low sensitivity of the water surface level to the depth of the Sai Gon, employing the assumption stipulated in section 2.1.1 is rendered sufficient for the flood model.

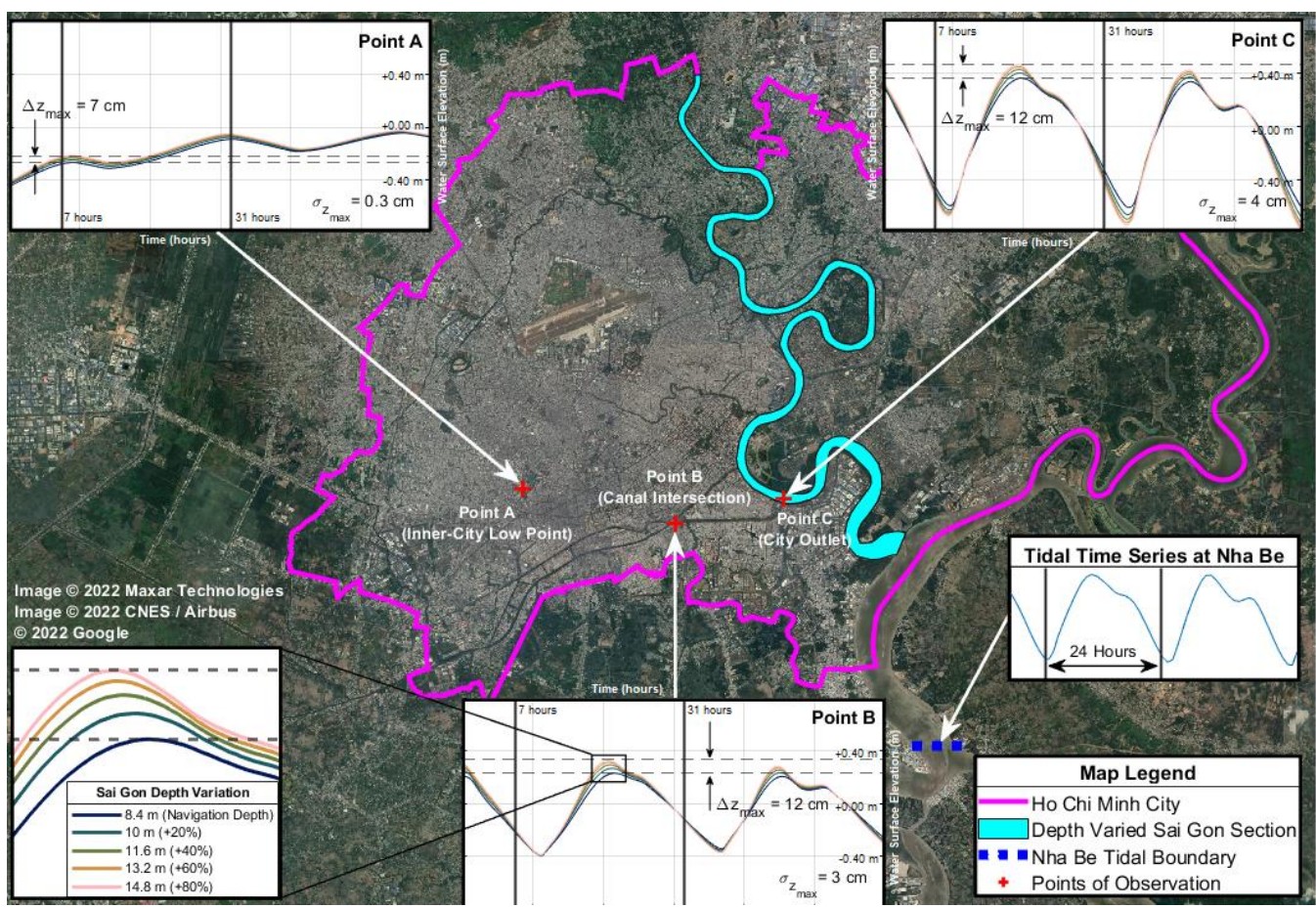


**Figure 8. DEPTH SENSITIVITY: Impact of varying the depth of the Sai Gon River on simulated water depths at three different locations (Point A: inner-city low point, Point B: canal intersection and Point C: city outlet). The zoom box in the lower left corner highlights the maximum difference of 12 cm at a +80% increase of the river depth. Data visualized using scientific color maps created by Crameri (2021).**

**3.3 Performance of the Flood Intensity Proxies**

The 3-hour duration, 1-year return period rain event with a precipitation depth of 54 mm can be investigated using the flood intensity proxies defined in Section 2.3.2. The choice of this particular precipitation event is explained in Section 2.3.1. Comparing Figures 9(a) and 9(b) illustrates the similarities and differences between maximum flood depth ($d_{max}$) and duration over threshold ($T_{d>10cm}$). As can be seen, a high $d_{max}$ does not necessarily translate to a high $T_{d>10cm}$ and vice versa as evident

by the areas on the western bank of the Sai Gon River. At this location, a relatively high $d_{max}$ but a relatively short $T_{d>10cm}$ can be observed. This example epitomizes the usual shortcomings of using only one of the classical proxies for assessing flood damage potential. By combining these, however, inundation hotspots with significant damage potential can be discovered in the distribution of dimensionless $I_{NFS}$ values (Figure 9 (c)). In particular, the locations of reported inundations where sustained flooding demonstrably occurred, and the $I_{NFS}$ heat map show considerable spatial overlapping. While the $I_{NFS}$ only covers 19%

of the total area of HCMC, around 73% of the reported inundations lie inside or within 100 meters of the highlighted areas. These figures are opposed to 78% and 73% for the $d_{max}$ and $T_{d>10cm}$, that cover 38% and 34% of the area, respectively (Table 4). The small spatial extent of the $I_{NFS}$ heat map, relative to the $d_{max}$ and $T_{d>10cm}$ maps, coupled with the relatively high coverage of reported flooding locations corroborates the usefulness of the proposed index in successfully localizing flooding hotspots and quantifying their spatial extents.

**Table 4. Performance of the different flood proxies in terms of the spatial overlapping with the locations of reported inundations**

| Flood Proxy | Spatial Overlap with Reported Inundations (%) | Area Coverage (%) | Accuracy Ratio vs. a Random Area with Equal Coverage (-) |
|---|---|---|---|
| Maximum Flood Depth $d_{max}$ | 78% | 38% | 2.05 |
| Duration over Threshold $T_{d>10cm}$ | 73% | 34% | 2.15 |
| Normalized Flood Severity Index $I_{NFS}$ | 73% | 19% | 3.84 |

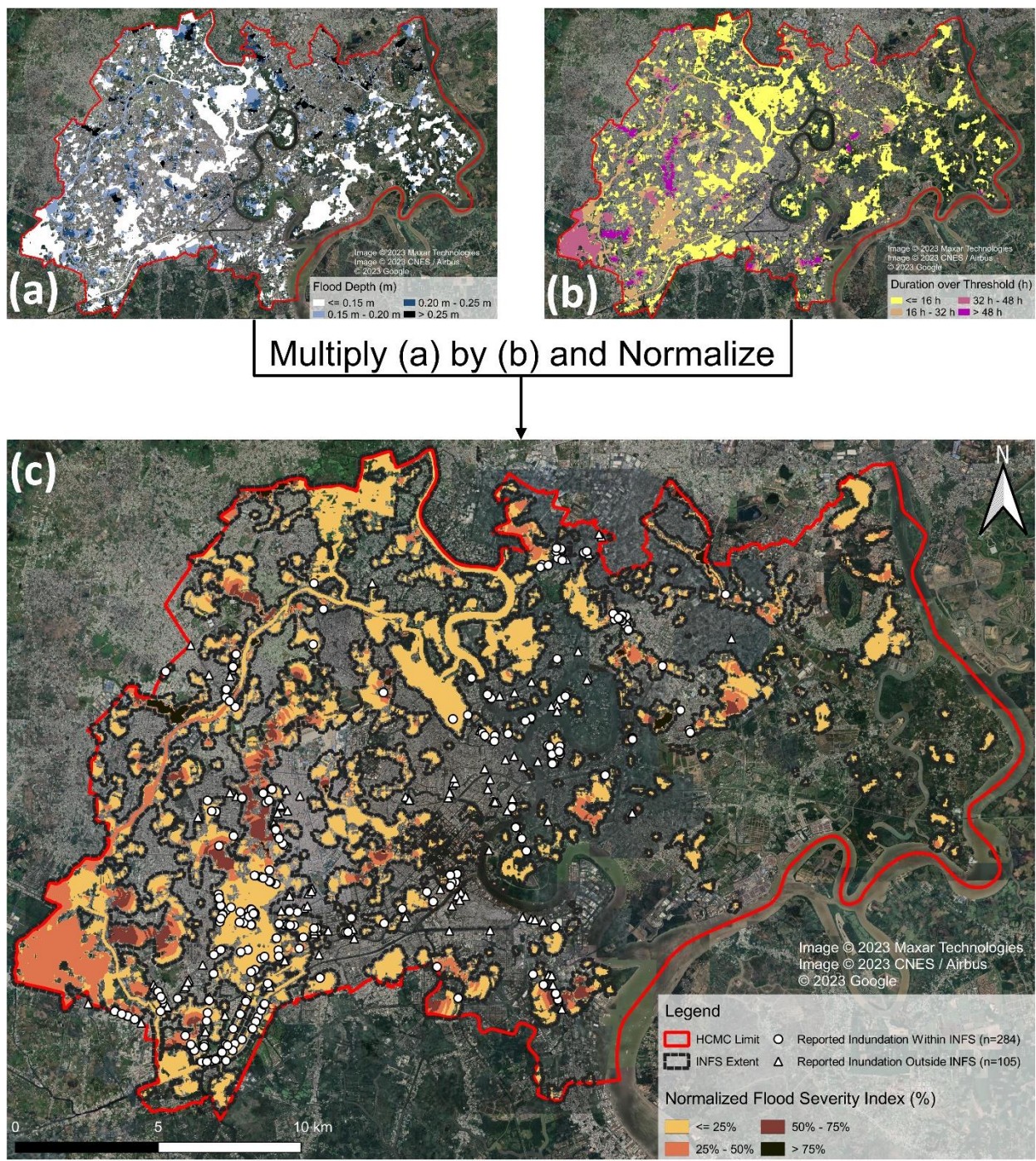

*Normalized Flood Severity Index (INFS)*

**Figure 9. FLOOD INTENSITY: (a) depicts the time-independent maximum flood depth in meters, while (b) depicts the duration over a threshold of 10 cm in hours. (c) reveals the results of I$_{NFS}$, whereby hotspots of this index (covering 19% of HCMC) show a high spatial overlapping with the reported inundations (73% inside or within 100 m). All data was visualized using scientific color maps created by Crameri (2021).**


**4 Discussion**

Since topographic data plays a significant role in flood modeling, its validation is imperative. However, difficulties in this respect arise from the lack of ground truthing data in many countries if local topographic surveys or LiDAR data are inaccessible. The only data close to ground truth in the case of HCMC is the JICA report from 2001 (JICA, 2001), in which various canal bank elevations can be found. Furthermore, there is a substantial difference between high-resolution LiDAR data and satellite-derived DEMs that cannot be closed independent of the amount of processing. As for the satellite DEMs, there exists a multitude of such models that need to be carefully considered for each specific task. Some more recently provided terrain and elevation models, like the COPERNICUS DEM, do offer advantages in terms of lower noise levels and resolution, but do not represent the actual surface elevation in an urban environment, which is especially problematic in urban coastal agglomerations where flawed terrain heights can have a significant impact on flooding extents due to tidal effects. Even the assumption that CoastalDEM or FABDEM represent the actual surface elevation is vague in the context of Southeast Asian coastal cities. In fact, Vernimmen et al. (2020) calculated an average error for the Mekong Delta area in Vietnam of +1.23 m for the SRTM and -1.35 m for the CoastalDEM, concluding that the SRTM generally overestimates surface elevation, while CoastalDEM underestimates it. Building on that approach, a comparison of the performance of the various DEMs in terms of representing the canal bank elevations reported by JICA (2001) can be undertaken for the Tau Hu - Ben Nghe Canal (cf. Figure 3), with the results shown in Table 5.

**Table 5. Absolute bank elevations for the Tau Hu - Ben Nghe canal according to ground truth data by JICA (2001) and from seven freely available satellite-based DEMs. The statistical ranges suggest an overestimation for SRTM, ALOS, ASTER, COPERNICUS DEM, CoastalDEMv2.1 and FABDEM as well as an underestimation for the CoastalDEMv1.1, respectively.**

| Tau Hu - Ben Nghe Canal Bank Elevations | | | | | | | |
|---|---|---|---|---|---|---|---|
| | JICA | SRTM | ALOS | ASTER | COPERNICUS DEM | CoastalDEMv1.1 | CoastalDEMv2.1 | FABDEM |
| Minimum (m MSL) | +0.9 | +7.4 | +6.6 | +7.6 | +4.3 | -2.1 | +2.3 | +2.6 |
| Average (m MSL) | +1.8 | +5.9 | +9.1 | +11.3 | +12.1 | -0.3 | +3.2 | +3.9 |
| Maximum (m MSL) | +2.9 | +13.2 | +40.1 | +17.7 | +16.2 | +1.0 | +8.2 | +5.3 |

The findings in Table 5 are similar to those of Vernimmen et al. (2020), whereby the Tau Hu - Ben Nghe canal bank elevation is overestimated by +4.1 m on average in the SRTM while being underestimated by -2.1 m in the first version of CoastalDEM, thus corroborating the conclusions reached by Schumann and Bates (2018) on the inadequacy of most open access DEMs for flood simulations, especially in urban environments. The newer version of the CoastalDEM (CoastalDEMv2.1), with supposedly improved accuracy, overestimates the canal bank elevations and shows a great divergence from CoastalDEMv1.1, which highlights the difficulty of accurately representing topography in densely built environments even with the help of artificial intelligence.

The reliability of these findings was further reinforced by the comparison of SRTM, CoastalDEM and the generated DEM with three LiDAR areas presented in Section 2.1.1., which showed that SRTM overestimates the terrain by up to 1 m, while CoastalDEMv1.1 and the generated DEM tend to underestimate the terrain elevation by 1 m and 0.5 m, respectively. This

clearly shows that the proposed processing steps to leverage SRTM and CoastalDEM lead to a DEM with a smaller bias than the two original data sets. Furthermore, it is important to measure the amplitude of this bias with regards to other open-access DEMs (SRTM, ALOS, ASTER, COPERNICUS). The positive bias of these traditional DEMs can reach up to +7.5 m against the LiDAR data, rendering them completely unreliable for urban flood modelling purposes. This corroborates the conclusions made by Hawker et al. (2018) in regards to the limited usability of existing DEMs on the global scale. In this regard, the corrected DEM is far more reliable than any other open-access DEM and can confidently be used, especially in the outlined context of preliminary flood estimations.

Additionally, the topography of HCMC is affected by varying degrees of land subsidence, ranging from 0.3 to 5.3 cm/yr (Duffy et al., 2020). In some areas, peak values even reach 8.0 cm/yr (Ho Tong Minh et al., Preprint), which further exacerbates the uncertainty in elevation. Nevertheless, in the presented workflow, the underestimation of the CoastalDEM is successfully counteracted with the use of difference plots (cf. details in Section 2.3.1), through which only additional water levels (in excess of the normal conditions) are considered as actual flooding. Backed up by the model calibration and validation, the joint use of the final (corrected) DEM and the difference plots delivers flood simulations that successfully reproduce known inundation hotspots in HCMC.

In terms of the roughness coefficient, the optimal value determined through model calibration matches the value of a more recent study by Beretta et al. (2018), who concluded that using a value of 0.10 in the absence of buildings had similar flood results as incorporating those elements. This reinforces the idea that replacing buildings with a higher (macro-)roughness coefficient could account for the obstruction effect seen during urban floods when only coarse elevation data is available. However, another method that was implemented by Taubenböck et al. (2009) and Schlurmann et al. (2010) lies in the usage of a building mask within the DEM as a replacement to mimic infrastructure footprints, thereby limiting flood flow dynamics to residual open spaces. Although this method may prove useful in case the resolution of the DEM is 10 m or higher, it might not be easily implemented at DEM resolutions of 30 m or coarser. In the present case, the elevated roughness coefficient offers an adequate solution to this problem that does not substantially alter the maximum flood depths and durations, especially when considering that buildings themselves are not impermeable, yet basements can get flooded during rain events (Sandink, 2016). Looking at the tidal data, the case of HCMC reveals a particular shortcoming of the proposed methodology, namely the temporal phase shift between the tidal time series at Vung Tau and Nha Be cannot be determined from one data set alone. However, it can be assumed that this relatively small phase shift (1.8 hours in this case) has a negligible impact when investigating flooding or backwater effects during storm events given that the phase shift between the start of a rain event and tidal high water can be of much greater importance. Accordingly, sensitivity analyses have to show the worst-case scenario for each particular setting anyway.

Comparing the open-access daily precipitation time series with the official hourly precipitation time series at the Tan Son Hoa weather station shows a certain discrepancy between the two data sets, which becomes evident when comparing yearly mean values (94.7 mm vs. 104.3 mm) and standard deviations (69.13 mm vs. 40.64 mm) of the daily maxima, respectively. While the differences are reasonable especially for return periods of 5 years and less, the effect of this discrepancy, driven mainly by

the big difference in the standard deviation, are accentuated for higher return period intensities. As for the temporal scaling factor $\beta$, the fitting to the hourly precipitation data reveals that $\beta$ decreases with increasing return periods where a value of 0.858 corroborates the average calculated through literature. Taking into account the variation in $\beta$ relative to the return period improved the goodness of fit for the temporal scaling function. However, it was not sufficient to offset the discrepancy between the two data series.

In regards to the validation and calibration data, it is a well-known problem that reliable measurements of flood depth and extent during urban floods are hard to acquire (Wang et al., 2018). This study could fortunately rely on reported inundation depths and locations across HCMC that were provided by local partners. To remedy this limitation, it could be argued that existing surveillance cameras throughout cities could be used to monitor time-varying water levels during flooding (Muhadi et al., 2021), which can either be done manually (Liu et al., 2015) or automatically (Moy de Vitry et al., 2019; Feng et al., 2020), providing crucial validation data that could go a long way in helping urban flood models to become more accurate without additional costs. Furthermore, user-generated images can also offer an additional way of quantifying flooding (Ahmad et al., 2018), whose acquisition became much easier with the proliferation of social media (Chaudhary et al., 2020).

Open-access data do not usually offer the detail required to build models to estimate flood damage, which typically require extensive data, whose acquisition is oftentimes laborious and prohibitively costly. The $I_{NFS}$, presented in Section 2.3.2, combines flood depth and duration from a hydro-numerical model that may further be used as input of flood damage models. The comparison with inundation hotspots across HCMC as documented by local partners, proved the usefulness of this indicator in estimating concentrated flood risk. Equal weighting was given for both flood depth and duration to ensure that the results are not biased, especially considering the lack of additional data clarifying whether flood depth or duration plays a bigger role in damage for a particular location. This weighting can be different depending on the case and the local composition of flood damage. Future users are, of course, free to change the weighting and adapt it to a specific use case.

One limitation of the $I_{NFS}$ can be seen in the exclusion of flow velocity, which was shown to play a significant role in pedestrian casualties (Musolino et al., 2020). However, quantifying this component can only be done through highly resolved flood models for particular city districts where flow obstacles can be accurately represented. Furthermore, flow velocity demonstrably plays a secondary role in LECZs where urban or rural terrain is rather flat (Wagenaar et al., 2017; Amadio et al., 2019). In such settings, the impact of flow velocity is rather small when compared to those of flood depth and duration, particularly for estimating monetary loss (Kreibich et al., 2009), and even more so in the rainfall-runoff scheme presented here. Nevertheless, through the proposed methodology, open-access data can be leveraged to determine urban areas with high damage potential where the procurement of highly resolved data for a more detailed flood model is required. In these highly resolved models, even flow velocity can be considered to quantitatively determine the associated risk to pedestrians. Moreover, it can be argued that the $I_{NFS}$ lacks the detail as well as the complexity of sophisticated flood damage models that are based on much more extensive and comprehensive data. However, the purpose of the $I_{NFS}$ concept and demonstrated application is not to replace established flood damage estimations but rather to complement these by enhancing the basic interpretation of hydro-numerical results through the combination of flood depth and duration. This makes the $I_{NFS}$ an effective tool in terms of a first

estimation when striving to determine inundation hotspots by robust mathematical models with high damage potential that demand attention in terms of emergency efforts and/or relief. This tool enables stakeholders as well as researchers to narrow down the focus to those areas with the highest damage potential in order to advance adaptation schemes under climate change and its projected impacts to LECZs (Scheiber et al., in review).

## 5 Conclusion

Hydro-numerical models are a powerful instrument to understand the dynamics of urban flooding, assess areas of exposure (flooding hotspots) and progress possible mitigation strategies. In many settings, however, essential information about topographic, bathymetric and hydro-meteorological constraints is hard to acquire without substantial costs, rendering independent but trustworthy analyses and evaluation for adaptation measures difficult, especially when such studies are to be done on wider scale. The present paper addresses this shortcoming and presents a methodology to create a surface runoff

model, which is capable of producing urban flood estimations for the exemplary case of HCMC, albeit solely based on open data sources according to the FAIR principles (GO FAIR, 2016). The process used to build this schematic yet flexible model can, at least partially, be used to simulate flood drivers in any urban setting. In addition, a newly proposed flood intensity proxy with a 2-parametric representation of flood depth and duration, the normalized flood severity index ($I_{NFS}$), is defined as a means of localizing potential flood damage hotspots. The $I_{NFS}$ uncovers flooding hotspots in HCMC, whereby 73% of the

more than 300 reported inundations were inside or within 100 m of the spatial extent of the $I_{NFS}$ that, in turn, covered only 19% of the total area of the city. The employed methodology for the model setup alongside the enhancement of the $I_{NFS}$ is particularly helpful when trying to localize inundation hotspots where the procurement of highly-resolved data for more detailed urban flood modelling is more worthwhile. The findings add to the current research in urban hydrological modelling and flood risk management and exemplify, which opportunities lie in the continuously growing amount of freely available

data. At last, it hopefully encourages researchers to make their work accessible and thus contribute to independent and more equal sciences.

## Code availability

No code was used in this research. Details about the general processing of numerical data are provided in the methods section or can be inquired from the corresponding author.

## Data availability

The references and freely available data used in this study can be accessed through the respective journals or databases.

**Author contributions**

LS, MHJ and CJ developed the methodology for acquiring, processing and comparing the open-access data, which was then executed by MHJ. LS, MHJ and JV designed the hydro-numerical model finally set up and operated by MHJ. HQN provided the hydro-meteorological data required for validation. LS and MHJ developed the Normalized Flood Severity Index. LS and MHJ developed the underlying paper concept. MHJ and LS wrote the initial manuscript, while CJ, JV, HQN and TS edited and contributed to the final text. LS and MHJ contributed to the visualization of the results. JV and TS (co-)designed the overarching research project, were responsible for funding resources and provided guidance throughout the entire study.

**Acknowledgements**

The authors wish to express their gratitude towards Dr. Nguyen Quy from EPT Environment & Target Public Ltd for providing us with the locations and depths of reported inundations across a variety of flood events in Ho Chi Minh City that were necessary for the model validation. Moreover, sincere thanks go to both the editor at NHESS for handling the manuscript and two anonymous reviewers for their helpful comments.

**Financial support**

This research has received funding from the DECIDER project sponsored by the German Federal Ministry of Education and Research (BMBF; grant no. 01LZ1703H).

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
