# Peer review of "The Potential of Open-Access Data for Flood Estimations: Uncovering Inundation Hotspots in Ho Chi Minh City, Vietnam, through a Normalized Flood Severity Index"

_Natural Hazards and Earth System Sciences, 2022_

## Referee Comment (RC1)

**Review of the manuscript "Uncovering Inundation Hotspots through a Normalized Flood Severity Index: Urban Flood Modelling Based on Open-Access Data in Ho Chi Minh City, Vietnam" by M. H. Jalloul, L. Scheiber, C. Jordan, J. Visscher, H. Q. Nguyen & T. Schlurmann submitted for publication in "Natural Hazards & Earth System Sciences"**

**General comments**

The authors present some kind of methodology to build a 2D hydraulic model based on freely available data. The objective is praiseworthy but since these data appears to be of relatively low quality and having in mind the sensitivity of a hydraulic model to the DEM for flood simulation, it seems pointless, even dangerous. Indeed, an urban flood model have to be of high quality (DEM, hydraulic calibration and validation) having in mind the repercussion of modelling results. The authors presents an interesting discussion on the DEM uncertainties based on freely available data. They should try to propagate these uncertainties using the numerical model; it could lead to any kind of results. Most institutions or insurance companies will use flood maps provided by scientists as a truth. If the model is not properly calibrated nor validated, it may lead to very problematic situation for people living in these areas. Clearly here, for the case of the River Sai Gon next to Ho-Chi-Minh-City (very flat system largely influenced by tide, complex system of canals, heavy rains, etc.), a numerical model of the city needs data of much better quality for the construction and validation of the model. Is it reasonable to have a DEM resolution of 30 m or more with a vertical uncertainty up to 1 m to build a 2D numerical model? Eventually, the proposed model is not really calibrated nor validated. Results in Fig. 5 are correct but there are many unclear assumptions behind. And main results presented in Fig. 7 are quite poor. In general, although the manuscript is well written, many technical details are missing. It is often difficult to understand how the bathymetry and boundary conditions are built.

The authors introduce a new index to evaluate the flood risk (normalized flood severity index), which can be interesting. However, they should verify if the normalization with a maximum value cannot bias the result in case of numerical divergence. Also, since the results of the model are quite poor, it appears difficult to validate the use of the index here. This index should be discussed for a case, which is much better described and a numerical model that is of higher quality. Flood hazard assessment for pedestrian often combine water depth and flow velocity (Musolino et al., 2020). Since this criteria is based on results from a 2D model, it could be interesting to introduce a second index based on velocity and duration. Anyway, this part of the paper appears a little bit off-topic.

**Minor comments**

- L32: For a list of reference, use the chronological order
- L34: next decade**s**
- L40 skip "C.R."
- L75 (Figure 1): I do not see any step of calibration and validation of the model
- L79: What do you mean by "similar sources"?
- L137: Please detail the characteristics of the Lidar data
- Tab. 2: An error of one meter for a DEM is huge! How accurate can you be for hydrodynamic calculations?
- Fig. 2: use (a), (b) etc. Instead of (A) (B); $3 \times 3$ instead of $3\,x\,3$ (times and not x-letter)
- L157: In many countries such as in Vietnam, bathymetric data exist and could be obtained through collaborations or by paying for it.
- L158: Again, such data base provides very rough estimations of the bathymetry. How accurate will be the model using such data?
- L172: What is the reference here? How do you set the bed level of the canal? Is this averaged depth a tidal-averaged depth?
- L180: "expedient" is maybe a little bit strong. For the moment, the model construction seems very crude, especially for a complex and very flat system such as the Ho-Chi-Minh-City area.

- Fig. 3: Please provide a proper figure caption and not a discussion of the figure. Also, most of the legend has no clear meaning (i.e. difference, exemplary colours, etc. ?). What do A, B, and C red squares mean? I guess they correspond to the Lidar samples.
- L190: I'm not sure I understood. Are building represented as non flowing area? Or is an equivalent Manning friction coefficient used to represent building effects on the averaged flow velocity?
- L195: The Manning coefficient has a unit; don't use the term "roughness coefficient" while talking about the Manning friction coefficient
- L197: Is a unique roughness coefficient used for the whole model ( $n=0.1$ s/m$^{1/3}$ )? What about canals and main channels (Sai Gon and Dong Nai Rivers)?
- L222: The Sai Gon water discharge is mostly influenced by the tide (Camenen et al., 2021). They provide some estimation of the net discharge for the years 2017-2018.
- L238: So, as far as I understood, you had access to Nha Be data.
- L239: It would be interesting to present a plot showing these results
- L255: variables in italic: $n=28$
- Eq. 1: functions in roman: $n=28$ ; define all variables introduced in this equation
- L260: $\alpha$ ?
- L263: The variable $n$ is already introduced for a number of years
- Eq. 2: This is not an equation; to be written within the text
- L269: Do you mean $\beta=0.854$ for the Ho-Chi-Minh-City area?
- L294: Arguable
- L304: This is not a proper argument. If there is some protection measure, there won't be any flow toward some of the lowest elevations. These zones may be eventually flooded but for other reasons (rain, groundwater, etc.) and so with a different dynamics.
- L308: It would be interesting to present this reference. And this methodology is also arguable. If this reference is not realistic compared to observed flooded zones, how can we trust simulations with more extreme conditions?
- L321: Flow depth is often not sufficient to evaluate the risk for people. One also needs the flow velocity (which can be provided by a 2D model of properly calibrated)
- L334: This sentence should appear after the introduction of Eq. 3
- Eq. 3: Even if this error is very common, it is not correct to introduce a variable made of multiple letters, i.e. $NFSI=N\times F\times S\times I$ . I would suggest to write:

$$I_{NFS}(x,y)=\frac{z_{max}(x,y)\times D_o(x,y)}{\max(z_{max}(x,y))\times \max(D_o(x,y))}$$

Isn't it a problem to use the maximal flood depth and duration as a reference. If the model provides some local unrealistic values for $z_{max}$ and or $D_o$ , it would significantly affect the results.
- L346: This is a significant issue. In many cases, institutions or insurance companies will use such flood maps as a truth. If the model is not properly calibrated nor validated, it may lead to very problematic situation for people living in these areas…
- L352: What about calibration?!
- L354: What about discharge and water level (tidal) conditions on the River Sai Gon?
- L357: Is this specific event representative of all events occurring on the HCMC area? Are there some cases with high discharges for the River Sai Gon and/or strong tidal effects for which the model could also be validated?
- Fig. 5: Do not add a linear regression when comparing simulation to observation; I see only 14 points on the plot whereas 25 are shown on the map. As far as I understood, the simulated water depths correspond to a difference between simulation results and results of the simulation for the 3h1y rain event with mean tide and mean river discharge. How sensitive are the results to this choice?
- L363: Just tot be sure I understood, you increased the Sai Gon bed level from +8.4 m (above see level?!) to 14.8 m (Fig. 6). Is it realistic? Anyway, I'm amazed that such variations don't

affect the results. How deep is the River Sai Gon for normal flows?

- L365: How were selected these three points?
- Fig. 6: Define the location of the points where sensitivity analysis is provided on the map Fig. 3 (use other letters since A, B, and C corresponds to other areas) and present plots only. Add a proper scale with axis legend for the three plots (or 4 if you include Nha Be water level time series)
- L375: There is no Fig. 6a and b. If you're talking about the plots in Fig. 6, it is not clear for me how you evaluate mFD and DoT from these plots.
- L377: How do you explain this behaviour? Is it based on observations from the field or from the numerical results?
- L380: What do you mean by "highlights previously hidden inundation hotspots"? Again, if the model is not really validated (at least not everywhere in the studied area), how sure are you about such results?
- L381: "considerable spatial overlapping"! I'm not that enthusiastic. Most of the reported inundation points do not overlap with the zones with a NFSI>0! What about all the zones with a high NFSI value? I can understand there is also a bias in the reported inundation points but you cannot say here that results are good.
- Fig. 7c: It is not very consistent to compare the flood severity index with reported inundation. A reported inundation corresponds to a water depth; so, these points should be compared to the modelled maximum flood depth (Fig. 7a). Again, do not provide comments of the figure in the figure caption (redundant with the text)
- L437: Due to the limitation of data to calibrate/validate the model, it is logical to use a single Manning friction coefficient for the whole domain. However, in reality, this coefficient should vary spatially depending on the city structure (presence of vegetation or not, porosity of the system, etc.)
- L474: True but the velocity is important in term of flood hazard for pedestrian (Musolino et al., 2020)
- L479: True but you need a robust and well calibrated model
- L491: I'm not sure such model can be used to simulate flood drivers, even partially.
- L528: Use European convention for dates: 12/06/2018
- L531: Use "doi:" instead of the full link "https/::doi.org/"
- L541: de Andrés, M.; be homogeneous with journal title (abbreviated or not)
- L546: Use capital letters for acronyms only, i.e. Bennghe Port Company Limited
- L554: Initials for first names after the name
- L557: Add all authors (instead of "et al."), initials of authors
- L567: reference?!
- L574: date, doi
- L595: Use capital letters for acronyms only, i.e. Go Fair
- L602: Skip "available at ..."
- L608: Skip "available at …"
- L614 NGO?!
- L615: journal?!
- L630: Explain the acronym JICA
- L638: de Moel, H.
- L672: Add all authors (instead of "et al."), initials of authors
- L685: Explain all acronyms
- L689: Don't use capital letters for the title and journal (International Journal of Geomate), some co-authors are missing
- L701: Skip "available at …"
- L716: Add all authors (instead of "et al."), initials of authors; Don't use capital letters for the journal name (?); Add (in Vietnamese)
- L726: Don't use capital letters for the author name

- L740: Skip references in review
- L776: Don't use capital letters for the author name

**Additional references**

Camenen, B., Gratiot, N, Cohard, J.-A., Gard, F., Tran, V.Q. , Nguyen, A.-T., Dramais, G., van Emmerik, T. & Némery, J. (2021). Monitoring discharge in a tidal river using water level observations: Application to the Saigon River, Vietnam. Science of The Total Environment, 761 (143195), doi: 10.1016/j.scitotenv.2020.143195.

G. Musolino, G., Ahmadian, R. & Falconer, R.A (2020). Comparison of flood hazard assessment criteria for pedestrians with a refined mechanics-based method, Journal of Hydrology X, 9(100067), doi: 10.1016/j.hydroa.2020.100067

---

## Author Comment (AC1)

RESPONSE TO REVIEW #1

We highly appreciate and are very thankful for the time and effort that was invested in reviewing our manuscript. Thank you for initiating this fruitful discussion. After carefully studying the constructive queries and comments, and following lengthy discussions among the co-authors, we have thoroughly revised our manuscript in an attempt to refine our key motivation and messages: to derive a simple but reliable methodology for localizing urban inundation hotspots by means of a numerical model, which makes best use of open access (geo) data, and a new and easy-to-apply flood severity index. Please find our responses (blue) and revised text blocks (*blue, italic*) below each review comment (**black, bold**).

**General Comments:**

**Part I:**

**The authors present some kind of methodology to build a 2D hydraulic model based on freely available data. The objective is praiseworthy but since these data appears to be of relatively low quality and having in mind the sensitivity of a hydraulic model to the DEM for flood simulation, it seems pointless, even dangerous. Indeed, an urban flood model have to be of high quality (DEM, hydraulic calibration and validation) having in mind the repercussion of modeling results. The authors present an interesting discussion on the DEM uncertainties based on freely available data. They should try to propagate these uncertainties using the numerical model; it could lead to any kind of results. Most institutions or insurance companies will use flood maps provided by scientists as a truth. If the model is not properly calibrated nor validated, it may lead to very problematic situation for people living in these areas. Clearly here, for the case of the River Sai Gon next to Ho-Chi-Minh-City (very flat system largely influenced by tide, complex system of canals, heavy rains, etc.), a numerical model of the city needs data of much better quality for the construction and validation of the model. Is it reasonable to have a DEM resolution of 30 m or more with a vertical uncertainty up to 1 m to build a 2D numerical model? Eventually, the proposed model is not really calibrated nor validated.**

We are thankful for receiving this constructive feedback and are reassured in our motivation to disseminate our findings, given that the reviewer sees the objective as praiseworthy, too. This opinion confirms the added value of communicating the presented methodology for building urban surface runoff models based on open-access data to a wider audience. Upon carefully examining this general comment, we came to the conclusion that the purpose of our methodology was not

communicated as clearly as intended. To clarify our overarching motivation, changes were made to the manuscript starting off with the title which now reads: *"Uncovering Inundation Hotspots through a Normalized Flood Severity Index: The Potential of Open-access Data for Flood Estimations in Ho Chi Minh City, Vietnam"*

This should reduce the misleading impression that open-access models could be the "be-all and end-all" instrument for producing highly accurate flood maps. In contrast, the title now emphasizes the inherent uncertainty and limitations introduced by using open-access data for this purpose. Furthermore, the abstract was modified to better reflect the main objective of the presented work and now reads as follows (ll. 18-21):

*"(…) To help alleviate this problem, this paper explores the usability and reliability of flood models built on open-access data in regions where highly-resolved (geo)data (e.g., from LiDAR campaigns, bathymetric surveys or hydrological data acquisition) are either unavailable or difficult to access, yet evaluation of risk from flooding is crucial. To that end, the example of Ho Chi Minh City, Vietnam, is taken to describe a comprehensive methodology for obtaining, processing and applying the necessary open-access data (topography, bathymetry, tidal water level, river flow and precipitation time series) to the fullest. The goal is to produce preliminary flood maps that provide first insights and estimations about potential flooding hotspots that demand closer attention in subsequent, more detailed flood risk analyses. As a key novelty of the paper, a normalized flood severity index ($I_{NFS}$) that combines flood depth and flood duration is proposed. The index serves as an indicator that further narrows down the focus to areas of significant flooding. It helps to uncover elements at risk, where particular scientific or practical attention is needed, be it in terms of precautionary relief efforts or training to prepare in advance to cope with flood risks (…)"*

Furthermore, the wording of the introduction (ll. 45-46) has also been modified to better articulate our intentions regarding the cost and time intensive character of data acquisition and processing of on-site procured high resolution data. It now reads as follows:

*"(…) which complicates numerical studies, especially for independent parties. Furthermore, when made available, not only are these data sets prohibitively costly, but they also often lack the necessary spatial and temporal coverage needed for proper derivation of boundary conditions and model set-up."*

These amendments to the manuscript should make it clearer for any reader that the open-access data flood model of HCMC does not promise to deliver results that can be considered as truth, but rather estimations that open up opportunities to gain insights for subsequent decision-making processes regarding more detailed modeling for critical areas. The presented methodology can also be seen as an orientation for city planners and authorities from the developing world, helping them to readily estimate where hotspots with particularly high damage potential are located in a first flood risk assessment. Furthermore, it is not uncommon to find regional flood studies of HCMC that rely on coarse terrain data. For example, Scussolini et al. (2017) used a terrain mesh that ranged from 100 m to 500 m for their regional flood model, while Nhut Duy et al. (2019) relied on a 1-D model with 1000 data points on a 15 m grid for the river network and 28600 points on a 15 m grid for built-up areas. Undoubtedly, Progress has been made in flood modeling in recent years, yet highly resolved (geo)data is neither readily available nor always accessible to independent users. The presented manuscript deals with those situations, where highly resolved data is missing or inaccessible.

Lastly, we would like to address the impression of the reviewer that the model was not properly calibrated nor validated. In our opinion, this comment is not fully justified but we acknowledge the lack of emphasis in the presentation of calibration results and therefore added the following table to section 2.1.3 to avoid any misunderstanding:

| *Model calibration for different Manning friction coefficients focusing on reported inundations during three rain events (left column) and corresponding RMSE, NSE and PBIAS values* | | | | | | | | | |
|---|---|---|---|---|---|---|---|---|---|
| | *n = 0.08 s/m$^{1/3}$* | | | *n = 0.10 s/m$^{1/3}$* | | | *n = 0.12 s/m$^{1/3}$* | | |
| *Calibration Events* | *RMSE* | *NSE* | *PBIAS* | *RMSE* | *NSE* | *PBIAS* | *RMSE* | *NSE* | *PBIAS* |
| *Event 1 Date: 01/07/2010 P = 79 mm HWL = 1.10 m 23 Observations* | *0.02* | *-5.25* | *37.5* | *0.01* | *0.50* | *5* | *0.02* | *-1.75* | *-25.6* |
| *Event 2 Date: 09/07/2012 P = 58 mm HWL = 1.12 m 19 Observations* | *0.03* | *0.14* | *21.4* | *0.02* | *0.64* | *10.7* | *0.03* | *0.29* | *-15.3* |
| *Event 3 Date: 01/10/2012 P = 74 mm HWL = 1.15 m 18 Observations* | *0.04* | *-3.23* | *33.7* | *0.03* | *0.52* | *6.2* | *0.05* | *-1.42* | *-17.9* |

The table shows that a Manning friction coefficient of 0.10 s/m$^{1/3}$ does indeed provide the best results for all three statistical parameters. Albeit far from 100% accuracy, a NSE value of

0.5 to 0.64 is satisfactory for first flood estimates and is sufficient for a model whose goal is to determine inundation hotspots rather than quantitatively predict flood depths. This is especially valid when compared to the flood model by Hoa Binh et al. (2019) which relied on non-open-access 2 m resolution LiDAR data and still achieved NSE values of 0.51 to 0.89. Last but not least, our method of calibration is better designated for rain events as it relies on flood depths measured within the city and not on discharge and tidal gauges that are remote from the affected urban areas. Considering our results against the backdrop of comparable models, we are confident about the robustness of the presented approach.

**Part II:**
**Results in Fig. 5 are correct but there are many unclear assumptions behind. And main results presented in Fig. 7 are quite poor.**

We have made special efforts to edit Fig. 5 so that the assumptions are clearer. Furthermore, a table was composed, where flood depths observed at scattered locations and their corresponding simulated flood depths can be compared. The geolocations are numbered and supplemented by street names that are depicted on a map that can be part of the Supplementary Material. In regards to Figure 7, it is worth noting that this figure is meant to exemplify the application of the $I_{NFS}$ by highlighting the differences between the maximum flood depth, inundation duration over threshold and their combination in the form of the $I_{NFS}$. Furthermore, this figure is best understood when simultaneously looking at the results in Table 4, which clearly highlight how accurate and trustworthy the $I_{NFS}$ was in covering the locations of reported inundation as opposed to the other two flood indicators. Nevertheless, we edited the text to better define the goal of this section.

**Part III:**
**In general, although the manuscript is well written, many technical details are missing. It is often difficult to understand how the bathymetry and boundary conditions are built.**

Regarding the bathymetry, section 2.1.2 entitled "Bathymetric Data" is completely dedicated to explaining and discussing how the bathymetric data was acquired and integrated into our model. Admittedly, this methodology was proposed mainly because of the lack of comprehensive bathymetric data for the model area even after consultation with local partners. Unfortunately, it is not always the case that local institutions or authorities have knowledge about or are mandated to grant access to available geodata. Nevertheless, we developed an additional figure for the bathymetry that complements our original explanations.

**Part IV:**

**The authors introduce a new index to evaluate the flood risk (normalized flood severity index), which can be interesting. However, they should verify if the normalization with a maximum value cannot bias the result in case of numerical divergence. Also, since the results of the model are quite poor, it appears difficult to validate the use of the index here. The index should be discussed for a case, which is much better described and a numerical model that is of higher quality.**

We are very pleased that the reviewer regards the original idea and derived concept of the $I_{NFS}$ as valuable, given that its proof-of-concept was one of the primary motivations for submitting this manuscript. Yet, we acknowledge the concerns regarding result bias due to false maximum values in case of numerical divergence. In fact, we thoroughly examined the simulation results in order to exclude any divergence, artifacts or outliers, which in our case were not found. Based on this comment, we decided to increase the robustness of the $I_{NFS}$ against divergence and outliers by relying on quantiles of flood depth and duration for normalization. Through this method, the maximum flood depth is capped to the 95th quantile, keeping the value of the $I_{NFS}$ between 0 and 100, while eliminating potential artifacts due to numerical divergence.

**Part V:**

**Flood hazard assessment of pedestrian often combine water depth and flow velocity (Musolino et al., 2020). Since this criteria is based on results from a 2D model, it could be interesting to introduce a second index based on velocity and duration. Anyway, this part of the paper appears a little bit off-topic.**

The combination of water depth and velocity is definitely an interesting prospect to determine risk to pedestrians in an urban environment and needs to be examined in more detail. The decision to neglect the velocity component from integration into the index was based on its negligible impact on flood damage modeling attempts (Amadio et al., 2019; Wagenaar et al., 2017, Kreibich et al., 2007) in low-elevation coastal zones (LECZ), where urban or rural terrain is rather flat, putting economic damage rather than pedestrian casualties in focus. This argument holds true when considering the high economic damage caused by less severe but much more frequent urban floods that Ho Chi Minh City regularly suffers from (ADB, 2010). Furthermore, the nature of the presented surface runoff model, where barriers such as buildings and vegetation cannot be easily represented, does not allow for the computation of peak flow velocities due to changes in

cross-section. The proposed combination could be useful for a more detailed model for certain areas or districts where a surface elevation model with a fine resolution (5 m or lower) can be built. Through our proposed methodology, the areas of greater risk (hotspots) can be identified where more detailed simulations are worthwhile.

**Minor Comments:**

1- **L32: For a list of reference, use the chronological order**

Thank you for this comment. References with multiple entries were changed accordingly.

2- **L34: next decades, L40: skil "C.R", Fig 2. Use (a), (b), etc. Instead of (A), (B): 3×3 instead of 3x3 (times and not x-letter)**

Thank you for these corrections, the text was changed accordingly

3- **L75 (Figure 1) I do not see any step of calibration and validation of the model**

This is a valid point and we amended said figure accordingly.

4- **L79: What do you mean by "similar sources"?**

We have taken this comment into consideration and changed the wording of this sentence that now reads as follows:

*"Generally, the search priority of terrain data, as well as hydro-meteorological data, follows the same path, with official sources at the top, followed by global repositories, peer reviewed literature, grey literature (i.e. publicly available reports and assessments), and finally regional and global models."*

5- **L137: Please detail the characteristics of the LiDAR data**

A sentence was added that lists the corresponding characteristics.

6- **Tab. 2: An error of one meter for a DEM is huge! How accurate can you be for hydrodynamic calculations?**

We agree with the reviewer that differences of one meter are significant for a DEM, which is why special emphasis was put on the discussion of the differences in Section 4. However, the use of

difference plots as described in 2.3.1 counteracts these uncertainties, which is confirmed by the model calibration and validation. Furthermore, it is important to measure the amplitude of the bias of the proposed DEM with regards to other open-access DEMs (SRTM, ALOS, ASTER, COPERNICUS). The positive bias of these traditional satellite DEMs can reach up to 13 m vis-à-vis the LiDAR data samples, rendering them completely unreliable for flood modeling purposes. This corroborates the conclusion made by Hawker et al. (2018) on the global scale in regards to the usability of the existing global DEMs. In this regard, the proposed DEM of this manuscript is far more reliable than any other open-access DEM and can confidently be used in preliminary flood estimations.

**7- In many countries such as in Vietnam, bathymetric data exist and could be obtained through collaborations or by paying for it**

Thank you for this comment. According to our knowledge and local networks, no open-access data exists for the Sai Gon River, while open-access bathymetric data for the Dong Nai River stems from US Army Corps of Engineers maps created in 1965 (Gugliotta et al., 2020). It is also correct that bathymetric data is available for sale. However, it is mostly provided in deep sections (e.g. river mouths) for transportation purposes and in hard-copy only. Accordingly, access and use of data from HCMC, if available, would be limited by commercial interests. This fact again underlines the need for the utilization of open-access data in flood modeling, which is overarching the objective of our manuscript.

**8- L158: Again, such data base provides very rough estimations of the bathymetry. How accurate will be the model using such data?**

We are pleased that the reviewer raises this point. This is exactly why we did the sensitivity analysis, whose results are presented in Section 3.2, showing that even a depth change of +80% of the river bed influences urban flood depths by only a few centimeters (7 to 12 cm).

**9- L172: What is the reference here? How do you set the bed level of the canal? Is this average depth a tidal-average depth?**

The canal depths are given relative to mean sea level. This detail was incorporated in the revised manuscript.

**10- L180: "expedient" is maybe a little bit strong. For the moment, the model construction seems very crude, especially for a complex and very flat system such**

**as the Ho Chi Minh City Area**

Thank you for this comment. We agree with the reviewer's opinion on the wording and have omitted the word "expedient" from line 180.

11- **Fig. 3: Please provide a proper figure caption and not a discussion of the figure. Also, most of the legend has no clear meaning (i.e. difference, exemplary colours, etc. ?). What do A, B and C red squares mean? I guess they correspond to the LiDAR samples**

Thank you for pointing out these mistakes and unclarities. We fixed the figure according to your comments to enhance its readability.

12- **L190: I'm not sure I understood. Are buildings represented as non-flowing area? Or is an equivalent Manning friction coefficient used to represent build effects on the average flow velocity?**

We agree that this sentence might have caused misinterpretations. Therefore, the sentence (ll 190-195) was adjusted and now reads as follows:

*"Buildings and extensive vegetation that significantly reduce the available cross-section for water routing are not represented in the final DEM. Furthermore, given the 1 arc second spatial resolution, structural footprints of buildings cannot be represented as no-flow areas. Instead, an equivalent Manning friction coefficient was adjusted accordingly to compensate for no-flow areas that would otherwise be hydraulically misattributed."*

**:L195: The Manning coefficient has a unit; don't use the term "roughness coefficient" while talking about the Manning friction coefficient**

Thank you for this valuable comment, the text was adjusted accordingly by substituting the term "roughness coefficient" with "Manning coefficient" and by adding its corresponding unit.

13- **L197: is a unique roughness coefficient used for the whole model? (n=0.1 s/m$^{1/3}$)? What about canals and main channels (Sai Gon and Dong Nai Rivers)?**

We are aware that the wording might have caused misinterpretation in reference to the application of Manning friction coefficients. Therefore, the wording of this sentence was changed and now reads as follows:

*"Following this approach, the best results are obtained for a Manning coefficient of 0.1 s/m1/3 uniformly applied across the whole modeling domain, (…)"*

**14- L222: The Sai Gon water discharge is mostly influenced by tide (Camenen et al., 2021). They provide some estimation of the net discharge for years 2017-2018**

Thank you for sharing this reference. We have examined the given net discharges of 30 and 65 m3/s for the years 2017 and 2018, respectively, and are satisfied that our chosen net discharge of 54 m3/s, which is the long-term net discharge according to Trang Ngoc et al. (2016), falls within this range. Nevertheless, we now mention these additional values in our manuscript as part of Section 2.2.1.

**15- L238: So, as far as I understood, you had access to Nha Be data**

This is correct. We had access to both tidal water level data for Nha Be and rainfall data for the Tan Son Hoa rain station from our local partner. We used this data to critically evaluate the reliability of the open-access data and have specified the results of the comparison for both tidal water levels and rainfall data in Section 2.2.2 and 2.2.3.

**16- L239: It would be interesting to present a plot showing these results**

We found this idea very interesting as well and developed a graph that summarizes these results. Given that parts of this information are inherently contained in the publication by Gugliotta et al. (2017), we are deliberating, whether the illustration should be included in the revised manuscript or the Supplemental Material.

**17- L255: variables in italic: *n*=28, Eq. 1; functions in roman: *n*=28; define all variables introduced in this equation, L263: The variable *n* is already introduced for a number of years, Eq. 2: this is not an equation; to be written within the text**

Thank you for these comments, we have corrected the manuscript accordingly.

**18- L260: α?**

The value of α serves to indicate the goodness of fit of a theoretical distribution to the actual data as developed by Dyck (1980). This should show that the Gumbel distribution delivers a very good fit for the rain data set made available by NOAA for the Tan Son Hoa rain station.

**19- L269: Do you mean ß=0.854 for the Ho Chi Minh City area?**

Correct. We have added the reference Ho Chi Minh City to avoid confusion.

**20- L294: Arguable**

We understand the reviewer's concern regarding our conclusion. The question to be asked here is whether these derived boundary conditions are of sufficient quality to be used for the generation of qualitative flood risk estimates (as opposed to quantitative and highly-detailed results). The comparison between the open-access rainfall data and the rainfall data provided by local partners for Tan Son Hoa shows that there are indeed notable differences as can be seen from Table 3. However, these differences are reasonable especially for return periods of 5 years and less, which are the focus of this study.

**21- L304: this is not a proper argument. If there is some protection measure, there won't be any flow toward some of the lowest elevations. These zones may be eventually flooded but for other reasons (rain, groundwater, etc.) and so with a different dynamic.**

We share the reviewer's concern regarding this point. However, the argument greatly depends on the nature of the protection measure being approximated. Larger-scale, above-ground measures (e.g., dikes, pumping stations, detention/retention ponds) could still be accurately represented in this DEM, especially in the case where the impact of adaptation measures is evaluated relative to a no-adaption base case (see Scheiber et al. (Preprint)). In contrast, underground stormwater drainage systems are harder to represent, especially when relying on open-access data. However, there is significant evidence for the ineffectiveness of the storm water drainage system in the case of Ho Chi Minh City (Le Phu et al, 2021, Q.T. Nguyen, 2016), so that a worst-case scenario modeling is conceivable (Scussolini et al, 2017). These drainage systems are not well maintained and have limited functionality, so that their capacity is heavily diminished (Nguyen et al., 2019). This warrants a conservative engineering approach, in which they are deliberately omitted in the modelling scheme.

**22- L308: It would be interesting to present this reference. And this methodology is also arguable. If this reference is not realistic compared to observed flooded zones, how can we trust simulations with more extreme conditions?**

We understand the concern of the reviewer in regards to this chosen reference. This reference is indeed pivotal to the results of our flood simulation. If the reference, is not correctly defined, then the model would not perform as intended. In the case of our study, this reference is a theoretical inundation layer from mean tidal and fluvial conditions which do not lead to inundations in reality. It was defined based on flooding threshold values that were determined through joint work with local partners as well as information from grey literature like the JICA reports (JICA, 2001; URL: https://openjicareport.jica.go.jp/618/618_123.html) as well as different media articles accessible at the following URLs:

https://www.c40.org/case-studies/mitigate-urban-flooding-in-ho-chi-minh-city-phase-1/

https://global.royalhaskoningdhv.com/projects/flood-management-in-ho-chi-minh-city-vietnam

https://borneobulletin.com.bn/poor-urban-development-cause-of-flooding-congestion-in-ho-chi-minh-say-experts/

https://e.vnexpress.net/news/news/major-anti-flooding-project-in-hcmc-misses-deadline-for-four-years-4444006.html

https://www.channelnewsasia.com/cnainsider/siege-climate-man-made-problems-sinking-ho-chi-minh-city-floods-2052231

The reference was thus set according to tidal time series for Nha Be presented in Fig. 6, which represents the employed flooding threshold for Ho Chi Minh City. The confirmation of this reference method is further reinforced by the calibration and validation results.

**23- L321: Flow depth is often not sufficient to evaluate risk for people. One also needs the flow velocity (Which can be provided by a 2D model of properly calibrated)**

The reviewer's argument is valid. 2D models that have a high spatial resolution to properly reflect changes in flow cross sections for urban environments can definitely deliver flow velocities relevant to evaluate risk for people. This, however, does not belong to the scope of our study, especially because our model is not appropriate for such a purpose and because high flow velocities do not pose a significant threat in a flat LECZ such as HCMC as explained in Section 4.

**24- L334: This sentence should appear after the introduction of Eq. 3**

Thank you for this clarification, we have changed the text accordingly.

**25-** Eq. 3: even if this error is very common, it is not correct to introduce a variable made of multiple letters, i.e. NFSI = N×F×S×I. I would suggest to write:

$$I_{NFS}(x, y) = \frac{z_{max}(x, y) \times D_o(x, y)}{max\big(z_{max}(x, y)\big) \times max(D_o(x, y))}$$

Isn't it a problem to use the maximal flood depth and duration as a reference/ If the model provides some local unrealistic values for $z_{max}$ and or $D_o$, it would significantly affect the results.

We find this comment very valuable and have implemented the suggested change. In regards to the effect of using the maximal flood depth and duration, we would like to refer to our answer to the general comments Part IV on the use of 95th quantiles.

**26-** L346: This is a significant issue. In many cases, institutions or insurance companies will use such flood maps as truth. If the model is not properly calibrated nor validated, it may lead to very problematic situation for people living in these areas.

We agree with the reviewer's comment. A model that is meant to deliver highly accurate flood depths, durations and velocities that can be considered as truth should definitely be set-up, calibrated and operated on data with higher resolution and little uncertainty. However, the goal of this study is to assess the performance of open-access data (in cases, where that highly resolved information is inaccessible) in delivering first estimates of potential flooding as well as gaining understanding of underlying flood mechanisms. An open-access model built for regions where data is scarce can hardly be the "be-all and end-all" instrument for producing highly accurate flood maps but can still be very valuable for specific use cases, like determining inundation hotspots, especially when combined with the use of the INFS.

**27-** L352: What about calibration?

To alleviate the concerns of the reviewer, we have revised our explanations regarding model calibration and added a table under Section 2.1.3. Furthermore, we changed the title of the section which now reads as follows:

"*Hydraulic Roughness Coefficient and Model Calibration*"

**28- L354: What about discharge and water level (tidal) conditions on the River Sai Gon?**

The reviewer's question surely warrants a closer examination of the discharge and tidal conditions of the River Sai Gon. However, there is a considerable lack of data in regards to this particular waterway especially in open-access data, so that we are afraid nothing can be done in this regard.

**29- L357: Is this specific event representative of all events occurring on the HCMC area? Are there some cases with higher discharges for The River Sai Gon and/or strong tidal effects for which the model could also be validated?**

We have chosen the 14/06/2010 event for model validation, firstly, because the corresponding boundary conditions are known and, secondly, because flood depths were measured at different locations in HCMC during this event which is a pre-requisite for proper validation. Furthermore, our focus is on rain-induced flooding in Ho Chi Minh City and the special backwater effect caused by high tide, which is epitomized by this event (P=73 mm, WL=1.15 m). In regard to the discharges of the Sai Gon River, we have assumed that the reservoir mitigates higher discharge values associated with extreme river discharges. As for stronger tidal effects, such events can surely be simulated, but were not the focus of our study, which addresses frequent and disruptive rather than extreme flood events.

**30- Fig. 5: Do not add a linear regression when comparing simulation to observation; I see only 14 points on the plot whereas 25 are shown on the map. As far as I understood, the simulated water depths correspond to a difference between simulation results and results of the simulation for the 3h1y rain event with mean tide and mean river discharge. How sensitive are the results to this choice?**

We have taken the reviewer's comment into consideration and have removed the linear regression. Instead, we now provide a table, where the observed and simulated depth according to street name are given, and another table, which presents the results of the validation in addition to the graph. The simulated water depths are the difference between the results of the validation event (P=73 cm and HWL=1.15 m) on the day of the event (14/06/2010) along with mean river discharge and the results of the "reference" case which characterized by mean tidal conditions and mean discharge, but no rain, for which no flooding occurs. Both data collections can be provided either in the revised manuscript or as Supplementary Material.

**31- L363: Just to be sure I understood, you increased the Sai Gon bed level from +8.4 m (above sea level?!) to 14.8 m (Fig. 6). Is it realistic? Anyway, I'm amazed that such variations don't affect the results. How deep is the River Sai Gon for normal flows?**

We would like to thank the reviewer for this valuable comment. We have changed the text and Fig.6 so as to make it clear that the depth was varied from -8.4 to -14.8 m above mean sea level (MSL) in the context of our sensitivity analysis. This variation in depth of the Sai Gon River was done in order to determine the impact of this parameter on the simulated flood depth. We don't have robust information pertaining to the depth of the Sai Gon River, they were solely derived using the official navigational depth maintained for access to the ports along the river. The results of this sensitivity analysis show that a flood model could indeed be built on such an assumption without largely interfering with the robustness of the results when the goal is to generate flood risk estimates in an environment where data is scarce.

**32- :365: How were selected these three points?**

Point C was selected because it represents the outlet of the Ben Nghe canal, and Point B represents the intersection between the Ben Nghe and the Tau Hu canals, where frequent flooding occurs, while point A is a known inundation hotspot at the endpoint of the Tau Hu canal. The goal is to show the impact of the change in the depth of the Sai Gon on the maximum water levels at different distances upstream from the outlet of the Ben Nghe canal to the Sai Gon River. Point C is located directly at the Sai Gon River, Point B is located 3 Km upstream from Point C, while Point A is located 13 Km upstream from Point C.

**33- Fig. 6: Define the location of the pints where sensitivity analysis is provided on the map Fig. 3 (use other letters since A, B, and C corresponds to other areas) and present plots only. Add a proper scale with axis legend for the three plots (or 4 if you include Nha Be water level time series)**

Thank you for these valuable comments, we revised the figure accordingly.

**34- L375: There is not Fig. 6a and b. If you're talking about the plots in Fig. 6, it is not clear for me how you evaluation mFD and DoT from these plots.**

Thank you for pointing out this mistake, Fig. 7 was actually meant here. These figures serve to show the difference in spatial extent as well as spatial variability of mFD and DoT. The main argument is that areas with high mFD do not necessarily translate to areas with high DoT.

**35- L377: How do you explain this behaviour? Is it based on observations from the field or from the numerical results?**

Interesting question. This behavior can be explained by the fact that these areas have smaller catchments with lower concentration times and are thus drained more quickly. Furthermore, their proximity to the Sai Gon River also plays a role in the drainage behavior.

**36- L380: What do you mean by "highlights previously hidden inundation hotspots"? Again, if the model is not really validated (at least not everywhere in the studied area), how sure are you about such results?**

Good question. This sentence was meant to set aside the inherent bias of the spatiality of the reported inundations. However, after careful reviewing this paragraph, we have decided to omit this sentence. We cannot say for sure whether these results 100% reflect the truth for unvalidated areas but it would definitely be highly interesting to validate them accordingly. Unfortunately, no inundations were reported in the depicted areas.

**37- L381: "considerable spatial overlapping"! I'm not that enthusiastic. Most of the reported inundation points do not overlap with the zones with a NFSI>0! What about all the zones with a high NFSI value? I can understand there is also a bias in the reported inundation points but you cannot say here that results are good.**

We are aware that is it not very easy to determine the spatial overlapping using the color scheme presented in Fig. 7, which is exactly why we created Table 4, where it is obvious that the INFS map is 4 times as accurate in matching the locations of reported floods when compared to a random area with the same size. Nevertheless, we agree with the reviewer in this regard and intensely discussed whether the results can be presented in a way that effective spatial overlapping becomes clearer in Fig. 7 itself.

**38- Fig. 7c: It is not very consistent to compare the flood severity index with reported inundation. A reported inundation corresponds to a water depth; so, these points should be compared to the modelled maximum flood depth (Fig. 7a). Again, do not provide comments of the figure in the figure caption (redundant with the text)**

We understand the concern of the reviewer in this regard. However, the reported inundations that were used in this figure do not only correspond to places were high flood depths were recorder but also places that are persistently flooded over a relatively long time. What we wanted to

highlight here is that the INFS is better at identifying these areas as opposed to the flood depth, which does indeed cover more reported inundation, but performs worse relative to size and thus does not serve the purpose of the methodology, which is to better pinpoint areas that require closer attention and where the procurement of higher quality data is worthwhile.

**39- L437: Due to the limitation of data to calibrated/validate the model, it is logical to use a single Manning friction coefficient for the whole domain. However, in reality, this coefficient should vary spatially depending on the city structure (presence of vegetation or not, porosity of the system, etc.)**

We agree with the reviewer on this point. The spatial variation of the Manning friction coefficient can be and actually was considered. We left this out of the scope of this paper because the land use classes were not available in open-access. One might argue that remote sensing could be used to the determination of land use classes and then using that to determine the Manning friction coefficient of some parts of the city, but that would be out of the scope of this paper.

**40- L474: True but the velocity is important in term of flood hazard for pedestrian (Musolino et al., 2020)**

We thank the reviewer for this insightful comment. We may consider this caveat in our text but as stated in the responses to Part IV of the general comments of the reviewer, our model is not designed to forecast precise flow velocities, which also play a secondary role in a flat, low-lying system such as HCMC.

**41- L479: True but you need a robust and well calibrated model**

We agree that a more robust and model that is calibrated for more events can deliver more robust and trustworthy results, but we don't insist on this argumentation since the overall objective is led by another intention as previously pointed out in the answers to Part I of the general comments.

**42- L491: I'm not sure such model can be used to simulate flood drivers, even partially.**

It is an important fact that the provided methodology was calibrated and validated for rain events only. The aforementioned explanations regarding the scope and limitations of our modelling scheme, hopefully, add some context and robustness to the highlighted statement. In any case, we revised this sentence in the new version of the manuscript.

**43- L528: Use European convention for dates: 12/06/2018**

Thank you for this and the following plethora of comments and corrections, which is very helpful to ensure the correctness of this paper. The invested efforts are much appreciated. The bibliography was changed accordingly.

44- L531: Use "doi:" instead of the full link "https/: doi.org/"

45- L541: De Andrés, M.; be homogeneous with journal title (abbreviated or not)

46- L546: Use capital letters for acronyms only, i.e. Bennghe Port Company Limited

47- L554: Initials for first names after the name

48- L557: Add all authors (instead of "et al."), initials of authors

49- L567: reference?!

50- L574: date, doi

51- L595: Use capital letters for acronyms only, i.e. Go Fair

52- L602: Skip "available at…"

53- L608: Skip "available at…"

54- L614: NGO?!

55- L615: Journal?!

56- L630: Explain the acronym JICA

57- L638: de Moel, H.

58- L672: Add all authors (instead of "et al."), initials of authors

59- L685: Explain all acronyms

60- L689: Don't use capital letters for the title and journal (International Journal of Geomate), some co-authors are missing

61- L701: Skip "available at…"

**62- L716: Add all authors (instead of "et al."), initials of authors; Don't use capital letters for the journal name (?); Add (in Vietnamese)**

**63- L726: Don't use capital letters for the author name**

**64- L740: Skip references in review**

**65- L776: Don't use capital letters for the author name**

**References:**

Scussolini, P., Tran, T. V. T., Koks, E., Diaz-Loaiza, A., Ho, P. L., & Lasage, R. (2017). Adaptation to sea level rise: a multidisciplinary analysis for Ho Chi Minh City, Vietnam. *Water Resources Research*, *53*(12), 10841-10857.

Duy, P. N., Chapman, L., & Tight, M. (2019). Resilient transport systems to reduce urban vulnerability to floods in emerging-coastal cities: A case study of Ho Chi Minh City, Vietnam. *Travel behaviour and society*, *15*, 28-43.

Binh, L. T. H., Umamahesh, N. V., & Rathnam, E. V. (2019). High-resolution flood hazard mapping based on nonstationary frequency analysis: case study of Ho Chi Minh City, Vietnam. *Hydrological Sciences Journal*, *64*(3), 318-335.

Amadio, M., Scorzini, A. R., Carisi, F., Essenfelder, A. H., Domeneghetti, A., Mysiak, J., & Castellarin, A. (2019). Testing empirical and synthetic flood damage models: the case of Italy. *Natural Hazards and Earth System Sciences*, *19*(3), 661-678.

Kreibich, H., Müller, M., Thieken, A. H., & Merz, B. (2007). Flood precaution of companies and their ability to cope with the flood in August 2002 in Saxony, Germany. *Water resources research*, *43*(3).

Asian Development Bank, (2010). Ho Chi Minh City Adaptation to Climate Change. Manilla, Phillipines.

Hawker, L., Bates, P., Neal, J., & Rougier, J. (2018). Perspectives on digital elevation model (DEM) simulation for flood modeling in the absence of a high-accuracy open access global DEM. *Frontiers in Earth Science*, *6*, 233.

Gugliotta, M., Saito, Y., Ta, T. K. O., Nguyen, V. L., Uehara, K., Tamura, T., ... & Lieu, K. P. (2020). Sediment distribution along the fluvial to marine transition zone of the Dong Nai River System, southern Vietnam. *Marine Geology*, *429*, 106314.

Tran Ngoc, T. D., Perset, M., Strady, E., Phan, T. S. H., Vachaud, G., Quertamp, F., Gratiot, and N.: Ho Chi Minh City growing with water related challenges, UNESCO, Paris, France, 2016.

Le Phu, V., Lan, N. H. M., Tien, N. T. C., & Hiep, L. D. (2021, February). Sustainable Urban Drainage System Model for The Nhieu Loc–Thi Nghe Basin, Ho Chi Minh City. In *IOP Conference Series: Earth and Environmental Science* (Vol. 652, No. 1, p. 012012). IOP Publishing.

Nguyen, Q. T. (2016). The main causes of land subsidence in Ho Chi Minh City. *Procedia engineering*, *142*, 334-341.

---

## Author Comment (AC2)

RESPONSE TO REVIEW #2

We highly appreciate and are very thankful for the time and effort that was invested in reviewing our manuscript. Thank you for initiating this fruitful discussion. After carefully studying the constructive queries and comments, and following lengthy discussions among the co-authors, we have thoroughly revised our manuscript in an attempt to refine our key motivation and messages: to derive a simple but reliable methodology for localizing urban inundation hotspots by means of a numerical model, which makes best use of open access (geo) data, and a new and easy-to-apply flood severity index. Please find our responses (blue) and revised text blocks *(blue, italic)* below each review comment (**black, bold**).

**General comments:**

**Part I:**

**The paper demonstrated how to process and applying open-access data to an urban surface run-off model. The authors also combined flood depth and duration into a so-called normalized flood severity index (NFSI) to identify urban inundation hotspots. Overall, this paper might be useful in demonstrating how open access data can be processed into hydrological model. However, the methodology to achieve this need to be more systematically presented.**

We are very pleased to learn that Referee #2 agrees with our core idea and the usefulness of our approach in demonstrating how open-access data can be processed and incorporated into a hydrological model. Thank you for making us aware that the presented methodology is not always perfectly clear to the reader. The first step to remedy this was to add additional processing steps to the work flow presented in Figure 1 (e.g. validation/calibration, simplification of model inputs). Furthermore, more attention was given to the derivation of bathymetric data. In order to improve this aspect, we discussed adding a figure to Section 2.1.2 to better illustrate this process and/or to Section 2.2.2 that visualizes the derivation of the tidal boundary condition.

**Part II:**

**Whether this methodology can be considered novel or not is unclear as the process seemed quite intuitive. Besides, the applicability of the research is thin. Vulnerability assessment is being conducted in cities to identify the areas that need response measures. Moreover, the application of the flood severity index is rather thin. As mentioned above, inundation hotspots can be identified through vulnerability assessment.**

**The thresholds for the NFSI were not mentioned. What insights or new implications can be extracted from using the NFSI?**

We agree that the model set-up, data processing and boundary condition implementation seem intuitive. This, however, does not influence the overarching goal of this manuscript, which is to investigate the usability and reliability of hydro-numerical models that are built exclusively on open-access data. These models have the potential to offer preliminary, low-cost and low-effort flood hazard assessment in any flood risk analysis, especially in urban agglomerations in the developing world, where data is scarce and modeling expertise may be limited. Moreover, we are convinced that one needs to differentiate between a hazard and a vulnerability assessment, the latter of which requires gathering a substantial amount of socio-economic data and inputs. These inputs are not directly part of a numerical model and a flood hazard assessment. Our proposed methodology works to determine flood hazard, for which we combined two major components (flood depth and flood duration) through the proposed NFSI to better pinpoint areas where the level of hazard, in contrast to (socio-economic) vulnerability, is at its highest. Our discussed methodology does indeed not substitute but rather complement a vulnerability assessment and can be seen as a valuable contribution to flood risk assessment. This is achieved by identifying the locations or districts that are particularly exposed to flood hazard. This, in turn, allows vulnerability researchers to quickly identify areas of high concern, where more focused attention, awareness raising or training programs are required. The major advantage of the proposed methodology is its applicability and replicability since it does not rely on locally sourced and processed data. Nevertheless, we have revised the manuscript to include this explanation and thereby reduce any uncertainty in regards to the added value of flood hazard assessment through the NFSI.

**Part III:**

**How can the data processing method be applied to other megacities? Why the authors selected HCMC for model validation? Why not different locations around the planet?**

This question is praiseworthy as it aligns very well with our overarching goals. Ultimately, the practical objective of the presented methodology is to allow researchers to build low-cost, low-effort and fully transparent hydro-numerical models for any parts of the globe, especially for those locations where data accessibility, availability or both is lacking. Our methodology is unique and easily applicable on other coastal megacities that are particularly at risk from increasing flood severity due to climate change, relative SLR or other drivers and processes. Furthermore, a

pronounced focus was laid on the relative changes of flood hazard due to climate change in the revised manuscript, which should also alleviate calibration and validation concerns, given that the relative changes are of interest. The case of HCMC was chosen since this city epitomizes the complex interplay of the aforementioned disaster risk components (see e.g. Kreibich et al, 2022) in an environment where accessibility to official data is limited.

**Specific comments:**

1- **Line 24: adaptation to what, increasing precipitation? Sea level rise? Usually, adaptation refers to responses to changing risk. I don't think it is applicable to this manuscript. In this case it is more like responding to floods.**

We thank Reviewer #2 for this comment. Ultimately, there are three main factors that control severity and extent of urban floods in regards to HCMC, namely extreme river discharge, heavy precipitation and storm surges. These drivers already pose a great problem for the inhabitants of Ho Chi Minh City, especially the combination of heavy rain and high tidal water levels that hampers the effectiveness of drainage, which will require adaptation in the near future. Besides the major threat of high intensity, low frequency floods that cause significant material and human damage, additional attention needs to be given to more frequent floods (with lower intensity) that cause significant socio-economic disruptions (ADB, 2010). This problem is projected to increase in the future due to changing hydro-meteorological conditions that are particularly troublesome for a flat, low-lying area such as HCMC. Even though precipitation is not projected to increase in intensity, relative sea level rise (combination of secular sea level rise and land subsidence) will make such precipitation more problematic due to an increase in the backwater effects, rendering stormwater drainage systems in the city useless. Through the proposed NFSI, critical flooding hotspots, that are controlled and dependent on the flood drivers can be identified. A more focused numerical investigation of the effectiveness of such future adaptation options can be found in Scheiber et al. (Preprint).

2- **Line 55 – 59: why there is a need for the complete surface runoff model while vulnerability assessment is being conducted? What is the application of the proposed flood severity index?**

As outlined in a previous response, risk is composed of three factors according to the IPCC (SREX, 2012; SROCC, 2019) namely a combination of hazard, exposure and vulnerability. The

investigation of hazard is therefore one of three equivalent contributions when quantifying risk. It is true that some studies rely on topographic data alone to determine whether certain areas will be flooded in the future due to sea level rise, but such analysis does not capture the complex dynamics of flooding which can only be done through the use of hydro-numerical models. The proposed NFSI helps in giving a clearer, more concentrated picture of the spatial distribution of hazard and its local hotspots by combining two critical components, the flood depth and the flood duration.

3- **Line 66: how about flood frequency? Why is flood frequency excluded from this index?**

An integration of the frequency of exceedance of a certain flood level is definitely needed when quantifying flood damage and losses. Nevertheless, our hydro-numerical model provides an estimated distribution of flood depths and its associated duration due to set boundary conditions, i.e. hydro-meteorological time series. The frequency of occurrence of these boundary conditions is defined by their return period which was set to 1 year our study. It is certainly interesting to combine different events with different return periods and investigate the ensuing losses. This, however, would go beyond the scope of this particular paper since our main objective is to elucidate the level of hazard and extent of exposure in certain urban areas through the NFSI.

4- **Table 1: this table can be improved by incorporating errors or each DEM, and how these errors can be addressed.**

We find this comment very valuable and revised the table so that these errors are now included.

5- **Figure 2: if my understanding is right, it should be: subtract (c) from (b) rather than add (b) to (c).**

Thank you very much for revealing this mistake of ours. We appreciate the alertness and corrected the graph accordingly.

6- **Line 172 – 174: how about natural waterways inside HCMC? There is informal settlement encroaching on natural waterways inside HCMC, which also get flooded frequently during high tides and heavy rains (ex. Tran Xuan Soan street).**

Thank you for this valuable comment. There are indeed both natural and man-made waterways inside Ho Chi Minh City and both of them were considered in our model. We edited the text to make this clearer to our readers.

7- **Line 231: why not using data from Phu An station?**

We intensely discussed this question among the co-authors. In summary, there are two reasons for not using data from Phu An. First of all, we chose to build an expanded model of the HCMC area with a southern boundary at Nha Be. As a consequence, the tidal boundary condition needs to be applied at that location, which only allows the use of the Nha Be tidal levels. The second reason is the apparent lack of open-access data for the Phu An station, which makes it impossible to determine the water levels there using our methodology.

8- **Line 244: why an eight-day time series?**

An eight-day time series was chosen for two purposes. First of all, a certain run-up time needs to be considered in the hydro-numerical model in order to allow the stabilization of water levels. Furthermore, adequate time needs to be given after precipitation occurs in order for rain runoff to properly route and concentrate within the model. Thanks to your comment, we have added these details to our text for better clarity.

9- **Line 261: why only a 2-year flood selected? How about 5-year, 10-year floods?**

The 1-year event was adopted here in order to highlight the substantial divergence in the literature in regards to the calculation of the precipitation depth. Furthermore, our focus was on lower intensity, higher frequency rain events that regularly cause sustained socio-economic disruption in the city (ADB, 2010), for which the model was calibrated and validated. Lower frequency events with certainly larger flood extents and depths could also be easily simulated, but these scenarios of flooding would go beyond the scope of our study.

10- **Comment for the 2.2. section: How about reservoirs and groundwater? Why are these excluded from the model?**

The impact of reservoirs and groundwater is indeed a considerable aspect of hydrological modeling. However, for the present setting, we have assumed that groundwater is saturated during the rainy season, and thus no infiltration occurs, which particularly holds true when

considering the significant percentage of impermeable surfaces in the urban areas of HCMC. In regards to the upstream reservoirs, no positive effect on their potential in reducing inundation depth was considered, which aligns with a conservative engineering approach.

11- **Method section: there should be one graph summarize the proposed methodology. So far, these are scatter over different section of each type of date, which is difficult to grasp the bif picture of the proposed methodology. Besides, methodology for processing each element should be presented in equation form rather than figures (i.e., figure 2 and figure 7).**

Thank you for this comment. We made sure that additional processing steps are integrated in Figure 1 to make the whole process more accessible to a wide audience. It is true that equations could be used instead of Figure 2 and 7, but we would prefer to visualize the results of the processing and include the equations in the illustration, as it allows the reader to intuitively grasp what happens at each step of the work-flow.

12- **Line 335: what do the authors mean by the threshold of the NFSI is at its maximum? Maximum of what? How are the factors of changing climate considered? Why did the authors give equal weights to flood depth and duration? How about flood frequency?**

We would like to clarify that it is not the threshold of the NFSI that is at its maximum but rather the combination of flood depth and duration over the threshold of 10 cm are at their maximum. For climate change considerations, the NFSI needs to be computed for that particular case and then normalized according to the base case where no climate change effects are considered. Regarding the equal weight, this was done in order to ensure that the results are not biased, especially considering the lack of additional data that are necessary to determine whether flood depth or duration play a bigger role in damage for a particular location. This weighting can be different depending on the case and the local composition of flood damage. Future users are, of course, free to change the weighting and adapt it to a specific use case. Flood frequency can surely be considered when changing the boundary conditions of the model depending on the investigated return period. To that end, the IDF curves presented in Fig. 4 offer a valuable starting point to estimate precipitation depth for HCMC.

**References:**

Asian Development Bank, (2010). Ho Chi Minh City Adaptation to Climate Change. Manilla, Phillipines.

Kreibich, H., Van Loon, A. F., Schröter, K., Ward, P. J., Mazzoleni, M., Sairam, N., ... & Di Baldassarre, G. (2022). The challenge of unprecedented floods and droughts in risk management. *Nature*, *608*(7921), 80-86.

IPCC Managing the Risks of Extreme Events and Disasters to Advance Climate Change Adaptation (eds Field, C. B. et al.) (Cambridge Univ. Press, 2012).

IPCC, *Special Report on the Ocean and Cryosphere in a Changing Climate (SROCC)* (2019)

---

## Author Response (AR1)

GENERAL RESPONSE

To whom it may concern,

We highly appreciate and are very thankful for the time and effort that was invested in reviewing our manuscript. Thank you for initiating this fruitful discussion. After carefully studying the constructive queries and comments, and following in-depth discussions among the co-authors, we have thoroughly revised our manuscript. Most importantly, we focused on refining our key motivation and messages: (1) to derive a simple but reliable methodology to set up a numerical model, which makes best use of open access (geo)data and (2) to introduce a new and easy-to-apply flood severity index. This preliminary risk assessment supports the overarching goal of localizing urban inundation hotspots that subsequently require special attention. Please find our responses (blue) and revised text blocks (*blue, italic*) below each review comment (**black, bold**).

With kind regards

Leon Scheiber, Mazen Hoballah Jalloul, Christian Jordan, Jan Visscher, Hong Quan Nguyen and Torsten Schlurmann
* * *
**RESPONSE TO REVIEW #1**

**General Comments:**

**Part I:**

**The authors present some kind of methodology to build a 2D hydraulic model based on freely available data. The objective is praiseworthy but since these data appears to be of relatively low quality and having in mind the sensitivity of a hydraulic model to the DEM for flood simulation, it seems pointless, even dangerous. Indeed, an urban flood model have to be of high quality (DEM, hydraulic calibration and validation) having in mind the repercussion of modeling results. The authors present an interesting discussion on the DEM uncertainties based on freely available data. They should try to propagate these uncertainties using the numerical model; it could lead to any kind of results. Most institutions or insurance companies will use flood maps provided by scientists as a truth. If the model is not properly calibrated nor validated, it may lead to very problematic**

**situation for people living in these areas. Clearly here, for the case of the River Sai Gon next to Ho-Chi-Minh-City (very flat system largely influenced by tide, complex system of canals, heavy rains, etc.), a numerical model of the city needs data of much better quality for the construction and validation of the model. Is it reasonable to have a DEM resolution of 30 m or more with a vertical uncertainty up to 1 m to build a 2D numerical model? Eventually, the proposed model is not really calibrated nor validated.**

We are thankful for receiving this constructive feedback and are reassured in our motivation to disseminate our findings, given that the reviewer sees the objective as praiseworthy, too. This perspective confirms the added value of communicating the presented methodology for building urban surface runoff models based on open-access data to a wider audience. Upon carefully examining this general comment, we came to the conclusion that the purpose of our methodology was not communicated as clearly as intended. To clarify our overarching motivation extensive changes were made to the manuscript starting off with the title which now reads: *"The Potential of Open-access Data for Flood Estimations: Uncovering Inundation Hotspots in Ho Chi Minh City, Vietnam, through a Normalized Flood Severity Index"*

This should reduce the misleading impression that open-access models could be the "be-all and end-all" instrument for producing highly accurate flood maps. In contrast, the title now emphasizes the inherent uncertainty and limitations introduced by using open-access data for this purpose. Furthermore, the abstract was modified to better reflect the main objective of the presented work and now reads as follows (ll. 21-30):

*" (…) To help alleviate this problem, this paper explores the usability and reliability of flood models built on open-access data in regions where highly-resolved (geo)data about the local topography, bathymetry or hydrology are either unavailable or difficult to access, yet evaluation of risk from flooding is crucial. To that end, the example of Ho Chi Minh City (HCMC), Vietnam, is taken to describe a comprehensive, but generic methodology for obtaining, processing and applying the required open-access data. The overarching goal of this study is to produce preliminary flood maps that provide first insights into potential flooding hotspots demanding closer attention in subsequent, more detailed flood risk analyses. As a key novelty, a normalized flood severity index (INFS), which combines flood depth and flood duration, is proposed to deliver key information in preliminary flood hazard assessments. This index serves as an indicator that further narrows down the focus to areas where flood hazard is significant. (…)"*

Furthermore, the wording of the introduction has also been modified to better articulate our intentions regarding the cost and time intensive character of data acquisition and processing of on-site procured high resolution data. It now reads as follows (ll.54-57; 71-76; 83-86; 89-91; 98-105):

*"(…) which complicates numerical studies, especially for independent parties. Not only are these data sets prohibitively costly, but they also often lack the required spatial and temporal coverage needed for proper derivation of boundary conditions and model set-up. Furthermore, it is often the case that such data are badly described and lack the necessary meta-data."*

*"(…) Even though such open-access data cannot always be the basis for flood maps that can be considered as truth (especially when validation data is lacking), their potential usefulness should not be overlooked. Especially, when the overarching goal is to improve system understanding (i.e. knowledge about the causalities between drivers and resulting impacts), generating flood estimation maps can open up opportunities to gain insights for subsequent decision-making processes regarding more detailed modelling for critical areas."*

*"(…) With the overarching goal of providing a methodology for researchers to build low-cost, low-effort and fully transparent hydro-numerical models for any part of the globe, where either data is scarce or capacities and competence are limited, this manuscript investigates the usability and reliability of hydro-numerical models that are built exclusively on open-access data."*

*"Such low-cost, low-effort models are ideal for preliminary food hazard assessment in any flood risk analysis, especially in rapidly developing urban agglomerations where data are scarce and modeling expertise is often limited."*

*"(…) The presented methodology can be seen as an orientation for city planners and authorities from data-scarce regions, helping them to readily estimate where inundation hotspots with particularly high damage potential are located in a first flood hazard assessment. It allows them to focus, subsequently, on building more detailed damage models for the most heavily exposed city districts. Such detailed damage models usually require more extensive and expensive data collection (e.g. detailed topography, detailed time series for certain flood events, drainage networks, flood protection systems, land use, socio-economic vulnerability, etc.) and are indispensable for quantifying risk as a function of hazard, exposure and vulnerability. The methodology proposed in the following is especially beneficial in those situations, where such highly resolved data isare (still) missing, inaccessible or require significant resources."*

These amendments to the manuscript should make it clearer for any reader that the open-access data flood model of HCMC does not promise to deliver results that can be considered as truth, but rather initial estimations that open up opportunities to gain insights for subsequent decision-making processes regarding more detailed modeling for critical areas. The presented generic methodology can also be seen as an orientation for city planners and authorities from the developing world, helping them to readily estimate, where hotspots with particularly high damage potential are located, in a first stage of flood risk assessment. Furthermore, it is not uncommon to find regional flood studies of HCMC that rely on coarse terrain data. For example, Scussolini et al. (2017) used a terrain mesh that ranged from 100 m to 500 m for their regional flood model, while Duy et al. (2019) relied on a 1-D model with 1000 data points on a 15 m grid for the river network and 28600 points on a 15 m grid for built-up areas. Undoubtedly, progress has been made in flood modeling in recent years, yet highly resolved (geo)data is neither readily available nor always accessible to independent users. The presented manuscript deals with those situations, where highly resolved data is missing or inaccessible.

Last but not least, we would like to address the impression of the reviewer that the model was not properly calibrated nor validated. In our opinion, this comment is not justified but we acknowledge the lack of emphasis in the presentation of calibration results and therefore added an additional table to section 2.1.3 to avoid any misunderstanding (Tab. 3). The table shows that a Manning friction coefficient of 0.10 $s/m^{1/3}$ does indeed provide the best results for all three statistical parameters. Albeit far from perfect agreement, an NSE value of 0.5 to 0.64 is satisfactory for first flood estimates and is sufficient for a model whose goal is to determine inundation hotspots rather than quantitatively predict flood depths. This is especially valid when compared to the flood model by Le Binh et al. (2019) which relied on non-open-access 2 m resolution LiDAR data and still achieved NSE values of 0.51 to 0.89. Moreover, our method of calibration is particularly suited for rain events as it relies on flood depths measured within the city during heavy rain events and not on discharge and tidal gauges that are remote from the affected urban areas. Considering our results against the backdrop of comparable models, we are confident about the robustness of the presented approach.

**Part II:**
**Results in Fig. 5 are correct but there are many unclear assumptions behind. And main results presented in Fig. 7 are quite poor.**

We have made special efforts to revise Figure 5 (now Fig. 7) so that the assumptions are clearer. Furthermore, a table was composed, where flood depths observed at scattered locations and their corresponding simulated flood depths can be compared. This table, in combination with the corresponding geolocations, is included in the Supplementary Material (cf. Tab. 3.1). The locations are numbered and further explained by street names that are depicted on a map.

In regards to Figure 7 (now Fig. 9), it is worth noting that this figure is meant to exemplify the application of the $I_{NFS}$ by highlighting the differences between the maximum flood depth, inundation duration over threshold and their combination in the form of the $I_{NFS}$. Furthermore, this figure is best understood when simultaneously looking at the results in Table 4, which clearly highlight how accurate and trustworthy the $I_{NFS}$ was in covering the locations of reported inundation as opposed to the other two flood indicators. Nevertheless, we revised the text to better define the goal of this section and edited Figure 7 (now Fig. 9), where the spatial overlapping of reported inundations and the $I_{NFS}$ is clearer.

**Part III:**

**In general, although the manuscript is well written, many technical details are missing. It is often difficult to understand how the bathymetry and boundary conditions are built.**

Regarding the bathymetry, section 2.1.2 entitled "Bathymetric Data" is completely dedicated to explaining and discussing how the bathymetric data was acquired and integrated into our model. Admittedly, this methodology was proposed mainly because of the lack of comprehensive bathymetric data for the model area even after consultation with local partners. Unfortunately, it is not always the case that local institutions or authorities have knowledge about or are mandated to grant access to available geodata. Nevertheless, we developed an additional figure for the bathymetry (Fig. 3) that complements our original explanations. Furthermore, a figure for the derivation of the tidal boundary condition was added to this section (Fig. 5).

**Part IV:**

**The authors introduce a new index to evaluate the flood risk (normalized flood severity index), which can be interesting. However, they should verify if the normalization with a maximum value cannot bias the result in case of numerical divergence. Also, since the results of the model are quite poor, it appears difficult to validate the use of the index here. The index should be discussed for a case, which is much better described and a numerical model that is of higher quality.**

We are very pleased that the reviewer regards the original idea and derived concept of the $I_{NFS}$ as valuable, given that its proof-of-concept was one of the primary motivations for submitting this manuscript. Yet, we acknowledge the concerns regarding a bias due to false maximum values in case of numerical divergence. In fact, we thoroughly examined the simulation results in order to exclude any divergence, artifacts or outliers, which in our case were not found. Nevertheless, based on this comment, we decided to increase the robustness of the $I_{NFS}$ against divergence and outliers by relying on quantiles of flood depth and duration for normalization. Through this method, the maximum flood depth is capped to the 95th quantile, keeping the value of the $I_{NFS}$ between 0 and 100, while eliminating potential artifacts due to numerical divergence. The respective paragraph now reads as follows (ll. 435-439):

*"(…) In order to increase the robustness of the dimensionless $I_{NFS}$ against numerical divergence and artifacts, the normalization is based on the 95$^{th}$ (spatial) percentile of flood depth and duration. Depending on the specific case, however, this reference for normalization may be adjusted. The $I_{NFS}$ at each grid cell (x,y) can be expressed as follows:*

$$I_{NFS}(x,y)(\%) = \frac{d_{max}(x,y) * T_{d>10cm}(x,y)}{d_{max,95\%}(x,y) * T_{d>10cm,95\%}(x,y)} * 100 \qquad (3)$$

**Part V:**

**Flood hazard assessment of pedestrian often combine water depth and flow velocity (Musolino et al., 2020). Since this criteria is based on results from a 2D model, it could be interesting to introduce a second index based on velocity and duration. Anyway, this part of the paper appears a little bit off-topic.**

The combination of water depth and velocity is definitely an interesting prospect to determine risk to pedestrians in specific urban environments that needs to be examined in more detail. The decision to neglect the velocity component from integration into the index here was based on its negligible impact on flood damage modeling attempts in low-elevation coastal zones (LECZ) (Amadio et al., 2019; Wagenaar et al., 2017; Kreibich et al., 2009). Since the urban or rural terrain in our focus area is rather flat and expected flow velocities are too small to pose a realistic risk to humans, it was decided to put economic damage rather than pedestrian casualties into focus. This is accompanied by the relatively high economic damages from less severe but much more frequent urban floods that Ho Chi Minh City regularly suffers from (ADB, 2010). Furthermore, the nature of the presented surface runoff model, where barriers such as buildings and vegetation

cannot be easily represented, does not allow for the computation of exact peak flow velocities due to changes in cross-section. The proposed consideration of velocities could be useful in a more detailed model for certain areas or districts where a surface elevation model with a fine resolution (5 m or lower) can be built. Through our methodology, specifically these areas of greater risk (hotspots) can be identified and more detailed simulations be conducted. In response to this comment, we added a full paragraph to the discussion section (ll. 617-626).

*"(…) One limitation of the INFS can be seen in the exclusion of flow velocity, which was shown to play a significant role in pedestrian casualties (Musolino et al., 2020). However, quantifying this component can only be done through highly resolved flood models for particular city districts where flow obstacles can be accurately represented. Furthermore, flow velocity demonstrably plays a secondary role in LECZs where urban or rural terrain is rather flat (Wagenaar et al., 2017; Amadio et al., 2019). In such settings, the impact of flow velocity is rather small when compared to those of flood depth and duration, particularly for estimating monetary loss (Kreibich et al., 2009), and even more so in the rainfall-runoff scheme presented here. Nevertheless, through the proposed methodology, open-access data can be leveraged to determine urban areas with high damage potential where the procurement of highly resolved data for a more detailed flood model is required. In these highly resolved models, even flow velocity can be quantified to determine the associated risk to pedestrians."*

**Minor Comments:**

**1-      L32: For a list of reference, use the chronological order**

Thank you for this comment. References with multiple entries were changed accordingly.

**2-      L34: next decades, L40: skil "C.R", Fig 2. Use (a), (b), etc. Instead of (A), (B): 3×3 instead of 3x3 (times and not x-letter)**

Thank you for these corrections, the text was changed accordingly.

**3-      L75 (Figure 1) I do not see any step of calibration and validation of the model**

This is a valid point and we amended said figure accordingly.

**4-      L79: What do you mean by "similar sources"?**

We have taken this comment into consideration and changed the wording of this sentence that

now reads as follows (ll. 114-117):

*"Generally, the search priority of terrain data, as well as hydro-meteorological data, follows the same path, with official sources at the top, followed by global repositories, peer reviewed literature, grey literature (i.e. publicly available reports and assessments), and finally regional and global models."*

**5-      L137: Please detail the characteristics of the LiDAR data**

An additional sentence now refers interested readers to the Supplemental Material which comprises detailed information about the LiDAR locations and characteristics.

**6-      Tab. 2: An error of one meter for a DEM is huge! How accurate can you be for hydrodynamic calculations?**

We agree with the reviewer that differences of one meter are significant for a DEM, which is why special emphasis was put on the discussion of the differences in Section 4. However, the use of difference plots as described in 2.3.1 alleviates this inherent epistemic uncertainty, which is confirmed by the model calibration and validation. Furthermore, it is important to measure the amplitude of the bias of the proposed DEM with regards to other open-access DEMs (SRTM, ALOS, ASTER, COPERNICUS). The positive bias of these traditional satellite DEMs can reach up to 13 m in comparison with the LiDAR data samples, rendering them completely unreliable for flood modeling purposes. This corroborates the conclusion made by Hawker et al. (2018) in regards to the usability of the existing global DEMs. In this regard, the proposed DEM of this manuscript is far more reliable than any other open-access DEM and can confidently be used in the intended preliminary flood estimations. To corroborate this, a new comparison was made, whose results now replace Table 2, with a larger LiDAR data set, showing significant improvement over SRTM and some improvement over CoastalDEMv1. The locations of these LiDAR data sets were added to the Supplementary Material of this article.

**7-      In many countries such as in Vietnam, bathymetric data exist and could be obtained through collaborations or by paying for it**

Thank you for this comment. According to our knowledge and local networks, no open-access data exists for the Sai Gon River, while open-access bathymetric data for the Dong Nai River stems from US Army Corps of Engineers maps created in 1965 (Gugliotta et al., 2019). It is also correct that bathymetric data is available but in a purely commercial framework. However, it is

mostly provided in deep sections (e.g. river mouths) for transportation purposes and in hard-copy only. Accordingly, access and use of data from HCMC, if available / affordable, would be limited by commercial interests. This fact again underlines the need for the utilization of open-access data in flood modeling, which is the overarching objective of our manuscript.

**8-      L158: Again, such data base provides very rough estimations of the bathymetry. How accurate will be the model using such data?**

We are pleased that the reviewer raises this point. This is exactly why we did the sensitivity analysis, whose results are presented in Section 3.2, showing that even a depth change of +80% of the river bed influences urban flood depths by only a few centimeters (7 to 12 cm).

**9-      L172: What is the reference here? How do you set the bed level of the canal? Is this average depth a tidal-average depth?**

Thank you for pointing at this lack of clarity. The canal depths are given relative to mean sea level. This detail was incorporated in the revised manuscript, i.e. on various occasions in Section 2.1.2.

**10-     L180: "expedient" is maybe a little bit strong. For the moment, the model construction seems very crude, especially for a complex and very flat system such as the Ho Chi Minh City Area**

Thank you for this comment. We agree with the reviewer's opinion on the wording and have omitted the word "expedient" from line 180.

**11-     Fig. 3: Please provide a proper figure caption and not a discussion of the figure. Also, most of the legend has no clear meaning (i.e. difference, exemplary colours, etc. ?). What do A, B and C red squares mean? I guess they correspond to the LiDAR samples**

Thank you for pointing out these mistakes and unclarities. We fixed Figure 3 (now Fig. 4) according to your comments to enhance its readability.

**12-     L190: I'm not sure I understood. Are buildings represented as non-flowing area? Or is an equivalent Manning friction coefficient used to represent build effects on the average flow velocity?**

We agree that this sentence might have caused misinterpretations. Therefore, the paragraph was adjusted and now reads as follows (ll. 255-260):

*"Buildings and extensive vegetation that significantly reduce the available cross-section for water routing are not represented as no-flow areas in the final DEM due to the 1 arc second resolution. Instead, an equivalent Manning friction coefficient was considered in the simulated hydraulic roughness, representing an additional macro-roughness effect that would be neglected if set to the value of, for example, concrete (Chen et al., 2012; Taubenböck et al., 2009; Vojinovic and Tutulic, 2009)."*

**13-   L195: The Manning coefficient has a unit; don't use the term "roughness coefficient" while talking about the Manning friction coefficient**

Thank you for this valuable comment, the text was adjusted accordingly by substituting the term "roughness coefficient" with "Manning coefficient" and by adding its corresponding unit.

**14-   L197: is a unique roughness coefficient used for the whole model? (n=0.1 s/m$^{1/3}$)? What about canals and main channels (Sai Gon and Dong Nai Rivers)?**

We are aware that the wording might have caused misinterpretation in reference to the application of Manning friction coefficients. Therefore, the wording of this sentence was changed and now reads as follows (ll. 274-276):

*"Following this approach, the best results for the RMSE, NSE and PBIAS are obtained for a Manning friction coefficient of 0.10 s/m$^{1/3}$, which corresponds to the higher bound of the proposed range for mimicking urban settings (Schlurmann et al., 2010). (…)"*

**15-   L222: The Sai Gon water discharge is mostly influenced by tide (Camenen et al., 2021). They provide some estimation of the net discharge for years 2017-2018**

Thank you for sharing this reference. We have examined the given net discharges of 30 and 65 m$^3$/s for the years 2017 and 2018, respectively, and are satisfied that our chosen net discharge of 54 m$^3$/s, which is the long-term net discharge according to Tran Ngoc et al. (2016), falls within this range. Nevertheless, we now mention these additional values in our manuscript as part of Section 2.2.1 (cf. ll. 303-304):

*" (…) with the long-term mean river discharge of the Sai Gon River corresponding well to the net discharge of 30 and 65 m$^3$/s for 2017 and 2018 calculated by Camenen et al. (2021).*

**16-     L238: So, as far as I understood, you had access to Nha Be data**

This is correct. We had access to both tidal water level data for Nha Be and rainfall data for the Tan Son Hoa rain station from our local partner. We used this data to critically evaluate the reliability of the open-access data and have specified the results of the comparison for both tidal water levels and rainfall data in Section 2.2.2 and 2.2.3.

**17-     L239: It would be interesting to present a plot showing these results**

We found this idea very interesting as well and developed a graph that summarizes these results. Even though parts of this information are inherently contained in the publication by Gugliotta et al. (2019) we have incorporated an additional figure (Fig. 5) in Section 2.2.2 of the manuscript.

**18-     L255: variables in italic: *n*=28, Eq. 1; functions in roman: *n*=28; define all variables introduced in this equation, L263: The variable *n* is already introduced for a number of years, Eq. 2: this is not an equation; to be written within the text**

Thank you for these comments, we have corrected the manuscript accordingly.

**19-     L260: α?**

The value of α serves to indicate the goodness of fit of a theoretical distribution to the actual data as developed and discussed by Dyck (1980). This value should show that the Gumbel distribution delivers a good fit for the rain data set made available by NOAA for the Tan Son Hoa rain station.

**20-     L269: Do you mean *ß*=0.854 for the Ho Chi Minh City area?**

Correct. We have added the reference Ho Chi Minh City to avoid confusion.

**21-     L294: Arguable**

We understand the reviewer's concern regarding our conclusion. The question to be asked here is whether these derived boundary conditions are of sufficient quality to be used for the generation of qualitative flood risk estimates (as opposed to quantitative and highly-detailed prediction). The comparison between the open-access rainfall data and the rainfall data provided by local partners for Tan Son Hoa shows that there are indeed notable differences as can be seen from Table 3. However, these differences are reasonable especially for return periods of 5 years and less, which are the focus of this study.

**22-     L304: this is not a proper argument. If there is some protection measure, there won't be any flow toward some of the lowest elevations. These zones may be eventually flooded but for other reasons (rain, groundwater, etc.) and so with a different dynamic.**

We share the reviewer's concern regarding this point. However, the argument greatly depends on the nature of the protection measure being approximated. Large-scale, above-ground measures (e.g., dikes, pumping stations, detention/retention ponds) could still be accurately represented in this DEM, especially in the case where the impact of adaptation measures is evaluated relative to a no-adaption base case (see companion paper by Scheiber et al. (in review)). In contrast, underground stormwater drainage systems are harder to represent, especially when relying on open-access data. However, there is significant evidence for the ineffectiveness of the storm water drainage system in the case of Ho Chi Minh City (Le Dung et al., 2021; Nguyen, 2016), so that a worst-case scenario modeling is conceivable (Scussolini et al., 2017). The local drainage system is not well maintained and it has limited functionality, so that its capacity is strongly diminished during heavy rain events (Nguyen et al., 2019). This warrants a conservative engineering approach, in which they are deliberately omitted in the modelling scheme. Based on this comment, we added the following clarifications (ll. 394-403):

*"Furthermore, there is significant evidence for the ineffectiveness of the stormwater drainage system in the particular case of HCMC (Le Dung et al., 2021; Nguyen, 2016). The local drainage system is not well maintained and has limited functionality (Nguyen et al., 2019). Drainage capacity is therefore heavily reduced in case of storm events, which justifies its exclusion from the model representing a conservative approach."*

*"In contrast, the absence of flood protection structures in the model has a significant impact on the run-off dynamics, whereby flooding can even occur in places where no inundation is plausible under normal conditions, i.e. no rain, mean tide and mean river flow. To counteract this effect, simulated water levels are corrected by taking the results of the regular conditions as a reference. This reference was defined based on flooding threshold values determined with local partners, information from grey literature like the JICA reports (JICA, 2001) as well as different media articles, whose URLs can be found in the Supplementary Material."*

**23-     L308: It would be interesting to present this reference. And this methodology is also arguable. If this reference is not realistic compared to observed flooded zones, how can we trust simulations with more extreme conditions?**

We understand the concern of the reviewer in regard to this chosen reference. This reference is indeed pivotal to the results of our flood simulation. If the reference, is not correctly defined, then the model would not perform as intended. In the case of our study, this reference is a theoretical inundation layer created by simulating mean tidal and fluvial conditions. Inundation by precipitation or more extreme tidal or fluvial conditions are only considered when they are above the aforementioned reference. As outlined above, it was defined based on flooding threshold values that were determined through joint work with local partners, information from grey literature like the JICA reports (JICA, 2001) as well as different media articles. Examples of such media reports are accessible at the following URLs:

https://www.c40.org/case-studies/mitigate-urban-flooding-in-ho-chi-minh-city-phase-1/
https://global.royalhaskoningdhv.com/projects/flood-management-in-ho-chi-minh-city-vietnam
https://borneobulletin.com.bn/poor-urban-development-cause-of-flooding-congestion-in-ho-chi-minh-say-experts/
https://e.vnexpress.net/news/news/major-anti-flooding-project-in-hcmc-misses-deadline-for-four-years-4444006.html
https://www.channelnewsasia.com/cnainsider/siege-climate-man-made-problems-sinking-ho-chi-minh-city-floods-2052231

The reference was thus set according to tidal time series for Nha Be presented in Figure 8, which represents the employed flooding threshold for Ho Chi Minh City. The confirmation of this reference method is further reinforced by the calibration and validation results.

**24-      L321: Flow depth is often not sufficient to evaluate risk for people. One also needs the flow velocity (Which can be provided by a 2D model of properly calibrated)**

The reviewer's argument is valid. 2D models that have a high spatial resolution to properly reflect changes in flow cross sections for urban environments can definitely deliver flow velocities relevant to evaluate risk for people. This, however, does neither address the idea nor follow the intended scope of our study, especially because our model is not appropriate for such a purpose and because high flow velocities evidently do not pose a significant threat in a relatively flat LECZ such as HCMC as explained in Section 4 (cf. ll. 618-627):

*"(…) One limitation of the INFS can be seen in the exclusion of flow velocity, which was shown to play a significant role in pedestrian casualties (Musolino et al., 2020). However, quantifying this component can only be done through highly resolved flood models for particular city districts, where flow obstacles can be accurately represented. Furthermore, flow velocity demonstrably plays a secondary role in low-elevation coastal elevation zones, where urban or rural terrain is rather flat (Wagenaar et al., 2017; Amadio et al., 2019). In such settings, the impact of flow velocity is rather small when compared to those of flood depth and duration, particularly for estimating*

*monetary loss (Kreibich et al., 2009), and even more so in the rainfall-runoff scheme presented here. Nevertheless, through the proposed methodology, open-access data can be leveraged to determine urban areas with high damage potential, where the procurement of highly resolved data for a more detailed flood model is required. In these highly resolved models, even flow velocity can be quantified to determine the associated risk to pedestrians."*

**25-     L334: This sentence should appear after the introduction of Eq. 3**

Thank you for this clarification, we have changed the text accordingly.

**26-     Eq. 3: even if this error is very common, it is not correct to introduce a variable made of multiple letters, i.e. NFSI = N×F×S×I. I would suggest to write:**

$$I_{NFS}(x,y) = \frac{z_{max}(x,y) \times D_o(x,y)}{max(z_{max}(x,y)) \times max(D_o(x,y))}$$

**Isn't it a problem to use the maximal flood depth and duration as a reference/ If the model provides some local unrealistic values for $z_{max}$ and or $D_o$, it would significantly affect the results.**

We find this comment very valuable and have implemented the suggested change. In regards to the effect of using the maximal flood depth and duration, we would like to refer to our answer to the general comments Part IV on the use of 95th quantiles (ll. 436-440):

*"(…) In order to increase the robustness of the dimensionless I_{NFS} against numerical divergence and artifacts, the normalization is suggested to be based on the 95th spatial percentile of flood depth and duration. The I_{NFS} at each grid cell (x,y) can thus be expressed as follows:"*

$$I_{NFS}(x,y)(\%) = \frac{d_{max}(x,y) * T_{d>10cm}(x,y)}{d_{max,95\%}(x,y) * T_{d>10cm,95\%}(x,y)} * 100 \tag{3}$$

**27-     L346: This is a significant issue. In many cases, institutions or insurance companies will use such flood maps as truth. If the model is not properly calibrated nor validated, it may lead to very problematic situation for people living in these areas.**

We agree with the reviewer's comment. A more sophisticated model to deliver highly accurate flood depths, durations and velocities that can be considered as truth is definitely desirable and

should be calibrated and operated on data with higher resolution and little uncertainty. However, as previously pointed out, the goal of this study is to assess the performance and consistency of open-access data (in cases, where that highly resolved information is inaccessible) in delivering first estimates of potential flooding as well as gaining understanding of underlying flood mechanisms. An open-access model built for regions where data is scarce can hardly be the "be-all and end-all" instrument for producing highly accurate flood maps but can still be very valuable for specific use cases, like determining inundation hotspots, especially when combined with the use of the $I_{NFS}$. We have undertaken major changes as elaborated under the General Comments section above. Yet, we agree that potential stakeholders should be cautioned in order not to overinterpret the presented methods and results. Without doubt, it should always be the goal to set-up and operate models that are as accurate as possible and, at best, based on highly-resolved (geo) data.

**28-    L352: What about calibration?**

To alleviate the concerns of the reviewer, we have revised our explanations regarding model calibration and added a table under Section 2.1.3 (Tab. 3). Furthermore, we changed the title of the section which now reads "*Hydraulic Roughness Coefficient and Model Calibration*".

**29-    L354: What about discharge and water level (tidal) conditions on the River Sai Gon?**

The reviewer's question surely warrants a closer examination of the discharge and tidal conditions of the River Sai Gon. However, there is a considerable lack of data in regards to this particular waterway especially in open-access data, so that we are afraid nothing can be done in this regard.

**30-    L357: Is this specific event representative of all events occurring on the HCMC area? Are there some cases with higher discharges for The River Sai Gon and/or strong tidal effects for which the model could also be validated?**

We have chosen the 14/06/2010 event for model validation, firstly, because the corresponding boundary conditions are known and, secondly, because flood depths were measured at different locations in HCMC during this event which is a pre-requisite for proper validation. Furthermore, our focus is on rain-induced flooding in Ho Chi Minh City and the special backwater effect caused by high tide, which is epitomized by this event (P=73 mm, WL=1.15 m). In regard to the discharges of the Sai Gon River, we have assumed that the reservoir mitigates higher discharge values associated with extreme river discharges. As for stronger tidal effects, such events can surely be

simulated, but were not the focus of our study, which addresses frequent and disruptive rather than extreme flood events.

**31-    Fig. 5: Do not add a linear regression when comparing simulation to observation; I see only 14 points on the plot whereas 25 are shown on the map. As far as I understood, the simulated water depths correspond to a difference between simulation results and results of the simulation for the 3h1y rain event with mean tide and mean river discharge. How sensitive are the results to this choice?**

We have taken the reviewer's comment regarding Figure 5 (now Fig. 7) into consideration and have added the results of a frequency analysis (Fig. 7c) as well as a table (Tab. 3.1) in the Supplemental Material. Here, the observed and simulated depths according to street name are given. And indeed, the simulated water depths are the difference between the results of the validation event (P=73 cm and HWL=1.15 m Q=54 m$^3$/s) on the day of the event (14/06/2010) along with mean river discharge and the results of the reference case which is characterized by mean tidal conditions and mean discharge, but no rain, for which no flooding occurs.

**32-    L363: Just to be sure I understood, you increased the Sai Gon bed level from +8.4 m (above sea level?!) to 14.8 m (Fig. 6). Is it realistic? Anyway, I'm amazed that such variations don't affect the results. How deep is the River Sai Gon for normal flows?**

We would like to thank the reviewer for this valuable comment. We have changed the text and Figure 6 (now Fig. 8) to make clear that the depth was varied from 8.4 to 14.8 m below mean sea level (MSL) in the context of our sensitivity analysis. This variation in depth of the Sai Gon River was done in order to determine the impact of this parameter on the simulated flood depth. We do not have robust information pertaining to the natural depth of the Sai Gon River. Used depths were solely derived based on the official navigational depths maintained for access to the ports along the river. The results of this sensitivity analysis show that a flood model can indeed be built on such an assumption without largely interfering with the robustness of the results when the goal is to generate flood risk estimates in an environment where data is scarce.

**33-    L:365: How were selected these three points?**

Point C was selected because it represents the outlet of the Ben Nghe canal, and Point B represents the intersection between the Ben Nghe and the Tau Hu canals, where frequent flooding occurs, while point A is a known inundation hotspot at the endpoint of the Tau Hu canal.

The rationale was to show the impact of the change in the depth of the Sai Gon on the maximum water levels at different distances upstream from the outlet of the Ben Nghe canal to the Sai Gon River. Point C is located directly at the Sai Gon River, Point B is located 3 km upstream from Point C, while Point A is located 13 km upstream from Point C. The respective sentence was further specified as follows (ll. 481-483):

*"Specifically, the simulated water surface levels increase at points A (inner-city low point that is a known flooding hotspot), B (canal intersection, where frequent flooding occurs), and C (outlet of the Ben Nghe canal) with increasing river bed elevation."*

**34-    Fig. 6: Define the location of the pints where sensitivity analysis is provided on the map Fig. 3 (use other letters since A, B, and C corresponds to other areas) and present plots only. Add a proper scale with axis legend for the three plots (or 4 if you include Nha Be water level time series)**

Thank you for these valuable comments, we revised Figure 6 (now Fig. 8) accordingly. Among other changes, we removed the locations A, B and C from the provided map.

**35-    L375: There is not Fig. 6a and b. If you're talking about the plots in Fig. 6, it is not clear for me how you evaluation mFD and DoT from these plots.**

Thank you for pointing out this mistake, Figure 7 (now Fig. 9) was actually meant here. These figures serve to show the difference in spatial extent as well as spatial variability of mFD and DoT. The main argument is that areas with high mFD do not necessarily translate to areas with high DoT and vice versa; Figure 9c exemplifies this finding.

**36-    L377: How do you explain this behaviour? Is it based on observations from the field or from the numerical results?**

This behavior can be explained by the fact that these areas have smaller catchments with lower concentration times and are thus drained more quickly. Furthermore, their proximity to the Sai Gon River also plays a role in the drainage behavior.

**37-    L380: What do you mean by "highlights previously hidden inundation hotspots"? Again, if the model is not really validated (at least not everywhere in the studied area), how sure are you about such results?**

This sentence was meant to set aside the inherent bias of the spatiality of the reported inundations. However, after carefully reviewing this paragraph, we have decided to omit this sentence. As you correctly point out, we cannot say for sure whether these results fully reflect the unquestionable truth for unvalidated areas but it would definitely be interesting to validate them accordingly. Unfortunately, no inundations were reported in the depicted areas.

**38- L381: "considerable spatial overlapping"! I'm not that enthusiastic. Most of the reported inundation points do not overlap with the zones with a NFSI>0! What about all the zones with a high NFSI value? I can understand there is also a bias in the reported inundation points but you cannot say here that results are good.**

We are aware that it is not very easy to determine the spatial overlapping using the color scheme presented in Figure 7 (now Fig. 9), which is exactly why we created Table 4, where it is obvious that the $I_{NFS}$ map is 4-times as accurate in matching the locations of reported floods when compared to a random area with the same size. Nevertheless, we agree with the reviewer in this regard and intensely discussed whether the results can be presented in a way that effective spatial overlapping becomes clearer in itself and adjusted the figure accordingly.

**39- Fig. 7c: It is not very consistent to compare the flood severity index with reported inundation. A reported inundation corresponds to a water depth; so, these points should be compared to the modelled maximum flood depth (Fig. 7a). Again, do not provide comments of the figure in the figure caption (redundant with the text)**

We understand the concern of the reviewer in this regard. However, the reported inundations that were used in Figure 7 (now Fig. 9) do not only correspond to places were high flood depths were recorded but also places that are persistently flooded over a relatively long time. What we intended to highlight here is that the $I_{NFS}$ is better at identifying these areas as opposed to the flood depth, which does indeed cover more reported inundation, but performs worse relative to size and thus does not serve the purpose of the methodology, which is to better pinpoint areas that require closer attention and where the procurement of higher quality data is worthwhile. We amended the respective paragraph as follows (ll. 503-510):

*"(…) In particular, the locations of reported inundations, where sustained flooding demonstrably occurred, and the $I_{NFS}$ heat map show considerable spatial overlapping. While the INFS itself only covers 19% of the total area of HMC, 73% of the reported inundations lie inside or within 100 m of the area highlighted by the $I_{NFS}$. These figures are opposed to 78% and 73% for the $d_{max}$ and*

*$T_{d>10cm}$, that cover 38% and 34% of the area, respectively (Table 4). The small spatial extent of the $I_{NFS}$ heat map, relative to the $d_{max}$ and $T_{d>10cm}$ maps, coupled with the relatively high coverage of reported flooding locations corroborates the usefulness of the proposed index in successfully localizing flooding hotspots and quantifying their spatial extents."*

**40-    L437: Due to the limitation of data to calibrated/validate the model, it is logical to use a single Manning friction coefficient for the whole domain. However, in reality, this coefficient should vary spatially depending on the city structure (presence of vegetation or not, porosity of the system, etc.)**

We agree with the reviewer on this point. The spatial variation of the Manning friction coefficient can be and actually was considered. We abandoned the idea for this paper because the land use classes were not available in open-access. One might argue that remote sensing could be used to determine land use classes and then deriving the Manning friction coefficient for certain parts of the city, but that would be out of scope.

**41-    L474: True but the velocity is important in term of flood hazard for pedestrian (Musolino et al., 2020)**

We considered this caveat in our text but as stated in the responses to Part IV of the general comments of the reviewer, our model is neither designed to forecast precise flow velocities nor do they play a significant role in risk assessments for flat, low-lying systems such as HCMC. Nevertheless, the whole topic was intensely discussed and incorporated in section 4.

**L479: True but you need a robust and well calibrated model**

We agree that a more robust model that is calibrated for more events can deliver more robust and trustworthy results, but we do not insist on this argumentation since the overall objective is led by another intention as previously pointed out in the answers to Part I of the general comments.

**42-    L491: I'm not sure such model can be used to simulate flood drivers, even partially.**

It is an important fact that the provided methodology was calibrated and validated for rain events. The aforementioned explanations regarding the scope and limitations of our modelling scheme, hopefully, add some context and robustness to the highlighted statement.

**43-    L528: Use European convention for dates: 12/06/2018**

Thank you for this and the following plethora of comments and corrections, which is very helpful to ensure the correctness of this paper. The invested efforts are much appreciated. The bibliography was changed accordingly.

**44-    L531: Use "doi:" instead of the full link "https/: doi.org/"**

Thank you for this suggestion. Due to this and one and the following comments, we ensured to apply the custom NHESS citation style to our complete list of references.

**45-    L541: De Andrés, M.; be homogeneous with journal title (abbreviated or not)**

See comment regarding NHESS citation style.

**46-    L546: Use capital letters for acronyms only, i.e. Bennghe Port Company Limited**

Adjusted accordingly.

**47-    L554: Initials for first names after the name**

See comment regarding NHESS citation style.

**48-    L557: Add all authors (instead of "et al."), initials of authors**

Information completed.

**49-    L567: reference?!**

Information completed.

**50-    L574: date, doi**

Already included.

**51-    L595: Use capital letters for acronyms only, i.e. Go Fair**

Adjusted accordingly.

**52-    L602: Skip "available at…"**

Skipped.

**53-    L608: Skip "available at…"**

Skipped.

**54-    L614: NGO?!**

Corrected.

**55-    L615: Journal?!**

Information completed.

**56-    L630: Explain the acronym JICA**

Information completed.

**57-    L638: de Moel, H.**

Corrected.

**58-    L672: Add all authors (instead of "et al."), initials of authors**

Information completed.

**59-    L685: Explain all acronyms**

Changed accordingly.

**60-    L689: Don't use capital letters for the title and journal (International Journal of Geomate), some co-authors are missing**

Well observed. Authors and title were changed accordingly.

**61-    L701: Skip "available at…"**

Skipped.

**62-     L716: Add all authors (instead of "et al."), initials of authors; Don't use capital letters for the journal name (?); Add (in Vietnamese)**

Completed and adjusted accordingly.

**63-     L726: Don't use capital letters for the author name**

Adjusted accordingly.

**64-     L740: Skip references in review**

Not recommended, because it is the companion paper which forms the methodological basis for this study.

**65-     L776: Don't use capital letters for the author name**

Adjusted accordingly.
* * *
**General comments:**

**Part I:**

**The paper demonstrated how to process and applying open-access data to an urban surface run-off model. The authors also combined flood depth and duration into a so-called normalized flood severity index (NFSI) to identify urban inundation hotspots. Overall, this paper might be useful in demonstrating how open access data can be processed into hydrological model. However, the methodology to achieve this need to be more systematically presented.**

We are very pleased to learn that the reviewer agrees with our core idea and the usefulness of our approach in demonstrating how open-access data can be processed and incorporated into a hydrological model. Thank you for making us aware that the presented methodology is not always perfectly clear to the reader. We tried hard to refine and clarify our methodology and objectives in the revised manuscript. The first step to remedy this was to add additional processing steps to the work flow presented in Figure 1 (e.g. validation/calibration, simplification of model inputs). Furthermore, more attention was given to the derivation of bathymetric data. In order to improve this aspect, we added a figure (Fig. 5) to better illustrate this process and visualize the derivation of the tidal boundary condition.

**Part II:**

**Whether this methodology can be considered novel or not is unclear as the process seemed quite intuitive. Besides, the applicability of the research is thin. Vulnerability assessment is being conducted in cities to identify the areas that need response measures. Moreover, the application of the flood severity index is rather thin. As mentioned above, inundation hotspots can be identified through vulnerability assessment. The thresholds for the NFSI were not mentioned. What insights or new implications can be extracted from using the NFSI?**

We agree that the model set-up, data processing and boundary condition implementation seem intuitive to an experienced scholar. This, however, does not influence the overarching goal of this manuscript, which is to investigate the usability and reliability of hydro-numerical models that are built exclusively on open-access data. These models have the potential to offer preliminary, low-cost and low-effort flood hazard assessment in any flood risk analysis, especially in urban

agglomerations in the developing world, where data is scarce and modeling expertise may be limited. Moreover, we are convinced that any disaster risk assessment should consider both the drivers and probability of occurrence of a (natural) hazard and the local vulnerability. While the former constraints can be studied from hydro-numerical simulations, an assessment of the latter requires collecting and analyzing a substantial amount of socio-economic data and inputs. These inputs are not directly part of a numerical model and a flood hazard assessment. Our generic methodology allows to determine the extents and effects of a flood hazard, for which we combined two major components (flood depth and flood duration) through the proposed $I_{NFS}$ to advance the knowledge on highly exposed urban areas as potential flooding hotspots in contrast to a (socio-economic) vulnerability assessment. Thus, our presented and discussed methodology determines one key factor of a flood risk assessment. This is achieved by identifying the locations or districts that are particularly exposed to a flood hazard. This, in turn, allows vulnerability researchers to quickly identify areas of high concern, where more focused attention, awareness raising or training programs are possibly required in order to empower the local community by means of strengthening its resilience and coping capacity. The major advantage of the proposed methodology is its applicability and replicability since it does not rely on locally sourced and processed data. Nevertheless, we have revised the manuscript to include this explanation and thereby reduce any uncertainty in regards to the added value of flood hazard assessment through the $I_{NFS}$ (cf. II. 24-35; 624-627).

*"To that end, the example of Ho Chi Minh City (HCMC), Vietnam, is taken to describe a comprehensive, but generic methodology for obtaining, processing and applying the required open-access data. The overarching goal of this study is to produce preliminary flood maps that provide first insights into potential flooding hotspots demanding closer attention in subsequent, more detailed flood risk analyses. As a key novelty, a normalized flood severity index ($I_{NFS}$), which combines flood depth and flood duration, is proposed to deliver key information in preliminary flood hazard assessments. This index serves as an indicator that further narrows down the focus to areas where flood hazard is significant. The approach is validated by a comparison with more than 300 locally reported flood samples, which correspond to $I_{NFS}$-based inundation hotspots in over 73% of all cases. These findings corroborate the high potential of open-access data in hydro-numerical modeling and the robustness of the proposed flood severity index, which may significantly enhance the interpretation and trustworthiness of risk assessments in the future. The proposed approach and developed indicators are generic and may be replicated and adopted in other coastal megacities around the globe."*

*"Nevertheless, through the proposed methodology, open-access data can be leveraged to determine urban areas with high damage potential where the procurement of highly resolved data for a more detailed flood model is required. In these highly resolved models, even flow velocity can be quantified to determine the associated risk to pedestrians."*

**Part III:**

**How can the data processing method be applied to other megacities? Why the authors selected HCMC for model validation? Why not different locations around the planet?**

This question is helpful as it aligns very well with our overarching goals. Ultimately, the practical objective of the presented methodology is to allow researchers to build low-cost, low-effort and fully transparent hydro-numerical models for any parts of the globe, especially for those locations where data accessibility and availability as well as capacities, budgets and probably even competences are lacking. Our methodology is unique and easily applicable to other coastal megacities that are particularly at risk from increasing flood severity due to climate change, relative SLR or other drivers and processes. Furthermore, a pronounced focus was laid on the relative changes of flood hazard due to climate change in the revised manuscript, which should also alleviate calibration and validation concerns, given that the relative changes are of interest. The case of HCMC was chosen since this city epitomizes the complex interplay of the aforementioned components in disaster risk assessments (see e.g. Kreibich et al, 2022) in an environment where accessibility to official data or capacities are limited. The following sentences from the manuscript may underline this rationale (II. 34-35; 80-82):

*"The proposed approach and developed indicators are generic and may be replicated and adopted in other coastal megacities around the globe."*

*"Studying the metropolitan area of Ho Chi Minh City (HCMC), Vietnam, a city that epitomizes the complex interplay of disaster risk components in an environment where accessibility to official data or capacities are limited (Kreibich et al., 2022) ..."*

**Specific comments:**

**1-      Line 24: adaptation to what, increasing precipitation? Sea level rise? Usually, adaptation refers to responses to changing risk. I don't think it is applicable to this manuscript. In this case it is more like responding to floods.**

We thank Reviewer #2 for this comment. Ultimately, there are three main factors that control severity and extent of urban floods in regards to HCMC, namely extreme river discharge, heavy precipitation and storm surges. These drivers already pose a great problem for today's inhabitants of HCMC, especially the combination of heavy rain and high tidal water levels that hampers the effectiveness of drainage, which will require adaptation in the near future. Besides the major threat of high intensity, low frequency floods that cause significant material and human damage, additional attention needs to be given to more frequent floods (with lower intensity) that cause significant socio-economic disruptions (ADB, 2010). This situation is projected to deteriorate in the future, indeed, due to changing hydro-meteorological conditions and SLR that are particularly detrimental for a flat, low-lying area such as HCMC. Even though precipitation is not projected to increase in intensity, relative sea level rise (combination of secular sea level rise and land subsidence) renders changing precipitation patterns even more problematic due to an increase in the backwater effects, leaving stormwater drainage systems in the city increasingly useless. Through the proposed $I_{NFS}$, critical flooding hotspots that are controlled and dependent on the flood drivers can easily, but robustly be identified. In addition, rapid urbanization increases the pressure on the local drainage system, which necessarily will require multi-faceted adaptation efforts in the near future. A more focused numerical investigation of the effectiveness of such future adaptation options can be found in a companion paper by Scheiber et al. (in review).

**2- Line 55 – 59: why there is a need for the complete surface runoff model while vulnerability assessment is being conducted? What is the application of the proposed flood severity index?**

As outlined in a previous response, risk is composed of three underlying factors according to the IPCC (SREX, 2013; SROCC, 2019), namely a combination of hazard, exposure and vulnerability. The quantification of drivers and processes inherently posing a hazard is therefore essential when quantifying disaster risk. It is true that some studies rely on topographic data alone in order to determine whether certain areas are or will be flood-prone in the future due to sea level rise. But such analyses do not capture the complex dynamics of flooding within an urban agglomeration which can only be done through the use of hydro-numerical models given that the physics of precipitation and run-off in interaction with tidal ranges and (extreme) sea levels during storm events control the dynamics of flood hazard. The following reference was added (ll. 49-51):

*"This knowledge can be gained and is typically enriched through the application of hydro-numerical models, which are increasingly becoming the preferred option for inundation mapping (Dasallas et al., 2022)."*

The proposed $I_{NFS}$ helps in giving a clearer, more concentrated picture of the spatial distribution of hazard and its local hotspots by combining two critical components, namely the flood depth and the flood duration.

**3-     Line 66: how about flood frequency? Why is flood frequency excluded from this index?**

An integration of the frequency of exceedance of a certain flood level is definitely needed when quantifying flood damage and losses. Nevertheless, our hydro-numerical model provides an estimated distribution of flood depths and its associated duration due to set boundary conditions, i.e. hydro-meteorological time series. The frequency of occurrence of these process-controlling boundary conditions is defined by their return period which was set to 1 year in the current study. It is certainly interesting to combine events with different return periods and investigate the ensuing losses as well as assessing different levels of exposure. This, however, would go beyond the scope of this particular paper since our main research question regards, how open-access data can be incorporated and employed in a hydrological model.  The overarching objective finally is to elucidate the hazard dynamics and extents of exposure in certain urban areas of HCMC by means of the suggested $I_{NFS}$.

**4-      Table 1: this table can be improved by incorporating errors or each DEM, and how these errors can be addressed.**

We value this comment and revised the table so that these errors are now included (cf. Tab. 1).

**5-      Figure 2: if my understanding is right, it should be: subtract (c) from (b) rather than add (b) to (c).**

Thank you very much for revealing this mistake of ours. We appreciate the alertness and corrected the graph accordingly.

**6-    Line 172 – 174: how about natural waterways inside HCMC? There is informal settlement encroaching on natural waterways inside HCMC, which also get flooded frequently during high tides and heavy rains (ex. Tran Xuan Soan street).**

Thank you for this valuable comment. There are indeed both natural and man-made waterways inside the city limits of Ho Chi Minh City and both of them were considered in our model. We edited the text to make this clearer to our readers (cf. ll. 220-221):

*"Both the natural and man-made waterways need to be incorporated into the DEM."*

**7-    Line 231: why not using data from Phu An station?**

Thanks for this question, which we intensely discussed among the co-authors. In summary, we chose to build an expanded model of the HCMC area with a southern boundary at Nha Be in order to assess flood risk in all urban districts. As a consequence, the tidal boundary condition of the model needed to be located at that exact location. Given that no open-access data for this (or Phu An) station was available, we determined tidal water levels by extrapolating the accessible data from Vung Tau in the described way. After local partners have provided us with official time-series from the Nha Be gauging station, meanwhile, we can report that our approach yields a correlation coefficient of $R^2 = 0.96$ as can be seen from the newly included Figure 5. Considering that the distance between Nha Be and Phu An is much smaller than between Vung Tau and Nha Be, we are confident that a similarly high resemblance could be expected for the Phu An station.

**8-    Line 244: why an eight-day time series?**

An eight-day time series was chosen for two purposes. First of all, a certain spin-up time needs to be considered in the hydro-numerical model in order to allow the stabilization of water levels. Furthermore, adequate time needs to be given after precipitation occurs in order for rainfall runoff to properly route and concentrate within the model. Thanks to your comment, we have added these details to our text for better clarity (cf. ll. 335-337):

*"The eight-day timeframe was chosen for two purposes: first, to ensure a so-called spin-up time needed for the numerical stabilization of water levels, and second, to allow for physically realistic routing and concentration of rainfall runoff within the model domain."*

**9-    Line 261: why only a 2-year flood selected? How about 5-year, 10-year floods?**

The 1-year event was adopted here in order to highlight the substantial divergence in the literature in regards to the calculation of the precipitation depth. Furthermore, our focus was on more frequent, lower intensity rain events that regularly cause sustained socio-economic disruption in the city (ADB, 2010), but are currently underrepresented in scientific literature. Finally, the model had to be calibrated and validated against events, for which local inundation reports were available. Lower frequency events with certainly larger flood extents and depths could be easily simulated on this basis as well, but such scenarios of flooding would go beyond the original scope of this study.

**10-     Comment for the 2.2. section: How about reservoirs and groundwater? Why are these excluded from the model?**

The impact and presumably flood alleviating effects of reservoirs and groundwater are indeed considerable aspects of hydrological modeling. However, for the present setting, we have made the conservative assumption that aquifers are fully saturated during the rainy season, and thus no further infiltration, naturally draining the city, is possible. Certainly, this particularly holds true when considering the significant percentage of impermeable surfaces in the urban areas of HCMC. In regards to the upstream reservoirs, no positive effect on their potential in reducing inundation depth was reported, which aligns with our conservative engineering approach.

**11-     Method section: there should be one graph summarize the proposed methodology. So far, these are scatter over different section of each type of date, which is difficult to grasp the bif picture of the proposed methodology.  Besides, methodology for processing each element should be presented in equation form rather than figures (i.e., figure 2 and figure 7).**

Thank you for this comment. We made sure that additional processing steps, such as calibration and validation, are integrated in Figure 1 to make the whole process more accessible to a wide audience. It is true that equations could be used instead of Figure 2 and Figure 7 (now Fig. 9), but we would prefer to visualize the results of the processing and include the equations in the illustration, as it allows the reader to intuitively grasp what happens at each step of the work-flow.

**12-     Line 335: what do the authors mean by the threshold of the NFSI is at its maximum? Maximum of what? How are the factors of changing climate considered? Why did the authors give equal weights to flood depth and duration? How about flood frequency?**

We would like to clarify that it is not the threshold of the $I_{NFS}$ that is at its maximum but rather the combination of flood depth and duration over the threshold of 10 cm. For climate change considerations, the $I_{NFS}$ may readily be computed for that particular case and then normalized according to the base case without climate change effects are considered. Regarding the equal weight of both contributing factors, we intended to ensure that the results are not biased, especially considering the lack of additional data that are necessary to determine whether flood depth or duration play a bigger role in damage for a particular location. This weighting can, of course, be adjusted depending on the case and the local composition of flood damage. Future users are free to change the weighting and adapt it to a specific use case. Flood frequency can surely be considered when changing the boundary conditions of the model depending on the investigated return period. To that end, the IDF curves presented in Figure 6 offer a valuable starting point to estimate precipitation depth for HCMC. A sentence regarding the transferability (incl. the addressed climate change scenarios) of the $I_{NFS}$ was added (ll. 445-448).

*"Due to its normalization, the application of the $I_{NFS}$ is not restricted to singular analyses, but can also be considered as a metric to express changes in flood severity due to changing boundary conditions. For example, when taking climate change considerations into account, the $I_{NFS}$ can be computed for a particular case and then normalized according to the base case without climate change effects."*

In addition, the weighting was discussed in the following paragraph (ll. 614-617):

*"Equal weighting was given for both flood depth and duration to ensure that the results are not biased, especially considering the lack of additional data clarifying whether flood depth or duration plays a bigger role in damage for a particular location. This weighting can be different depending on the case and the local composition of flood damage. Future users are, of course, free to change the weighting and adapt it to a specific use case."*

[revised manuscript text omitted]

---

## Referee Report (RR1)

The revised manuscript "The Potential of Open-Access Data for Flood Estimations: Uncovering Inundation Hotspots in Ho Chi Minh City, Vietnam, through a Normalized Flood Severity Index" explored the usability and reliability of flood models built on open-access data in regions where highly-resolved (geo) data are either unavailable or difficult to access. The findings are helpful to enhance the interpretation and trustworthiness of risk assessments and the approaches may be replicated in other coastal megacities around the world. My main comments and suggestions are listed below:

1. I suggest replacing some of the references published in earlier years and increasing the citation ratio of papers published in authoritative journals in the past 5 years.

2. I suggest some of the figures or tables in the manuscript could be removed to the supplementary materials and kept the most critical ones in the manuscript.

3. I suggest adding subheadings to the discussion section to make it easier to understand.

4. This study has concluded many advantages of using open-access data for flood estimations in diverse ways. I suggest complementing some potential disadvantages and plights faced and stating the directions of further research.

5. Some of the figures need to be modified to improve visibility, for example, enlarging or bold the texts in Figures 3b and 3c.

---

## Author Response (AR2)

**Dear authors,**

**Thanks again for your efforts in improving the manuscript. Two reviewers have again provided comments to the updated version and both didn't raise any major problems. I agree with the reviewers' comments. Addition to that, i suggest the authors to carefully address all the minor issues in the latest comments, especially to improve the visibility of maps and figures. Therefore, I suggest an overall minor revision before it can be accepted for publishing.**

Dear Special Issue Editors, dear Dr. Yang,

We are very happy to learn that our efforts to clarify the original idea and motivation of our study in order to improve the initial manuscript are acknowledged. Based on the newly provided comments, we have undertaken another revision, specifically focusing on enhancing the readability of figures. Nonetheless, we felt obliged to carefully address all other comments as well. Thanks to your and the reviewers' comments, the quality of the manuscript could further be improved.

With kind regards,
Leon Scheiber, Mazen Hoballah Jalloul, Christian Jordan, Jan Visscher, Hong Quan Nguyen, and Torsten Schlurmann
* * *
RESPONSE TO REPORT #1

**The manuscript uses costalDEM, Bathymetric, River Discharge, Tidal, Precipitation and other Open-Access Data to construct the Normalized Flood Severity Index. The purpose of this study is to address the situation that data are difficult to obtain or insufficient in some areas, especially in developing countries. The study has important implications for the study of extreme events in the context of climate change, and for the promotion of compound events of flooding, tides, and sea level rise.**

**There are several minor issues in the manuscript, the presentation of the map needs to be more precise, and it is suggested to include the location of the study area in the country or region, the extent of the study area.**

Thank you for this critical feedback. We admit that our study area should be presented more clearly to the potential reader, especially to those who are not familiar with the situation in HCMC and the greater region. Therefore, we added a new figure (Fig. 1) to clarify the location of (a) the HCMC province in Southern Vietnam and (b) the HCMC urban districts in the larger

province. In addition, the caption references the model domain, which is then depicted in the "urban catchments" figure (now Fig. 3).

**The Hotspots of the model are more in line with the actual situation, could the authors do some comparison of the distribution of losses.**

Thank you for introducing this perspective to our discussion. It is indeed highly relevant for a systematic and exhausting risk assessment. However, we understand risk (of damages / losses) to be a combination of hazard, exposure and vulnerability as presented and discussed in the recent literature, e.g. SROCC (2019) etc. The proposed normalized flood severity index ($I_{NFS}$) highlights those areas with significant hazard potential as a consequence of high flood depths and long flood durations, but the estimation of (monetary) losses also requires knowledge about the remaining two components following the original concept. The corresponding socio-economic data were neither acquired, nor processed or discussed in the present study that, instead, promotes the idea of a generic hydro-numerical model to prepare flood estimates to pave the way for exactly those in-depth risk assessments that include all three contributing factors of risk. Any statement regarding the distribution of losses is therefore beyond the scope of this article, yet a worthwhile objective for future studies.

RESPONSE TO REPORT #2

**My main comments and suggestions are listed below:**

**1. I suggest replacing some of the references published in earlier years and increasing the citation ratio of papers published in authoritative journals in the past 5 years.**

Thank you for this suggestion. We admit that our list of references is relatively long and includes some references which are considerably older than five years. Nevertheless, we think that we managed to document and critically review the current state of research in this field, accurately. In our opinion, the number of cited articles (N > 160) does also reflect the extensive efforts and care we put into reviewing literature, not only about the situation at our study site, Ho Chi Minh City, but also about the status of freely available data and its potential application in numerical modelling. After critically discussing the pros and cons of reducing the list of references among all co-authors, we finally decided against removing individual references. We are convinced that, in this way, we can provide a comprehensive work of references that can be of use to any upcoming study in the field of modelling floods with freely available data and subsequent implementation campaigns for risk reduction. We hence hope for your understanding.

**2. I suggest some of the figures or tables in the manuscript could be removed to the supplementary materials and kept the most critical ones in the manuscript.**

Thank you for reminding us to be concise in our presentation of methodological steps and results. Based on your meaningful comment, we decided to shift the following figures and tables to the Supplementary Material of this article:
- Supplementary Material A (originally Tab. 1: Freely available DEMs),
- Supplementary Material B (originally Fig. 2: Terrain Data),
- Supplementary Material E (originally Fig. 3: River Bathymetry),
- Supplementary Material F (originally Fig. 5: Comparison of water levels)

**3. I suggest adding subheadings to the discussion section to make it easier to understand.**

We agree that subheadings are useful to enhance readability and allow easy orientation within the discussion section. Accordingly, we introduced the following subheadings:

*4.1 Accuracy of Elevation Data*
*4.2 Sensitivity of Boundary Conditions*
*4.3 Significance of Modelling Results*

**4. This study has concluded many advantages of using open-access data for flood estimations in diverse ways. I suggest complementing some potential disadvantages and plights faced and stating the directions of further research.**

While happily implementing the above suggestion of introducing subheadings to the discussion section (chapter 4), we have ensured that already these headings pinpoint the disadvantages (limitations, assumptions or generalizations) of our approach. Furthermore, we have added an introductory paragraph that outlines the general limitations inherent to the individual components of the presented workflow. This section reads as follows (ll. 463-472):

*"Like in any other scientific discipline, every hydro-numerical model is subject to limitations. This applies particularly for a model, which exclusively draws on freely available data as envisioned in the presented framework. In this case, each of the model in- and outputs have to be evaluated in terms of their accuracy, reliability and significance. For instance, the topographic data come with a limited spatial resolution and uncertain vertical error. Significant differences between the available hydro-meteorological time series suggest a source of error as well. And finally, the essential validation of modelling results, in many cases, has to be based on citizen and media reports whose scientific standards cannot be taken for granted. Although this has to be seen as a disadvantage compared to studies that have the privilege to build on official and high-resolution data, the majority of inherent limitations can still be rebutted and accepted if taken into account reasonably. Nevertheless, the only valid argument against*

*infinitely increasing the level of detail of a model remains (acquisitional and computational) cost, so that high-resolution data should always be incorporated where accessible."*

In addition, we also complemented the discussion section with a paragraph on further research needs in the field of flood modelling with open-access data. The chapter now ends as follows (ll. 576-579):

*"Nevertheless, independent studies should apply the normalized flood severity index to other regions with comparable risk settings. The envisaged flood estimates may then be juxtaposed with sophisticated loss calculations, in order to further quantify the sensitivity and scrutinize the robustness of the proposed framework."*

**5. Some of the figures need to be modified to improve visibility, for example, enlarging or bold the texts in Figures 3b and 3c.**

Thank you for this valuable remark. There have, indeed, been some issues with the conversion of individual figures, which impaired readability in the final PDF version of our manuscript. We reviewed our illustrations (with a special focus at Figure 3) and, hopefully, managed that all figures are now rendered correctly.